# Dopamine and glutamate regulate striatal acetylcholine in decision-making

Lynne Chantranupong[1], Celia C. Beron[1], Joshua A. Zimmer[1], Michelle J. Wen[1], Wengang Wang[1] & Bernardo L. Sabatini[1✉]

Striatal dopamine and acetylcholine are essential for the selection and reinforcement of motor actions and decision-making[1]. In vitro studies have revealed an intrastriatal circuit in which acetylcholine, released by cholinergic interneurons (CINs), drives the release of dopamine, and dopamine, in turn, inhibits the activity of CINs through dopamine D2 receptors (D2Rs). Whether and how this circuit contributes to striatal function in vivo is largely unknown. Here, to define the role of this circuit in a living system, we monitored acetylcholine and dopamine signals in the ventrolateral striatum of mice performing a reward-based decision-making task. We establish that dopamine and acetylcholine exhibit multiphasic and anticorrelated transients that are modulated by decision history and reward outcome. Dopamine dynamics and reward encoding do not require the release of acetylcholine by CINs. However, dopamine inhibits acetylcholine transients in a D2R-dependent manner, and loss of this regulation impairs decision-making. To determine how other striatal inputs shape acetylcholine signals, we assessed the contribution of cortical and thalamic projections, and found that glutamate release from both sources is required for acetylcholine release. Altogether, we uncover a dynamic relationship between dopamine and acetylcholine during decision-making, and reveal multiple modes of CIN regulation. These findings deepen our understanding of the neurochemical basis of decision-making and behaviour.

The basal ganglia are a group of interconnected subcortical nuclei that integrate information from multiple brain centres to modulate goal-directed behaviour. The striatum is the principal input structure of the basal ganglia, and its function is controlled by a complex array of neurotransmitters and neuromodulators[2,3]. Among these is dopamine (DA), which is released in the striatum by long-range axons arising from midbrain ventral tegmental area (VTA) and substantia nigra pars compacta (SNc) neurons[4–6]. DA neurons (DANs) are thought to drive reinforcement learning by encoding reward prediction error—the difference between experienced and expected reward—and, mechanistically, by regulating multiple aspects of neuronal and synapse function[7–10]. Disruption of DA signalling contributes to many debilitating psychomotor disorders, including Parkinson's disease and drug addiction[11].

In addition to having the highest concentrations of DA and DA receptors in the mammalian brain, the striatum also contains some of the highest levels of acetylcholine (Ach)[12,13], which is primarily released by local CINs, a specialized and rare cell type[14,15]. Pioneering studies in primates revealed that CINs reduce or 'pause' their firing in response to both appetitive and aversive stimuli over the course of learning, leading to the hypothesis that they modulate reinforcement learning[15–19]. CIN pauses in turn might alter the plasticity of corticostriatal synapses to support procedural learning[20].

Bidirectional interactions between DA and Ach release have long been observed within the striatum during learning and in Parkinson's disease[15,16,21,22]. Subsequent in vitro studies uncovered a striatal circuit by which DA and Ach directly influence each other. Synchronized firing of multiple CINs activates nicotinic Ach receptors that are located on and depolarize DAN axons[23–26]. If of sufficient amplitude, this depolarization induces a propagating axonal action potential that evokes the release of DA within the striatum[25] (Fig. 1a). In turn, DA potently inhibits the activity of CINs by acting on D2Rs expressed by CINs[27–29] (Fig. 1a).

Despite a detailed mechanistic understanding of the interactions between DA and Ach in vitro, if, when and how these control the levels of DA and Ach to regulate striatal function in vivo are largely unknown. It is unclear whether sufficient CIN synchronization occurs and to what degree nicotinic Ach receptors are available to evoke DA release in vivo[30], nor is it known when and if the potential influence of CINs on DA signalling is functionally important. Although CIN-evoked DA release has been proposed to explain differences between DAN somatic activity and striatal DA levels during motivated approach behaviours and longer timescales of reward-value encoding[31], previous studies report a robust correlation between somatic and axonal signalling[32,33]. Finally, although CIN pauses can be induced by D2R activation, they can also be triggered by cortical and thalamic projections and GABAergic VTA inputs[34–36]. Indeed, CIN-specific deletion of the gene encoding D2Rs reduces, but does not abolish, the CIN pause in a reward-based task, suggesting that other sources of modulation exist[37].

[1]Department of Neurobiology, Howard Hughes Medical Institute, Harvard Medical School, Boston, USA. ✉e-mail: bsabatini@hms.harvard.edu

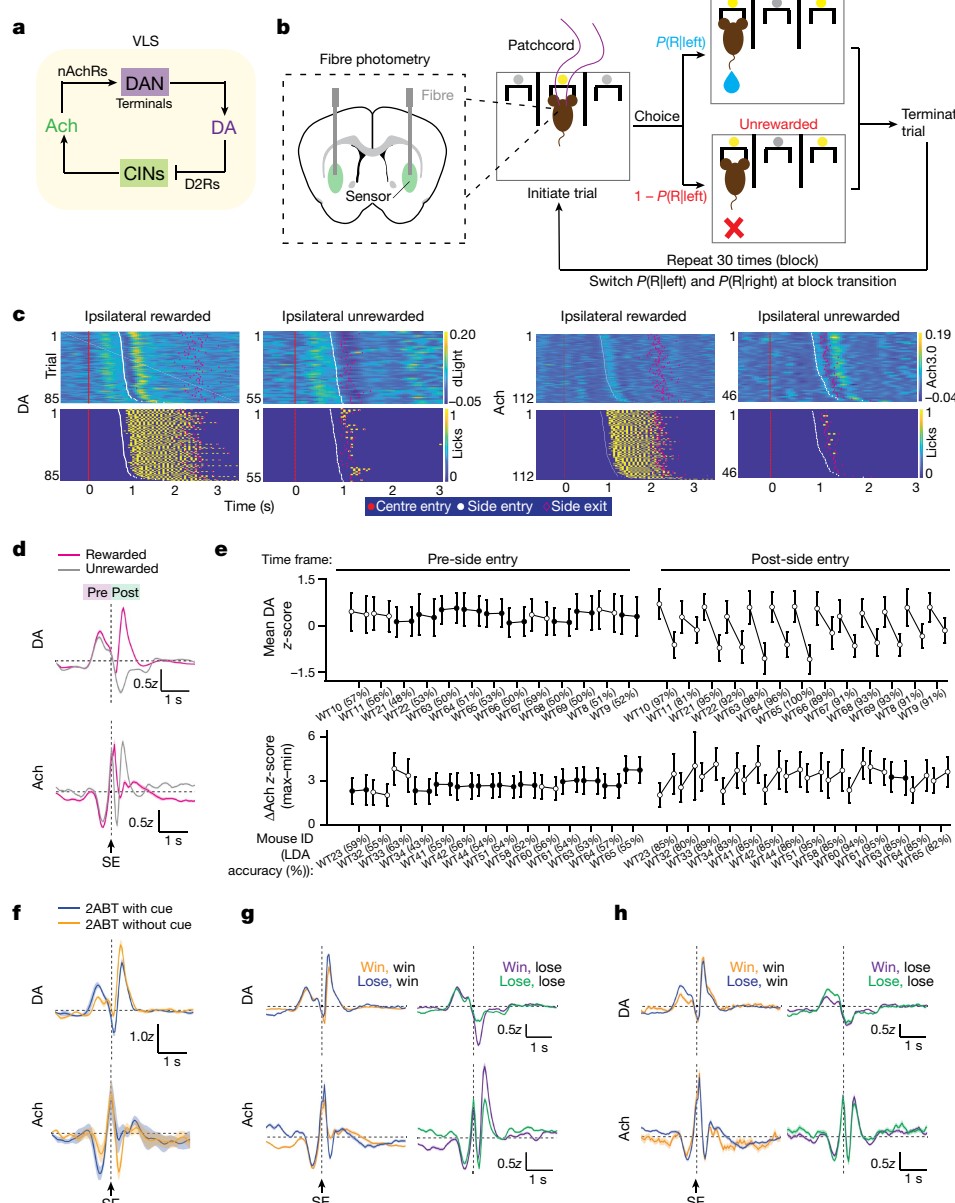

**Fig. 1 | Multiphasic dynamics of DA and Ach in the VLS during reward-based decision-making. a**, Proposed DA and Ach interactions. CINs release Ach, which evokes the release of DA through nicotinic acetylcholine receptors (nAchRs) on DAN terminals. Conversely, DA inhibits CINs through D2Rs. VLS, ventrolateral striatum. **b**, 2ABT parameters. An LED (yellow) signals trial initiation. A single reward is probabilistically delivered when the mouse makes the correct choice (*P*(R|left) is the probability of a reward delivered at the left port and *P*(R|right) is the probability of a reward delivered at the right port), and the trial is terminated with side-port exit. Fibre photometry is simultaneously performed. **c**, Ipsilateral DA and Ach dynamics and licks (yellow) recorded from an example mouse during a 2ABT session. Each row depicts the *z*-scored sensor signal of a trial. **d**, DA and Ach signals during different reward outcomes. The averaged *z*-scored signal ± s.e.m. is shown. Data are aligned to side-port entry

(SE) (DA: *n* = 13 mice; Ach: *n* = 14 mice). **e**, Statistical analysis of **d**. Rewarded versus unrewarded trials are compared (left and right dot in connected pair, respectively). Open circles denote a significant difference for each comparison (two-sided *t*-test (*P* < 0.05)). Mean DA and ΔAch ± s.d. are shown. Percentages represent LDA classification accuracy. **f**, DA and Ach release during rewarded 2ABT trials in which the LEDs that signal centre-port and side-port entry are present or omitted. Data are shown as in **d** (*n* = 4 mice). **g**, DA and Ach release for trials in which mice choose the same port in both the previous and the current trials, which are segregated by the reward outcome of the previous trial. Data are shown as in **d** (DA: *n* = 13 mice; Ach: *n* = 14 mice). **h**, DA and Ach release for trials in which mice switch ports. In **g**,**h**, the text denotes previous outcome followed by current outcome (for example, Win, win). Data are shown as in **d** (DA: *n* = 13 mice; Ach: *n* = 14 mice).

## Adaptive switching in decision-making

To examine the local circuit interactions between striatal Ach and DA during, and their contributions to, such behaviours, we monitored the levels of neuromodulators in mice performing a dynamic and probabilistic two-port choice task modelled after paradigms that engage striatal pathways and require striatal activity for optimal performance[38–40].

We used only male mice to avoid the variability of cholinergic signalling in females (Supplementary Information). In this two-armed bandit task (2ABT), mice move freely within a box that contains three ports separated by physical barriers (Fig. 1b). An LED above the centre port signals that the mouse can initiate a trial by placing its snout ('poking') into the centre port. The mouse must then choose to poke into either a left or a right port, each of which probabilistically delivers water after

snout entry. In a block structure (30 rewards between block transitions), either the left or the right port is designated as 'high reward probability' ($P$(reward) = $P_{high}$) and the other port as 'low reward probability' ($P$(reward) = $1 - P_{high}$). To efficiently obtain rewards, the mouse learns which is the high-reward-probability port in that block and detects when block transitions occur. This task structure requires mice to use flexible decision-making strategies and to integrate information about previous trial outcomes to make a choice.

Mice robustly alter their port selections at block boundaries to repeatedly choose the highly rewarded port and occasionally sample the low-reward-probability port (Extended Data Fig. 1a). After successive unrewarded trials resulting from reversals of the reward probability at block transitions, the mice transiently increase their probability of switching ports between trials ($P$(switch)) (Extended Data Fig. 1b), which facilitates the selection of the new high-reward-probability port ($P$(high port)) (Extended Data Fig. 1c). As a result of this behavioural flexibility, proficient mice achieve rapid decision times and high reward rates (Extended Data Fig. 1d). During the 2ABT, we capture the timing of key behaviour events, including the timing of port entries and withdrawals, the timing and number of licks at each port and the reward outcomes (Extended Data Fig. 1e). Entry into and exit from the centre port occur in rapid succession, followed by a delayed entry into the side port (Extended Data Fig. 1e). In rewarded trials, the water reward is triggered by entry into the side port, and mice repeatedly lick to consume the reward, whereas in unrewarded trials the mice rarely lick the port (Fig. 1c and Extended Data Fig. 1e).

To investigate the mouse behaviour and evidence accumulation in the task, we used a recursively formulated logistic regression (RFLR) that was developed from a 2ABT[40]. In this linear model, the conditional probability of the mouse's next choice is based on a latent representation of evidence about the interaction between its actions and reward outcome (i.e., action value). This variable decays over time and is recursively updated by new evidence from each trial's choice and outcome. There is additional bias towards or away from the mouse's most recent choice (Extended Data Fig. 1f). The RFLR model uses three parameters to capture, respectively, the tendency of an animal to repeat its last action (alpha, $\alpha$), the relative weight given to information about past action and reward (beta, $\beta$) and the time constant over which action and reward history decay (tau, $\tau$) (Extended Data Fig. 1f). The RFLR coefficients are comparable across mice that are proficient on the 2ABT (Extended Data Fig. 1g). Moreover, the RFLR model accurately predicts the switching dynamics at block transitions as well as the probability of switching on the current trial, which depends on the choice and reward history of previous trials (Extended Data Fig. 1h–j). Altogether, mice achieve high proficiency on a probabilistic reward task, and their behaviour is accurately captured by a reduced logistic regression model.

## Ach and DA are dynamically regulated

To determine how DA and Ach signals change during the 2ABT, we used frequency modulated fibre photometry to record the fluorescence of the genetically encoded sensors for DA (dLight1.1)[41] and Ach (GRAB-Ach3.0, abbreviated as Ach3.0)[42] expressed in separate hemispheres within the ventrolateral portion of the dorsal striatum (VLS), a region associated with controlling the behaviour of mice in reward-based decision-making tasks[43] (Extended Data Fig. 2a–d,i,j). We observed robust and multiphasic DA and Ach transients in individual trials that differed depending on reward outcome (Fig. 1c,d), and that depended on neuromodulator binding, because they were absent in ligand-binding-site mutants of the sensors (Extended Data Fig. 2k–p).

To understand which behavioural features affect DA and Ach transients, we compared their profiles during rewarded and unrewarded trials. As expected, DA signals changed at the instances of task-relevant behavioural events, and they diverged depending on reward outcome (Fig. 1d). To quantify these signals, we identified a single metric that best captured the changes in each neuromodulator across varying trial types, and we performed comparisons between pairs of signals in the denoted conditions. For DA, we calculate the mean of the $z$-scored signal in the designated time range before or after side-port entry (Extended Data Fig. 3a). For Ach, we take the difference in the maximum and minimum signal in a defined time window, which we call ΔAch (Extended Data Fig. 3b). Owing to the multiphasic nature of Ach transients, this metric captures differences in Ach signals across conditions more accurately than the mean. Reward outcome greatly alters DA and Ach transients, with unrewarded trials resulting in a robust decrease in mean DA and a consistent increase in ΔAch (Fig. 1e). Notably, the changes are significant after ('post'), but not before ('pre') side entry, and these signals are not lateralized (Supplementary Information).

Because our transients are complex, a single metric such as mean DA or ΔAch captures limited features of these signals. Therefore, we complemented this analysis with a supervised classification approach—linear discriminant analysis (LDA)—to quantify the degree to which the waveform of the photometry signals differs across conditions and the degree to which trial-by-trial signals can be used to classify the trial type (Extended Data Fig. 3c). Supporting our observation that reward outcome greatly alters DA and Ach transients, but only post-side entry, the LDA classification accuracy is very high when trained on the signals post-side entry (higher than 80%), but not when trained on those pre-side entry (around 50%) (Fig. 1e).

In contrast to past reports that striatal Ach is outcome-insensitive[17], we found that reward outcome robustly modulates Ach transients (Fig. 1d). Further support for reward-outcome modulation of both DA and Ach is revealed during sessions in which the LED cues that signal trial initiation and side-port entry are off but all other 2ABT conditions remain the same. In this scenario, rewards are less expected owing to the absence of the LED cues that normally signal that the centre and side ports are active. Consistent with this change in expectation, the DA transient in cue-omission rewarded trials is significantly increased after side-port entry and decreased before side entry, whereas Ach transients are modulated in the opposite direction (Fig. 1f and Extended Data Fig. 4a).

Choice and reward histories are integral to the decision-making process and can lead to different reward expectations, the effects of which on Ach signalling are poorly understood. We subdivided rewarded and unrewarded trials by the task history and found that the outcome of the previous trial strongly modulates DA and Ach signals in the current trial. When a mouse chooses the same port in two consecutive trials, a rewarded trial following a previously unrewarded trial ('lose–win') is more unexpected than a rewarded trial that follows a win ('win–win'). Indeed, mean DA and ΔAch signals increase post-side entry in the 'lose–win' scenario, reflecting different reward expectations owing to past experience (Fig. 1g and Extended Data Fig. 4d,e). Conversely, for an unrewarded trial, mean DA dips more and ΔAch rises more if this trial was preceded by a rewarded trial ('win–lose') rather than by an unrewarded trial ('lose–lose') (Fig. 1g, right and Extended Data Fig. 4d,e). Altogether, this is consistent with the encoding of reward prediction error. Notably, these effects of expectation on DA and Ach are absent if the mouse switches ports between trials—the signals after side-port entry are similar (Fig. 1h and Extended Data Fig. 4d,e). Instead, the DA signals during the transition from centre to side port are greater when the previous trial was unrewarded (Fig. 1h). Thus, when a mouse chooses to switch ports between trials, it approaches this choice in a different state shaped by outcome history; however, it resets any history-dependent reward expectation in the post-side-entry period, during which the mice evaluate the reward outcome. Analysis of the intertrial interval (ITI) signals reveals that, although the motor action may contribute to DA and Ach dynamics, reward expectation significantly shapes both neuromodulator transients (Supplementary Information). Altogether, prior choice and reward experience modulate both DA and Ach.

Finally, we observed that changes in DA and Ach are often temporally coincident but in the opposite direction; however, the relationship between DA and Ach is neither simple nor fixed, as there are periods in which both signals go up or down synchronously or independently, suggesting that there is a flexible and dynamic interaction between the two neuromodulators.

## Action-outcome history shapes DA and Ach

To evaluate the contribution of each behavioural event to DA and Ach dynamics formally and quantitatively, we developed a generalized linear model (GLM) to predict neuromodulator signals from behaviour (Supplementary Information). In our simplest GLM model, which we term the 'base GLM', we included variables based on key behavioural events. We find that it captures substantial variance across the trial-associated data (DA GLM $R^2$ = 0.206; Ach GLM $R^2$ = 0.206) (Extended Data Fig. 5c,d) and its performance is comparable to other GLMs that are used to predict photometry signals[33]. To assess the degree to which each behavioural variable contributes to GLM performance, we performed a 'leave-out analysis' in which we iteratively omit a single behavioural feature and evaluate the GLM performance (Extended Data Fig. 5f). For both DA and Ach models, the closely timed centre entry and centre exit are redundant because loss of either alone does not affect the model fit. By contrast, the omission of several variables greatly increased the mean squared error (MSE) and thus weakened the performance of the GLM, indicating that the inclusion of these variables is necessary to successfully capture the variance in the neural signal. DA and Ach signals are more accurately reconstructed with the addition of side entry and reward predictors, and, additionally, side exit and lick enable better reconstructions of the Ach signal. This analysis highlights the unique influence of each behavioural event on DA and Ach dynamics.

Although the base GLM robustly reconstructs the measured signals of both Ach and DA transients, there are discrepancies across several trial histories (Extended Data Fig. 5c,d). Given the importance of choice and reward history for modulating the signals of both neuromodulators (Fig. 1g,h), we expanded the feature set of the base GLM to include side-port entries segregated by the eight possible action-outcome combinations, which we term the 'history GLM' (Extended Data Fig. 5g,h). Inclusion of these parameters reduced the MSE between the predicted and the test data for both DA and Ach GLMs (Extended Data Fig. 5f, '+ history'). This reflects an improvement in the ability of the history GLMs to capture the variance of the trial-associated data (DA GLM $R^2$ = 0.213; Ach GLM $R^2$ = 0.214) without overfitting, despite the addition of multiple parameters (Extended Data Fig. 5i). Altogether, by modelling DA and Ach signals with GLMs, we reveal the influence of multiple and different behavioural variables and action-outcome history on the dynamics of each neuromodulator during decision-making.

## Ach and DA release are anticorrelated

Because DA and Ach might directly interact in vivo, we characterized the relationship between their signals to determine whether they support the proposed interactions. To more accurately assess the dynamics of and relationship between DA and Ach transients, we performed simultaneous recordings of both neuromodulators within the same hemisphere by coexpressing a red-shifted DA sensor, rDAh[44] and the green Ach sensor (Fig. 2b,c and Extended Data Fig. 6k,l). To understand the effect of switching DA sensors, we exploited the fact that the release of DA is highly correlated across hemispheres within the same mouse (Extended Data Fig. 6c), allowing us to directly compare DA signals detected by rDAh versus dLight1.1. Both sensors yield comparable signals, but with consistently reduced amplitudes for rDAh (Fig. 2a and Extended Data Fig. 6d,i,j), probably reflecting its slower kinetics and higher affinity for DA compared to dLight1.1.

Simultaneous DA and Ach recordings within the same hemisphere (Fig. 2b) reveal that DA and Ach responses are highly anticorrelated with a positive time lag, which indicates that increases in DA might suppress Ach with a short delay (Fig. 2d)—a finding that is recapitulated by recordings of DA and Ach across separate hemispheres (Supplementary Information). To examine whether the relationship between DA and Ach varies across trial types, we analysed the cross-correlation between these signals. We observed that the trial-segregated DA and Ach signals are anticorrelated with a time offset of around 100 ms across rewarded and unrewarded trials (Fig. 2e and Extended Data Fig. 6f). Because correlations between signals might be driven by external factors such as behavioural events, we also examined the cross-correlation of the fluctuations about the trial-averaged means (that is, 'noise correlations') for rewarded and unrewarded trials (Extended Data Fig. 6e). This revealed a similar correlation structure (Fig. 2e), suggesting that direct interactions exist between Ach and DA release, with DA potentially inhibiting the release of Ach. We complement this analysis with a GLM that incorporates photometry as a predictive variable, and this reveals a similar negative interaction between DA and Ach (Fig. 2h and Supplementary Information).

Cross-covariance analysis, as presented above, assumes that the mean and variance of the signal do not change over time, but these can be dynamic during behaviour. To account for this, we performed a covariance analysis in which we calculate how variance about the mean of DA at one time point ($t_1$) influences the variance in Ach at another time point ($t_2$) (see Methods and Extended Data Fig. 6g). This results in a two-dimensional function, $K(t_1,t_2)$, that describes the relationships between fluctuations in DA and Ach at specific times, such as entry into the side port. This revealed a strong time-lagged negative covariance (Fig. 2f), which we call the off-diagonal (Fig. 2g), showing that, at most time points, changes in DA precede changes in Ach by approximately 100 ms, which is consistent with prior analysis (Extended Data Fig. 6f). Notably, the negative, off-diagonal covariance nearly disappears when the mouse enters the side port (Fig. 2f, insets). This analysis highlights the dynamic and context-dependent relationship between DA and Ach within the trial and during the ITI. Phasic increases in DA typically inhibit the release of Ach, consistent with D2R-mediated suppression of CIN activity, but at specific moments, such as when the mouse enters the side port, this negative correlation is weakened.

## Striatal DA dynamics do not require CINs

The DA transients we observe during the 2ABT could be driven by the release of Ach from CINs, by the activity of DANs or by a combination of both. To determine whether the release of Ach from CINs contributes to DA release, we blocked Ach release from CINs in the VLS by expressing tetanus toxin (TelC), which prevents the fusion of synaptic vesicles in these cells (Fig. 3a,b) and potently inhibits the release of Ach from CINs in vitro (Fig. 3c) and in vivo (Fig. 3d and Extended Data Fig. 7g–j). Owing to the large extent of DAN axon arborization, we reasoned that synchronized CIN activity across the striatum might be sufficient to drive DA release within the VLS where we record. Therefore, we perturbed CINs through TelC expression in a striatum-wide manner using a multisite injection approach (Fig. 3e and Extended Data Fig. 7l–o). This widespread loss of Ach induced severe behavioural defects and greatly altered behaviourally evoked DA signals (Fig. 3e and Extended Data Fig. 9f–i). The marked changes in behaviour underscore the importance of CINs in regulating striatal function; however, they make it difficult to interpret the effects of Ach loss on the reward-encoding properties of DA. Nevertheless, DA retained its capacity to encode for reward, such that DA signals (Fig. 3e) and their associated GLM kernels (Extended Data Fig. 7k) maintain opposing polarity with reward outcome. Thus, reward-encoding features of DA can persist despite severe loss of Ach.

Given that the proposed mechanism is local—CIN activity triggers DA release within a local DAN axon field—we tested whether DA

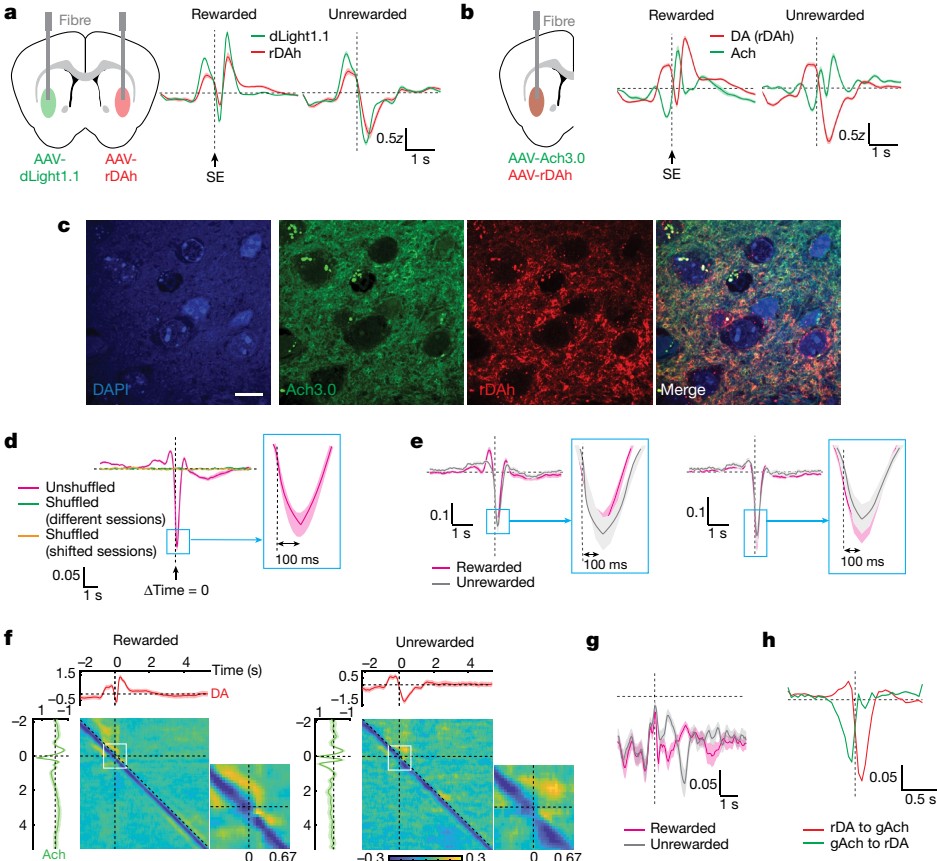

**Fig. 2 | Ach and DA signals are dynamically correlated during reward-based decision-making. a**, DA release detected by dLight1.1 and rDAh recorded from bilaterally injected mice (left). The average $z$-scored sensor signal ± s.e.m. is shown (errors are often smaller than line thickness) ($n$ = 3 mice). AAV, adeno-associated virus. **b**, Overlay of simultaneously recorded DA and Ach dynamics, and schematic of the injection strategy. Data are shown as in **a** ($n$ = 6 mice). **c**, Confocal images of sensor expression in neurons of the VLS for a representative mouse recorded in **b**. DAPI is a nuclear marker. Scale bar, 10 μm. **d**, Covariance of DA and Ach signals from a 2ABT session in which DA lags Ach. Ach signals are compared to DA signals from another session (green) or randomly shifted signals from the same session (orange). The average covariance ± s.e.m. is

depicted ($n$ = 6 mice). **e**, Covariance of trial-segregated DA and Ach signals (left) and their noise (right) in which DA lags Ach by the indicated time. Data are shown as in **d**. The insets highlight the time offset of the minimum covariance signal. **f**, Full time-dependent covariance analysis of DA and Ach signals. The average signals ± s.e.m. are shown within the top and left subplots. An enlarged view of the outlined region in white is shown to the right with the time (s) indicated ($n$ = 6 mice). **g**, Summary of the off-diagonal negative covariance calculated from the matrices in **e**. **h**, Photometry kernels produced by a GLM that incorporates behavioural, history and photometry variables. The mean kernels ± s.d. that predict Ach signals from rDAh signals (rDA to gAch) and DA signals from Ach3.0 signals (gAch to rDA) are shown ($n$ = 6 mice).

release is affected by VLS-selective inhibition of Ach release. In mice performing the 2ABT, we inhibited the release of Ach in the VLS of one hemisphere using TelC, and compared DA release to that of the other hemisphere, in which VLS CINs express a control protein (Fig. 3f, left and Extended Data Fig. 8a–c). Mice did not exhibit behavioural deficits (Extended Data Fig. 9j–m), and VLS-specific loss of Ach release did not affect DA dynamics in trials recorded with dLight1.1 (Fig. 3f) or with rDAh, in which we simultaneously validated the suppression of Ach release (Extended Data Figs. 8g,i and 9a,b). Notably, phasic DA release remained the same during motivated approach behaviours (that is, around centre-port entry and immediately before side-port entry), which are proposed to be periods during which CINs could drive DA release, because DA levels and DAN activity are poorly correlated[31]. To address whether CINs might mediate discrepancies between DAN firing and DA release across longer timescales[31], we parsed DA signals by the choice and reward outcome histories of one, two and three trials back (Extended Data Fig. 8d,h,j,k), but we did not observe any significant changes in DA dynamics or in the underlying GLM kernels of DA for each input feature after loss of Ach (Extended Data Fig. 8e). Finally, to address whether Ach loss lowers the overall magnitude of DA release throughout the trial, we analysed the amplitudes of DA

sensor fluorescence transients ($\Delta F/F_0$), but this analysis did not reveal consistent effects (Extended Data Fig. 8f).

Because the DA transients we observe during the 2ABT are not affected by local Ach release, DAN activity is likely to be the major driver of DA dynamics, not CINs. Indeed, inhibition of DAN activity robustly alters DA release, as evidenced by a significant reduction in the levels of DA after optogenetic inhibition of DANs with stGtACR2 (Fig. 3g and Extended Data Fig. 9c–e). Altogether, we find that loss of Ach release within the VLS does not impair DA dynamics. Although modulation of CIN activity is sufficient to drive DA release in vivo (Supplementary Information), the context in which it does so remains to be determined.

## DA inhibits Ach release through D2Rs

During a trial, there are two periods in which opposite-signed changes in DA and Ach signals coincide and during which we hypothesize that D2Rs might mediate the depression of Ach: first, as the mice move from centre to side port; and second, during rewarded trials after side-port entry. To test this, we assayed whether optogenetic manipulations of DA neurons in vivo affect striatal Ach levels in a D2R-dependent manner. We increased and decreased the levels of DA in the VLS through

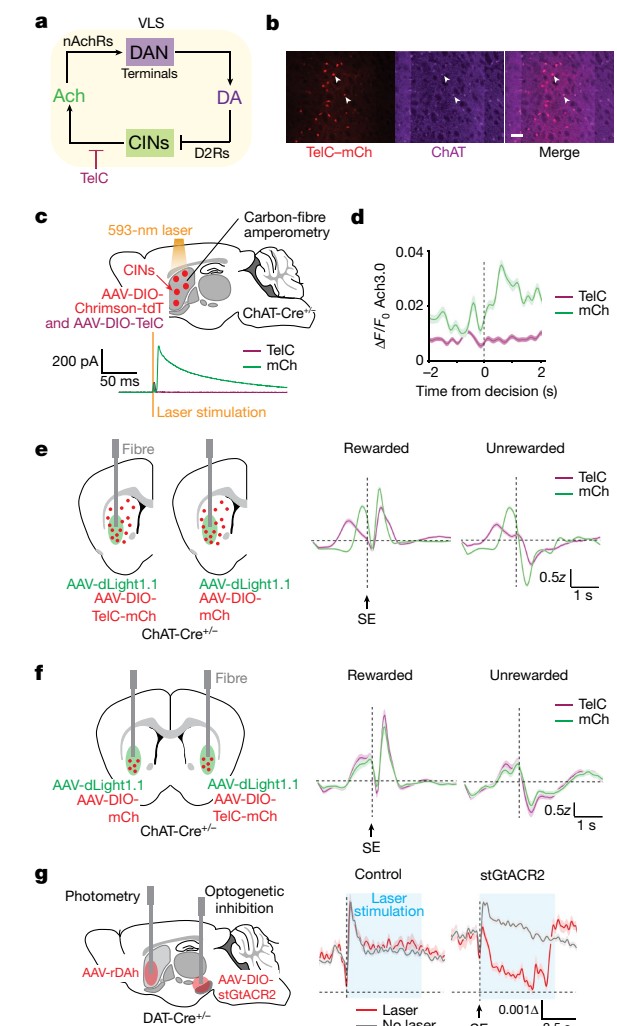

**Fig. 3 | Ach does not regulate DA dynamics during decision-making.**
**a**, Schematic of TelC perturbation of Ach release. **b**, Epifluorescence images of TelC linked to mCherry (TelC–mCh) expressed in choline acetyltransferase (ChAT)-positive cells in the VLS ($n = 12$ mice). White arrowheads denote two CINs that coexpress ChAT and TelC. Scale bar, 50 μm. **c**, DA release as measured by carbon-fibre amperometry in an acute striatal slice containing CINs coexpressing Chrimson with mCherry (mCh) or TelC. Amperometry recordings (mean ± s.e.m.) are aligned to laser stimulation. **d**, Ach release in the VLS during rewarded 2ABT trials recorded with fibre photometry from mice with CINs expressing mCh or TelC in the VLS. The average $\Delta F/F_0$ of the sensor signal ± s.e.m. is shown ($n = 3$ mice). **e**, Injections and implantations (left) for fibre photometry to measure DA release (right) in the context of striatum-wide expression of mCh or TelC in CINs. Unilateral injections were performed in two separate cohorts of mice. The side entry (SE)-aligned average $z$-score of the sensor signal ± s.e.m. is shown ($n = 5$ mice per condition). **f**, Injections and implantations (left) for fibre photometry to measure DA release (right) in the context of VLS-selective and CIN-specific expression of TelC or mCh in separate hemispheres of the same brain (left). Data are shown as in **e** ($n = 4$ mice). **g**, Optogenetic inhibition of DAN cell bodies with simultaneous recordings of DA release in the VLS. Schematic of the injections and implantations (left), and summary of DA release during rewarded 2ABT trials from mice lacking opsin expression (middle) or expressing the inhibitory opsin stGtACR2 (right). The average $\Delta F/F_0$ of rDAh ± s.e.m. is indicated ($n = 3$ mice).

photoactivation of DAN somas with excitatory (Chrimson) and inhibitory (stGtACR2) optogenetic proteins, in a head-fixed mouse on a wheel (Fig. 4a and Extended Data Fig. 10a–f). These manipulations altered the levels of Ach in the direction opposite to optogenetically evoked changes in DA levels, consistent with DA inhibiting Ach release

(Fig. 4b,c). Notably, these effects are D2R-dependent as they are abolished by the administration of eticlopride, a D2R antagonist (Fig. 4b,c). Thus, changes in DA are sufficient to bidirectionally regulate the levels of Ach in vivo, consistent with basal engagement and dynamic modulation of DA-dependent inhibition of CINs.

Because D2Rs are expressed by other cell types in the brain and D2R blockade has major behavioural effects that prevent mice from performing the task[45], we used an alternative method to determine whether DA suppresses the release of Ach during the task. We used a genetic strategy to knock out (KO) D2Rs specifically in CINs, and we refer to this transgenic mouse line (*ChAT-IRES-Cre; Drd2$^{f/f}$*) as Drd2-cKO. To confirm the functional loss of D2Rs in CINs, we compared the ability of DA to reduce CIN firing in striatal slices from wild-type versus Drd2-cKO mice (Fig. 4d). In wild-type mice, release of DA after laser stimulation of channelrhodopsin-expressing DAN terminals robustly reduced CIN firing, as measured by cell-attached recordings, but this effect was absent in CINs from Drd2-cKO mice (Fig. 4e,f).

To determine how D2R loss in CINs affects the release of Ach during the 2ABT, we compared neuromodulator dynamics in the VLS of Drd2-cKO CIN mice with that in two control groups: *ChAT-IRES-Cre* mice (referred to as wild type), and *Drd2*-floxed mice (referred to as Drd2 f/f) (Extended Data Fig. 10g,h). We found that the loss of D2Rs in CINs abolished both instances of Ach suppression that coincide with an increase in DA levels (Fig. 4g). Together, these changes lead to significantly increased Ach signals (Extended Data Fig. 11d). Modelling these signals with the history GLM recapitulates these effects (Supplementary Information). Of note, these changes occurred despite no significant changes in DA dynamics during the task across the three genotypes (Fig. 4g and Extended Data Fig. 11d), which, in turn, provides further support for the fact that Ach release has little or no effect on DA signals. Thus, D2Rs are required for DA to inhibit Ach release in vivo during precise moments within a trial.

To determine whether the loss of D2Rs in CINs affects decision-making, we assessed the performance of Drd2-cKO mice in the 2ABT. Although general performance metrics, block transition dynamics and RFLR coefficients (Extended Data Fig. 11f–i) are comparable between Drd2-cKO mice and both control groups, differences emerge when performance is parsed by history. Drd2-cKO mice are impaired in their ability to switch selection ports across some histories when compared to both Drd2 f/f and wild-type cohorts (Extended Data Fig. 11j). This supports a role for D2R-dependent reductions in Ach release in promoting complex changes in behaviour. In conclusion, we find that D2Rs are required for DA to repress Ach signals during precise moments within a trial, and loss of this regulation impairs the normal switching behaviour of mice.

## The cortex and thalamus drive Ach release

Although DA shapes Ach signals during decision-making, we observe additional fluctuations in Ach that are independent of DA. During unrewarded trials, Ach signals remain repressed after side-port entry, even in Drd2-cKO mice (Fig. 4g), and extra inputs are required to drive the increases in Ach that occur upon side-port entry and during the consumption period. Finally, the momentary disruption of the negative covariance between Ach and DA signals points to the existence of other factors that can independently alter Ach and DA dynamics (Fig. 2f,g).

To discover other potential sources of regulation of striatal Ach, we examined inputs to the striatum from the cortex and the thalamus, both of which synapse onto CINs and modulate their firing rates[34,35,46]. To determine whether these regions project to the VLS, we performed retrograde tracing with cholera toxin. We find that a broad distribution of cells from multiple cortical regions send afferents into the VLS (Fig. 5a). Meanwhile, thalamic inputs into this striatal region originate predominantly from the parafascicular nucleus, consistent with previous observations[47].

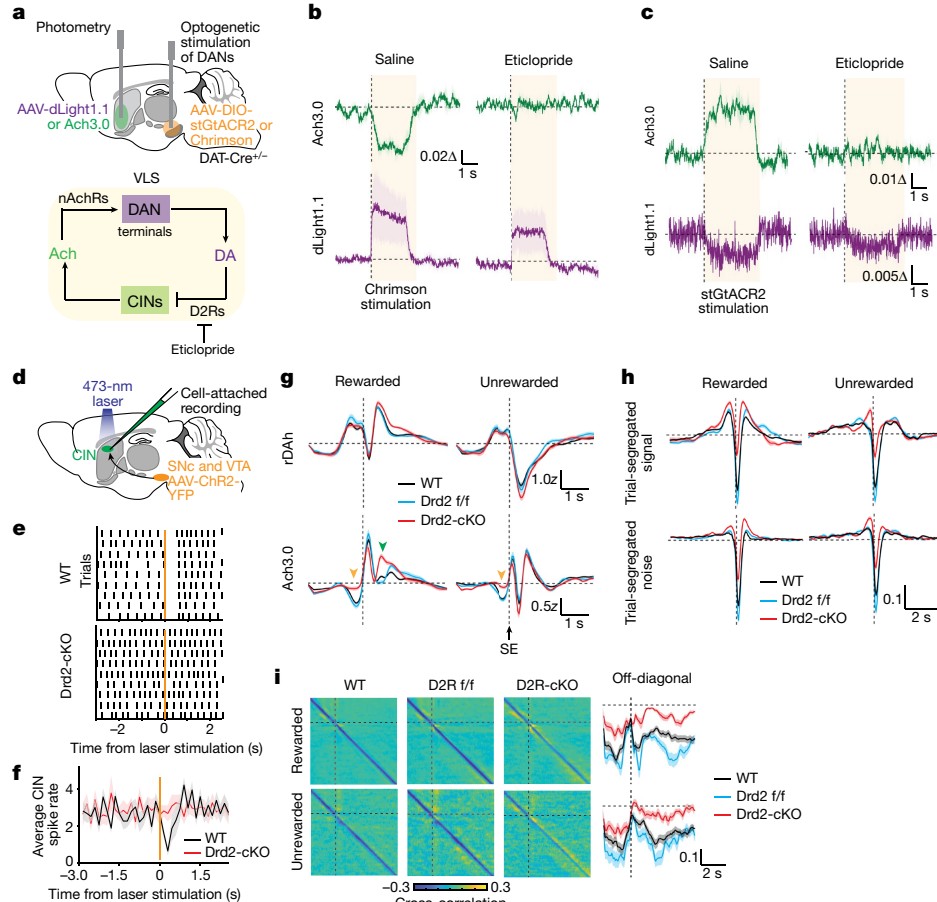

**Fig. 4 | D2Rs are required for DA-mediated inhibition of Ach signals.**
**a**, Injection and fibre-implantation strategy for optogenetic manipulation of DANs and photometric recordings in the VLS (top). Eticlopride was applied during these recordings (bottom). **b**, Ach and DA release during Chrimson-mediated DAN activation with an intraperitoneal injection of saline (left) or eticlopride (right) before the recording. The average $\Delta F/F_0 \pm$ s.e.m. is shown ($n = 3$ mice). Data are aligned to laser stimulation onset. **c**, Ach and DA release during stGtACR2-mediated inhibition of DANs. Data are shown as in **b** ($n = 3$ mice). **d**, Injection and recording set-up to determine the effect of DA release on CIN firing in an acute striatal slice. ChR2-positive DAN afferents are activated with a laser pulse, and CIN firing is recorded with a cell-attached pipette. **e**, Representative single-cell responses to stimulation (orange) of ChR2-positive DAN afferents recorded from CINs with (WT) or without (Drd2-cKO) D2R expression. Each black vertical line denotes a CIN action potential. **f**, Population perievent spike histograms of average spike discharge $\pm$ s.e.m. for cells recorded in **d** (WT: $n = 9$ cells; Drd2-cKO: $n = 9$ cells). **g**, Trial-averaged Ach and DA recordings aligned to side-port entry (SE). Average signals $\pm$ s.e.m. are depicted (WT: $n = 12$ mice; Drd2 f/f: $n = 7$ mice; Drd2-cKO: $n = 8$ mice). The changes in Ach signal that occur when the mice move from centre to side port (orange arrow) and after side-port entry (green arrow) are shown. **h**, Covariance of DA and Ach dynamics in mice shown in **g**, in which DA lags Ach by the indicated time. The average covariance of trial-segregated signals (top) and their noise (bottom) $\pm$ s.e.m. is shown. **i**, Moment-to-moment covariance matrices for mice in **g** of DA and Ach release with their respective off-diagonal signals $\pm$ s.e.m.

To assess the potential influence of these glutamatergic inputs on striatal Ach, we first determined whether striatal glutamate and Ach signals are correlated. Using the glutamate sensor iGluSnFR[48] (Extended Data Fig. 12a,b), we find that striatal glutamate levels vary during the task but are not lateralized (Fig. 5b), allowing us to combine signals from ipsiversive and contraversive trials. Glutamate signals are suppressed before the choice and increase during side-port entry in an analogous manner to Ach (Fig. 5c). In rewarded trials, glutamate exhibits an extra phase of sustained increase during consumption, which is absent in unrewarded trials (Fig. 5c). The activities of cortical and thalamic terminals in the VLS, measured with genetically encoded calcium sensors, coincide with glutamate release across both trial types, suggesting that both inputs can contribute to the release of glutamate in this region (Fig. 5d and Extended Data Fig. 12c–f). Altogether, these data show substantial coincident dynamics from both cortical and thalamic inputs into the striatum, providing a basis for the possibility that these glutamatergic inputs drive changes in Ach levels.

To test whether each input is required for Ach release, we expressed TelC unilaterally in the thalamus or cortex (Fig. 5e,f and Extended Data Figs. 12g–j and 13a–d). Reflecting the importance of cortical and thalamic inputs in regulating reward-based decision-making, both perturbations impaired multiple aspects of the performance of mice in the 2ABT, including impaired switch dynamics after rewarded trials and block transitions, across multiple choice-outcome histories (Extended Data Fig. 13f–m). Consistent with this impairment, the RFLR model description revealed a reduction in $\beta$ and an increase in $\tau$, reflecting a weakened incorporation of the weight given to previous evidence and a faster rate of information decay, respectively (Extended Data Fig. 13i,m).

Loss of neurotransmission from each region robustly dampened Ach transients across all trials, as seen in the lowered $\Delta F/F_0$ of the Ach sensor signal (Fig. 5e,f, bottom). The degree of suppression of $\Delta F/F_0$ was strong and consistent across mice and sufficient to overcome any underlying variability in signals (Extended Data Fig. 13e), unlike the effects of CIN perturbation of DA levels. Analysis of the remaining Ach

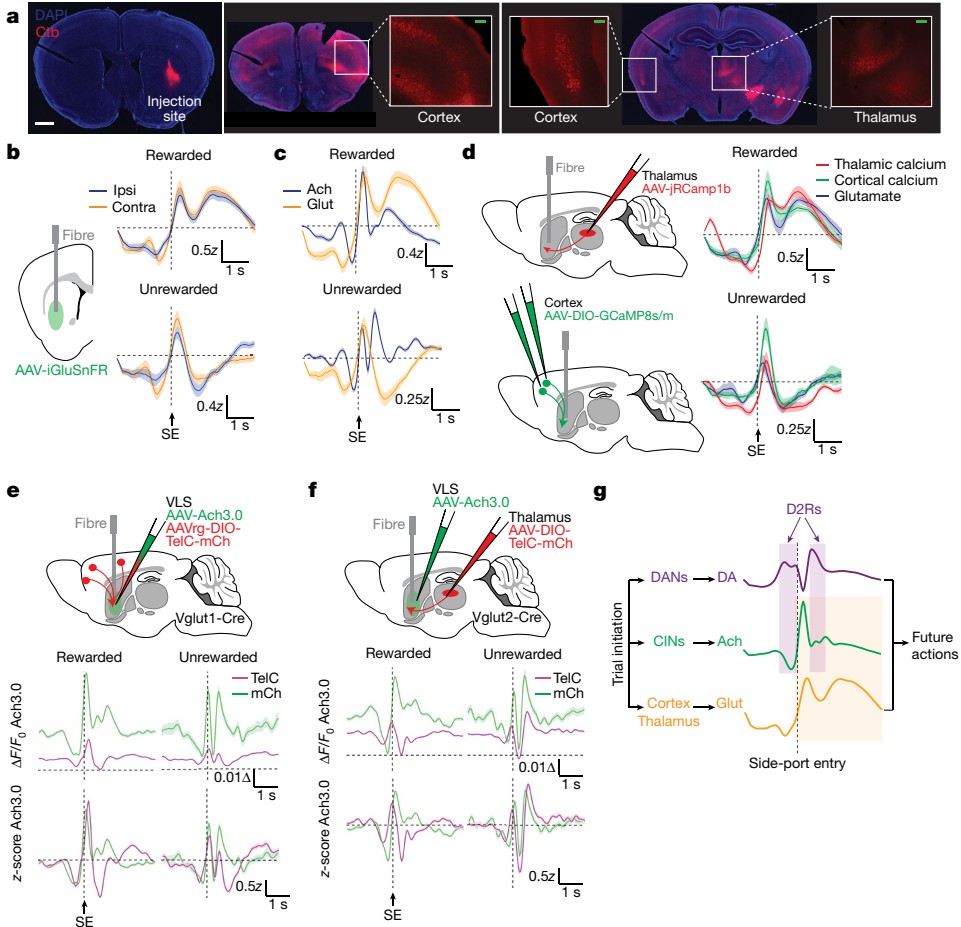

**Fig. 5 | Corticostriatal and thalamostriatal inputs are necessary to drive Ach signals in the VLS. a**, Retrograde labelling of cortex and thalamus with cholera toxin (Ctb) injected into the VLS. Images of the injection site (left) and representative coronal sections that depict Ctb-positive cortical and thalamic cell bodies (right) are shown (*n* = 3 mice). Scale bars: white, 1 mm; green, 200 µm. **b**, Glutamate release from 2ABT trials measured with iGluSnFR. Average signals ± s.e.m. are shown (*n* = 4 mice). Contra, contraversive; ipsi, ipsiversive. **c**, Overlay of glutamate (glut) and Ach release recorded from separate mice during the indicated trial types. Data are presented as in **b** (glutamate: *n* = 4 mice; Ach: *n* = 9 mice). **d**, Injection and fibre-implantation strategy for photometry of calcium dynamics in thalamic and cortical terminals in the VLS. Data are shown as in **b** (glutamate: *n* = 4 mice; thalamic calcium: *n* = 6 mice; cortical calcium: *n* = 5 mice). **e**, Injection and fibre-implantation strategy

for TelC or mCh expression in the cortex with Ach recordings in the VLS (top). $\Delta F/F_0$ (middle) and *z*-scored signals (bottom) of the average Ach release from the indicated treatment groups. Data are depicted as in (**b**) (mCh: n = 4 mice; TelC: n = 5 mice). **f**, Injection and fibre-implantation strategy for TelC or mCh expression in the thalamus with Ach recordings in the VLS (top). $\Delta F/F_0$ (middle) and *z*-scored signals (bottom) of the mean Ach release from the indicated treatment groups. Data are shown as in **b** (mCh: *n* = 3 mice; TelC: *n* = 5 mice). **g**, Schematic summarizing findings. Trial initiation evokes the release of multiple neurotransmitters in the VLS, all of which interact and influence decision-making. Glutamate release from cortex and thalamus are necessary to promote Ach release (orange box), while DA release inhibits it at specific trial moments through D2Rs (purple boxes). Altogether, this guides future actions.

transients reveals unique ways that cortical and thalamic inputs modulate Ach levels. In unrewarded trials, loss of cortical but not thalamic inputs perturbs Ach transients after side-port entry, suggesting that the cortex has a specific role in driving this signal. In addition, loss of the thalamic input shifts the timing of Ach transients more than does loss of the cortical input, which might reflect the greater degree of behavioural disruption in mice with thalamic TelC injections. Overall, our results reveal that both the cortex and the thalamus are required to sustain the levels of Ach during decision-making, and that each input can uniquely alter the dynamics of striatal Ach during a trial.

## Discussion

DA and Ach are crucial neuromodulators that directly affect each other's release in vitro in the striatum. However, whether these interactions regulate neuromodulator levels in vivo, particularly during decision-making, is largely unknown. To address this, we evaluated how striatal DA and Ach dynamics are regulated by the proposed

bidirectional circuit during a task that requires mice to make choices flexibly within a changing environment. We revealed that DA and Ach signals are generally anticorrelated across time, but that this relationship is dynamic and modulated by action-outcome history. Although striatal Ach release does not modulate DA dynamics during the 2ABT, DA exerts a key influence on Ach signals through D2Rs. Without this interaction, the ability of action and reward history to influence decision-making is diminished. As well as the inhibition of Ach release by DA, cortical and thalamic inputs concurrently drive the release of Ach and contribute to both basal Ach levels and reward-outcome-dependent transients. In conclusion, by using a diverse toolset to interrogate and alter neuromodulator levels during a complex behavioural task, we establish a precise in vivo role for a long-defined in vitro circuit and reveal new modes of CIN regulation by dopaminergic and glutamatergic inputs (Fig. 5g). Moreover, our findings provide a framework for further studies, with which we can gain a deeper understanding of the neurochemical basis of decision-making and behaviour (Supplementary Information).

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

## Methods

### Mice

The following mouse lines were used: C57BL6/J (The Jackson Laboratory, 000664); *ChAT-IRES-Cre* (The Jackson Laboratory, 006410); *DAT-IRES-Cre* (The Jackson Laboratory, 006660), *Drd2*[loxP] (The Jackson Laboratory, 020631); *Vglut2-IRES-Cre* (The Jackson Laboratory, 028863); and *Vglut1-IRES-Cre* (The Jackson Laboratory, 023527). All mice were bred on a C57BL/6J genetic background and heterozygotes were used unless noted. For behaviour experiments, 6–8-week-old male mice were used. For all experiments, a sample size of at least 3 was chosen in a manner that was not guided by a statistical test. No randomization or blinding was performed. All animal care and experimental manipulations were performed in accordance with protocols approved by the Harvard Standing Committee on Animal Care, following guidelines described in the US NIH Guide for the Care and Use of Laboratory Animals.

### Intracranial injections

Mice were anaesthetized with 5% isoflurane and maintained under surgery with 1.5% isoflurane and 0.08% $O_2$. Under the stereotaxic frame (David Kopf Instruments), the skull was exposed in aseptic conditions, a small craniotomy (around 300 µm) was drilled and the virus (Supplementary Information) was injected into the following regions with the associated coordinates listed from bregma: VLS (coordinates: 0.6 mm A/P, ±2.3 mm M/L and 3.2 mm D/V); SNc and VTA (coordinates: −3.35 mm A/P, ±1.75 mm M/L and 4.3 mm D/V); thalamus (coordinates: −2.1 mm A/P, ±1.0 mm M/L and 3.5 mm D/V); prefrontal cortex (PFC; coordinates: 2.0 mm A/P, ±0.4 mm M/L and 2.3 mm D/V).

Injections were performed as previously described[49]. A pulled glass pipette was held in the brain for 3 min, and viruses were infused at a rate of 50 nl min$^{-1}$ (VLS), 30-40 nl min$^{-1}$ (PFC) and 70 nl min$^{-1}$ (SNc and VTA) with a syringe pump (Harvard Apparatus, 883015). Pipettes were slowly withdrawn (less than 10 µm s$^{-1}$) at least 6 min after the end of the infusion, and 350 nl was infused per injection site except for Ctb 555 injections (50 nl at 4 µg µl$^{-1}$).

For AAV injections, the wound was sutured. For fibre implants, after AAV injection, the skull was scored lightly with a razor blade to promote glue adhesion. Then, a 200-µm blunt-ended fibre (MFC_200/230-0.48_4 mm, Doric Lenses) was slowly inserted into the brain until it reached 100 µm above the injection site. The fibre was held in place with glue (Loctite gel, 454) and hardening was accelerated with the application of Zip Kicker (Pacer Technology). A metal headplate was glued at lambda and white cement (Parkell) was applied on top of the glue to further secure the headplate and fibres. Fibre implants were protected with a removable plastic cap (Doric Lenses) until recordings.

After the surgery, mice were placed in a cage with a heating pad until their activity was recovered, before returning to their home cage. Mice were given pre- and post-operative oral carprofen (CPF, 5 mg per kg per day) as an analgesic and monitored daily for at least four days after surgery. At least four weeks passed after virus injection before experiments were performed, except for retrograde tracer injections, in which one week passed. Of note, to detect thalamic activity, we injected jRCaMP1b in the somas and recorded from thalamic terminals in the VLS (Fig. 5d and Extended Data Fig. 12c,d). Meanwhile, a multisite injection strategy was required for cortical inputs given their widespread distribution (Fig. 5d, bottom left and Extended Data Fig. 12e,f). In addition, we found that a brighter calcium sensor, GCaMP8, was necessary for the detection of cortical signals arising from these dispersed sources. For cortical inputs, we used a retrograde AAV approach in *Vglut1-IRES-Cre* mice to restrict expression of the toxin to cells that project into the VLS (Fig. 5e and Extended Data Fig. 13a–d). However, for thalamic inputs, we could not use a retrograde approach in *Vglut2-IRES-Cre* mice owing to the expression of *Vglut2* (also known as *Slc17a6*) in the cortex (Allen Institute); therefore, we instead injected

TelC directly into the thalamus of *Vglut2-IRES-Cre* mice (Fig. 5f and Extended Data Fig. 12g–j).

### Immunohistochemistry

Mice were anaesthetized by isoflurane inhalation and transcardially perfused with phosphate-buffered saline (PBS) followed by 4% paraformaldehyde (PFA) in PBS. Brains were extracted and stored in 4% PFA PBS for at least 8 h or in 4% PFA, 0.02% sodium azide and PBS for long-term storage at 4 °C. The right hemisphere of the brain was slightly slit with a razor to enable accurate identification of the hemispheres once the brains were sliced. Brains were sliced into 70-µm-thick free-floating sections with a Leica VT1000 S vibratome. Selected slices were transferred to a six-well plate and rinsed three times for 5 min each in PBS. They were then blocked with rotation at room temperature for an hour in blocking buffer (5% normal goat serum (Abcam), 0.2% Triton X-100 PBS). The blocking buffer was removed and replaced with 500–700 µl of a solution containing the indicated primary antibody (Supplementary Information). Slices were incubated overnight with side-to-side rotation at 4 °C. The next day, slices were transferred to a clean well and washed five times for 5 min each in PBST (PBS with 0.2% Triton X-100). After the final wash, slices were incubated for 1.5 h in 500–700 µl of the indicated secondary antibody diluted 1:500 in blocking buffer. Slices were washed four times in PBST for 5 min each, then four times in PBS for 5 min each before mounting with ProLong Diamond Antifade Mountant with DAPI (Thermo Fisher Scientific). Slices were imaged with an Olympus VS120 slide scanning microscope or a spinning disk confocal microscope.

### Behaviour apparatus, training and task

The apparatus used for the behaviour is as described previously[40] with the following modifications. Clear acrylic barriers 5.5 cm in length were installed in between the centre and side ports before training, to extend the trial time and to help produce better-resolved photometry recordings. Water was delivered in 3-µl increments. Hardware and software to control the behaviour box are available online: https://github.com/HMS-RIC/TwoArmedBandit.

Singly housed male mice were restricted to 1 ml water per day before training and were maintained at at least 80% of their initial body weight for the full duration of training and photometry. All training sessions were conducted in the dark under red light conditions. A blue LED above the centre port signals to the mouse to initiate a trial by poking in the centre port. Blue LEDs above the side ports are then activated, signalling the mouse to poke in the left or right port within 5 s. At any given instance, only one side port rewards water. Reward probabilities are defined by custom software (MATLAB). Withdrawal from the side port ends the trial and begins a 1-s ITI, after which the mouse can self-initiate the next trial. An expert mouse can perform 200–300 trials in a session.

To train the mice to proficiency, they were subjected to incremental training stages. Each training session lasted for 30–60 min, adjusted according to the mouse's performance. Mice progressed to the next stage once they were able to complete at least 100 successful trials with a reward rate of at least 75%. On the first day, they were habituated to the behaviour box, with water being delivered from both side ports and triggered only by a side-port poke. In the next stage, mice learned the trial structure—only a poke in the centre port followed by a poke in the side port delivers water. Then, the mice transitioned to learning the block structure, in which 30 successful trials on one side port triggers a deterministically rewarded port ($P_{high}$ = 100%) to switch to the other side port. Finally, mice performed trials in the presence of barriers in between the centre and the side ports. A series of transparent barriers of increasing size (extra-small (1.5 cm), small (3 cm), medium (4 cm) and long (5.5 cm)) aided in learning. Finally, the mice were trained on probabilistic reward delivery ($P_{high}$ = 95%). Once the mice were proficient, optical fibres were implanted into their brains.

After fibre-implant surgeries, mice were retrained to achieve the same pre-surgery performance level. Habituation to head fixation on a wheel followed by habituation to attachment of a mock photometry patchcord was performed over successive days for each mouse. Head fixation was done to temporarily restrain the mice to make it easier to attach and secure the patchcord for stable photometry recordings. Recordings were performed four weeks after surgery to allow for stable viral expression levels as well as a consistent and proficient level of task performance from the mice. In experiments in which the LED cue is omitted ('cue-omission' trials), we turned off the LEDs located above the centre and side ports but left all other task parameters and recording conditions unchanged.

### Photometry and behaviour recordings
Fibre implants on the mice were connected to a 0.48 NA patchcord (Doric Lenses, MFP_200/220/900-0.48_2m_FCM-MF1.25, low autofluorescence epoxy), which received excitation light and propagated its emission light to a Dorics filter cube (blue excitation light (465–480 nm); red excitation light (555–570 nm); green emission light (500–540 nm); red emission light (580–680 nm) (FMC5_E1(465-480)_F1(500-540) _E2(555-570)_F2(580-680)_S, Doric Lenses)). Excitation light originated from LED drivers (Thorlabs) and was amplitude-modulated at 167 Hz (470-nm excitation light, M470F3, Thorlabs; LED driver LEDD1B, Thorlabs) and 223 Hz (565-nm excitation light, M565F3, Thorlabs; LED driver LEDD1B, Thorlabs) using MATLAB. The following excitation light powers were used for the indicated sensors: dLight1.1 (25 μW); Ach3.0 (25 μW); rDAh (45 μW); and iGluSNFr (15 μW). Signals from the photodetectors were amplified in DC mode with Newport photodetectors or Dorics amplifiers and received by a Labjack (T7) streaming at 2,000 Hz. The Labjack also received synchronous information about behaviour events logged from the Arduino, which controls the behaviour box. The following events were recorded: centre-port entry and exit, side-port entry and exit, lick onset and offset, and LED-light onset and offset. Photometric recordings and behaviour performance were analysed as described (Supplementary Information).

### Optogenetic manipulations
All optogenetic stimulations were triggered by side-port entry and persisted for a set time duration that was adjusted for the average side-port occupancy of the mice in each experimental cohort. We used a stimulation duration that would not persist past the side-port entry and introduce ectopic effects on the next trial. For optogenetic stimulations with Chrimson during behaviour (Extended Data Fig. 7b), 15 mW of a 590-nm laser (Optoengine) was evoked in 25% of trials for 1.5 s interleaved throughout the session. The excitation light was delivered via the Doric filter cube, which led to a laser stimulation artefact, which is removed in the recordings. Only one hemisphere was illuminated in each session. For optogenetic manipulation of DANs (Figs. 3g and 4b,c), 15 mW of a 590-nm laser was used for Chrimson whereas 0.7 mW of a 463-nm laser was used for stGtACR2 stimulations, each for a duration of 5 s. For optogenetic stimulations of head-fixed mice on a wheel, in each session, the laser excitation duration was 1.5 s, with a 45-s ITI, repeated 20 times. The signals displayed are averages of each session (Fig. 4b,c and Extended Data Fig. 7a,e). The photometry signal baseline was calculated by averaging the signal 1.5 s before laser stimulation across the 20 sweeps.

### GLM
Photometry recordings and behavioural data used for GLMs were collected from the indicated mice, with 3–6 sessions per mouse and approximately 150–300 trials per session, of which typically more than 75% are rewarded. These data were aligned to behavioural events (see 'Signal demodulation' in Supplementary Information) to create a predictive matrix $X$ (of dimensions $N \times F$) and a response vector, $\mathbf{y}$ (of dimension $N$), where $N$ is the number of time steps recorded in the session and $F$ is the number of predictors in the analysis. Except for instances in which photometry variables were used as predictors, the GLM features consisted of values 0 and 1 to indicate if a behavioural event (for example, a lick) occurred in the time bin.

For each predictive matrix, a design matrix $\varphi(X)$ (of dimensions $N \times F (2T + 1)$) was constructed from $T$ time shifts forward and backward ($T = 20$, 54 ms each), resulting in GLM coefficients that corresponded to time-based kernels for each of the predictive features in $X$. Data from the ITI period, in which there are no task-relevant behavioural events, were excluded, and only data spanning shortly before centre entry and after side-port exit were modelled. When initial and final time shifts spanned the boundary between two trials, the overlapped data were included twice–once in each of the trials on either side of the boundary) to ensure sufficient representation of each event in training, validation and test datasets. Because of the variability in the ITIs, this duplication resulted in around 1.5% to around 17.3% of the data points being present in both the training and the test datasets.

To evaluate the performance of the GLMs, trials were partitioned into training and test datasets, each containing 50% of the data. For the results shown in Extended Data Fig. 5e,i, multiple model runs were carried out, with the number of repetitions designated $Y$ in this paragraph. For each run, the data were split into training and test datasets and were held constant for all the models tested in that run. $Y = 10$ for the leave-out analysis (Extended Data Fig. 5f) and $Y = 3$ for the hyperparameter analysis (Extended Data Fig. 5e,i). For each model run, a 10-fold group shuffle split (GSS) by trial was applied to the training set to obtain cross-validated ranges for the MSEs, based on an 80–20 training–validation split within each of the 10 GSS folds. Each validation MSE value in the box plots is the average of the concatenated squared residuals across all validation data points in these 10 GSS folds. Finally, the model was refit to and evaluated on the entire training dataset, and this refit model was evaluated on the test dataset, resulting in the training and test MSEs and $R^2$ values for each model run. The $R^2$ values presented in the text are the average values calculated from the test sets averaged across $Y$ model runs. Typically, these values had small variance, with ranges from maximum to minimum of less than 1.2%. Therefore, the ranges are not stated in the text.

For each of the models used, the algorithms minimize an associated cost function with respect to the fitted coefficients. The cost functions are as follows, where $J$ is the cost function to be minimized, $X$ is the design matrix (set of time-shifted tasks or behavioural events), $\mathbf{y}$ is the response vector (fluorescence indicator), $\beta$ is the set of fitted coefficients, $\|a\|_2^2$ is the sum of the squared entries in vector $\mathbf{a}$, $\|a\|_1$ is the sum of the absolute values of the entries in vector $\mathbf{a}$, $\alpha$ is the regularization parameter and $\lambda$ is the L1 ratio.

Ordinary least squares (OLS):

$$J(X, y) = \| y - X\beta \|_2^2$$

Ridge regression (L2):

$$J(X, y) = \| y - X\beta \|_2^2 + \alpha \, \| \beta \|_2^2$$

Elastic net and lasso regression (L1):

$$J(X, y) = \frac{1}{2N} \, \| y - X\beta \|_2^2 + \alpha\left(\lambda \, \|\beta\|_1 + \frac{1}{2}(1 - \lambda) \|\beta\|_2^2\right)$$

Note that for OLS, $\alpha = 0$ as there is no regularization. Furthermore, setting $\lambda = 1$ yields lasso regression (L1 regularization). However, setting $\lambda = 0$ does not give an equation equivalent to the version of ridge regression provided above, resulting in two different $\alpha$ scales (Extended Data Fig. 5e,i). In addition, for L2 regularization, the validation-based models were fit to 80% of the total of samples available to the final model; thus, the validation models performed worse than their training

or test counterparts because they are, in effect, facing an increased amount of regularization.

The sources for the least squares regression models are listed below:

OLS: https://scikit-learn.org/stable/modules/generated/sklearn.linear_model.LinearRegression.html.

L2: https://scikit-learn.org/stable/modules/generated/sklearn.linear_model.Ridge.html.

L1 and elastic net: https://scikit-learn.org/stable/modules/generated/sklearn.linear_model.ElasticNet.html.

All kernels ($\beta$ coefficients) depicted are the mean coefficients across the $Y$ model runs with one standard deviation above and below the mean represented in the shaded regions. All GLM reconstructions depict the average signal with an overlay of the bootstrapped 95% confidence intervals as the upper and lower bounds (shaded region).

## Preparation of acute brain slices

Brain slices were obtained from two- to four-month-old mice (both male and female) using standard techniques. Mice were anaesthetized by isoflurane inhalation and subjected to cardiac perfusion with ice-cold artificial cerebrospinal fluid (ACSF) containing 125 mM NaCl, 2.5 mM KCl, 25 mM $NaHCO_3$, 2 mM $CaCl_2$, 1 mM $MgCl_2$, 1.25 mM $NaH_2PO_4$ and 25 mM glucose (295 mOsm $kg^{-1}$). Brains were blocked and transferred into a slicing chamber containing ice-cold ACSF. Sagittal slices of striatum for amperometric or cell-attached recordings were cut at 300 μm thickness with a Leica VT1000 S vibratome in ice-cold ACSF, transferred for 10 min to a holding chamber containing choline-based solution (consisting of 110 mM choline chloride, 25 mM $NaHCO_3$, 2.5 mM KCl, 7 mM $MgCl_2$, 0.5 mM $CaCl_2$, 1.25 mM $NaH_2PO_4$, 25 mM glucose, 11.6 mM ascorbic acid and 3.1 mM pyruvic acid) at 34 °C, then transferred to a secondary holding chamber containing ACSF at 34 °C for 10 min and subsequently maintained at room temperature (20–22 °C) until use. All recordings were obtained within 4 h of slicing. Both choline solution and ACSF were constantly bubbled with 95% $O_2$/5% $CO_2$.

## Cell-attached recordings

Acute sagittal brain slices and electrophysiological recordings were obtained from the dorsal striatum as described before[50], with the following variations: CINs were identified using morphological and electrophysiological features[14]. Slices were sustained in ACSF with 10 μM of gabazine, CPP and NBQX (Tocris). For cell-attached recordings, bath temperatures for the acute slice recordings were maintained at 34 °C, pipettes were filled with ACSF, had 1–2 MΩ resistance, seal resistances were from 10 to 100 MΩ. Action potential firing was monitored in the cell-attached recording configuration in the voltage-clamp mode ($V_{hold}$ = 0 mV). ChR2 was activated by a single 2-ms pulse of 473-nm light delivered at 5.74 mW using full-field illumination through the objective at 120-s intervals.

## Amperometry recordings

Slices were stimulated with 593-nm light, delivered at 5.86 mW for 2 ms using full-field illumination through the objective at 180-s intervals. Constant-potential amperometry was performed as previously described[50]. In brief, glass-encased carbon-fibre microelectrodes (CFE1011 from Kation Scientific: 7 μm diameter, 100 μm length) were placed approximately 50–100 μm within dorsal striatum slices and held at a constant voltage of +600 mV for 9 s versus Ag/AgCl by a Multiclamp 700B amplifier (Molecular Devices). Electrodes were calibrated with fresh 5 μM dopamine standards in ACSF to determine the sensitivity of the carbon-fibre microelectrodes and to allow conversion of current amplitude to extracellular dopamine concentration.

## Reporting summary

Further information on research design is available in the Nature Portfolio Reporting Summary linked to this article.

## Data availability

The data and code that support the findings of this study are available upon request from the corresponding author. Source data are provided with this paper.

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

**Acknowledgements** We thank all members of the B.L.S. laboratory for experimental suggestions and advice, in particular M. L. Wallace and S. J. Lee for behaviour and photometry advice and P. Capelli for assistance with confocal imaging. We also thank A. E. Girasole, P. Capelli and C. Smillie for feedback on the manuscript; G. Radeljic for technical assistance; Y. Li for sharing the rDAh sensor; and O. Mazor and P. Gorelick for their help with implementing the hardware and software for the 2ABT. This work was supported by grants to B.L.S. (R37-NS046579, U19 NS113201 and Simons Center for the Global Brain), to L.C. (Hanna Gray Fellowship from the Howard Hughes Medical Institute) and to C.C.B. (NSF Graduate Research Fellowship Program).

**Author contributions** L.C. and B.L.S. conceptualized the study. L.C. and M.J.W. performed experiments except the electrophysiological ones, which were done by W.W. B.L.S., C.C.B., J.A.Z. and L.C. performed analyses. L.C. and B.L.S. wrote and edited the manuscript, with feedback from C.C.B. and J.A.Z.

**Competing interests** The authors declare no competing interests.

**Additional information**
**Correspondence and requests for materials** should be addressed to Bernardo L. Sabatini.

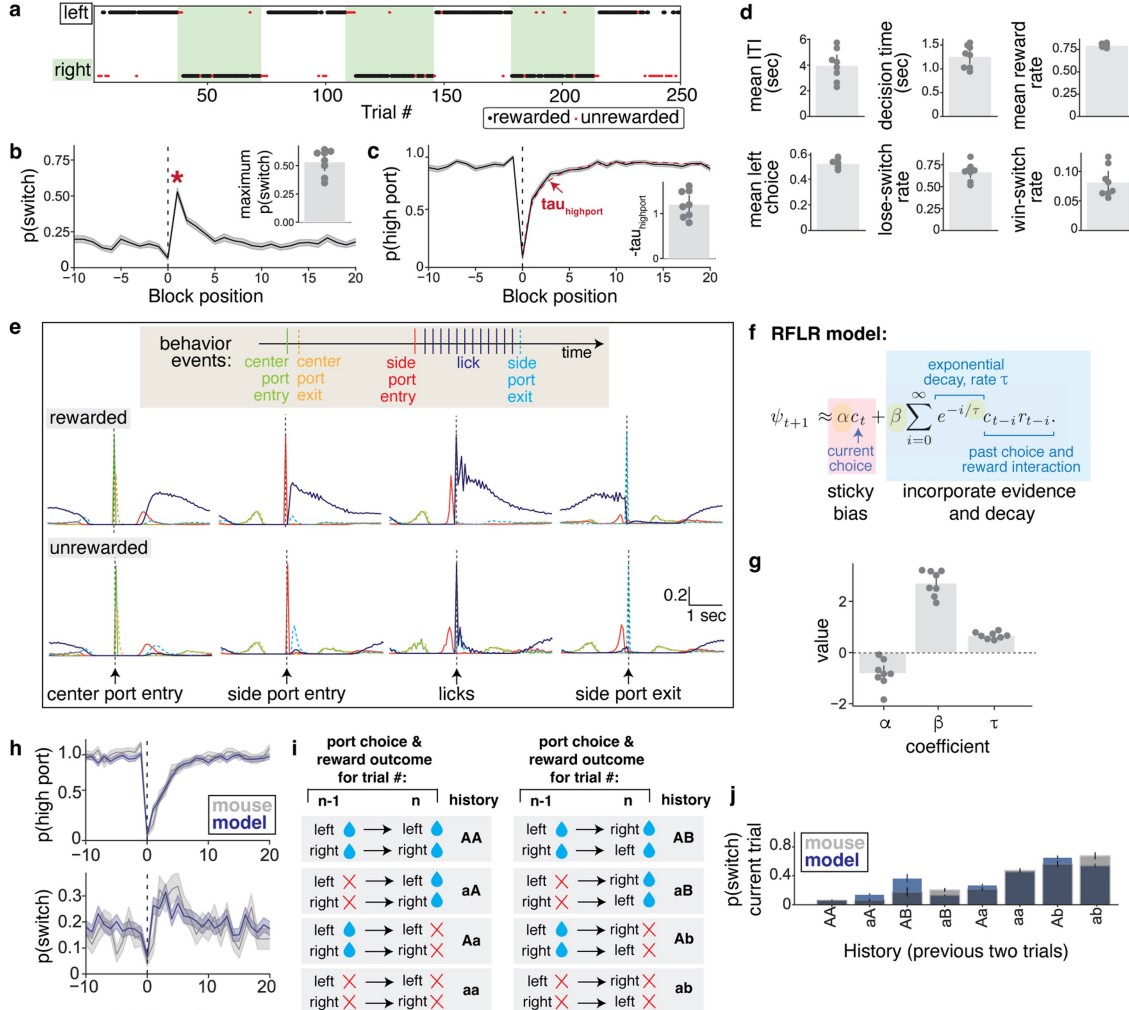

**Extended Data Fig. 1 | Mouse performance in the 2ABT. a**, Behaviour of a representative mouse in a 2ABT session. White and green denote the left and right port, respectively, as being the higher rewarded port. Dots represent the mouse's choice and the reward outcome. **b**, Probability of switching ports (*P*(switch)) shown as mean ± s.e.m., and maximum *P*(switch) (asterisk) as a function of trial number from the block transition at zero. Data are shown as mean ± 95% C.I., and each dot represents a unique mouse (*n* = 8 mice). **c**, Probability of occupancy at the highly rewarded port (p(high port)) and tau_highport (dashed red line) are shown. Data are depicted as in **b**. **d**, 2ABT performance metrics, shown as mean ± 95% C.I., with each dot representing a unique mouse (*n* = 7 mice). **e**, Timing of behavioural events. The probability of each event occurrence is plotted with respect to time and aligned to the indicated event (mean ± s.e.m., *n* = 7 mice). **f**, The RFLR model, which calculates log odds

of the mouse's next choice ($\psi_{t+1}$) given its most recent choice ($c_t$) and a series of prior choices and rewards. $c_t$ represents choice, $r_t$ represents the reward outcome on trial $t$, relative to the current trial $i = 0$. $\alpha$ is the weight on the most recent choice, $\beta$ is the weight on choice and reward outcome, which decays exponentially across trials at a rate of $\tau$. **g**, Summary of the RFLR model coefficients, shown as mean ± 95% C.I. with each dot representing a unique mouse (*n* = 8 mice). **h**, RFLR predicted probability (blue) versus the mouse behaviour (grey) of *P*(high port) and *P*(switch). The mean ± s.e.m. across trials is shown. **i**, Annotation of action-outcome sequences in **j**. **j**, Conditional switch probabilities for the current trial, given the action-outcome trial sequence of the past two trials (history). The original data (grey) are overlaid with data predicted by the RFLR model (blue). The bars show the mean with the binomial standard error (*n* = 8 mice).

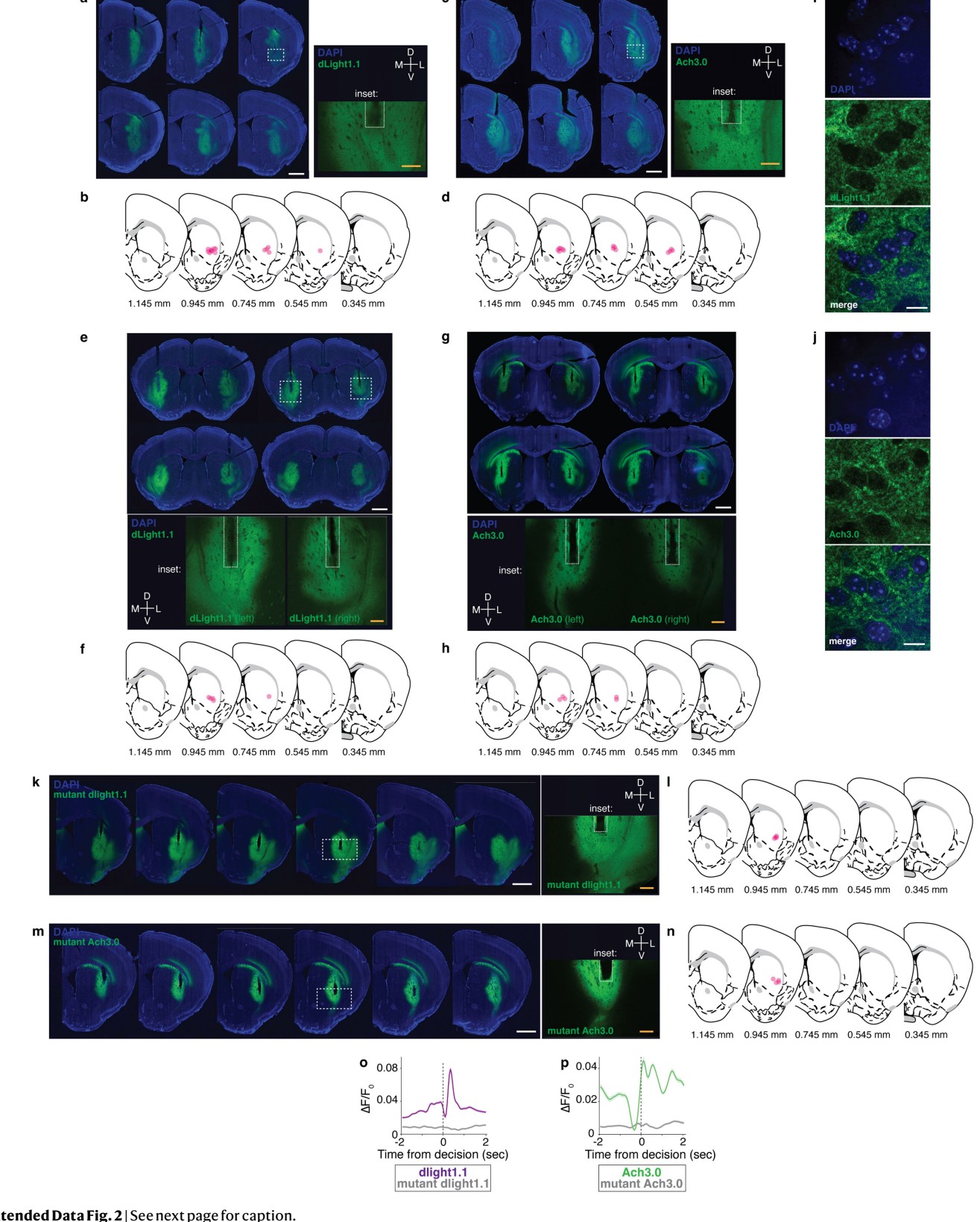

**Extended Data Fig. 2** | See next page for caption.

**Extended Data Fig. 2 | Histology for DA and Ach recordings in the VLS with wild-type and mutant sensors. a**, Images of dLight1.1 expression for a representative mouse recorded for Fig. 1d and Extended Data Fig. 3d. Coronal sections show spread of expression. A higher-resolution image of the recording site (dashed box) is shown with the fibre tract denoted (inset). Scale bars (white): 1 mm; (orange): 0.2 mm. **b**, Location of the optical fibre tip (pink dot) for mice recorded in Fig. 1d and Extended Data Fig. 3d. **c**, Images of Ach3.0 expression for a representative mouse recorded for Fig. 1d and Extended Data Fig. 3d. Images depicted as in **a. d**, Location of the optical fibre tip (pink dot) for mice recorded in Fig. 1d and Extended Data Fig. 3d. **e**, Images of dLight1.1 expression for a representative mouse recorded for Extended Data Fig. 3e. Images depicted as in (**a**). **f**, Location of the optical fibre tip (pink dot) for mice recorded in Extended Data Fig. 3e. **g**, Images of Ach3.0 expression for a representative mouse recorded for Extended Data Fig. 3f. Images are depicted as in **a. h**, Schematic depicting the location of the optical fibre tip (pink dot) for mice in Extended Data Fig. 3f. **i**, Confocal image of dLight1.1 expression in the VLS for a representative mouse recorded for Fig. 1d and Extended Data Fig. 3d. DAPI serves as a nuclear marker. Scale bar = 10 μm. **j**, Confocal image of Ach3.0 expression in the VLS for a representative mouse recorded for Fig. 1d and Extended Data Fig. 3d. Images are depicted as in **i. k**, Images of mutant dLight1.1 expression for a representative mouse recorded for **o**. Images depicted as in **a. l**, Location of the optical fibre tip (pink dot) for mice recorded in **o. m**, Images of mutant Ach3.0 expression for a representative mouse recorded for **p**. Images are depicted as in **a. n**, Location of the optical fibre tip (pink dot) for mice recorded in **p. o**, Average $\Delta F/F_0$ of DA release during rewarded trials from dLight1.1 or its binding mutant ($n$ = 4 mice). Mean signals ± s.e.m. are shown. **p**, Average $\Delta F/F_0$ of Ach release during rewarded trials from Ach3.0 or its mutant version ($n$ = 4 mice). Signals are depicted as in **o**.

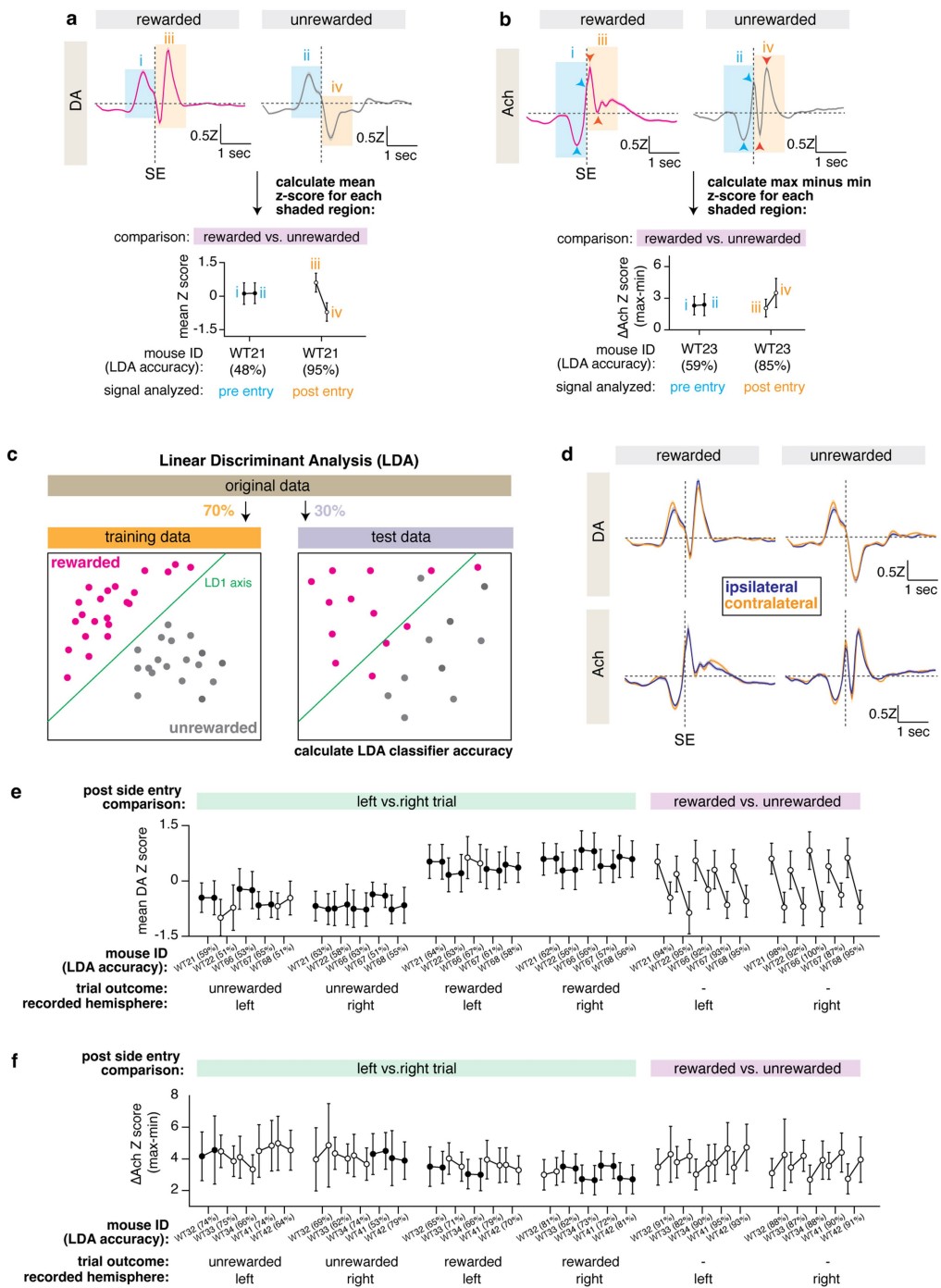

**Extended Data Fig. 3 | Analysis of fibre photometric recordings and their lateralization. a**, Schematic of the workflow for statistical analysis of DA signals. For each mouse, the mean *z*-scored DA signal ± s.e.m. (mock traces shown) is calculated for the pre-side entry period (blue box) and the post-side entry period (orange box). This is plotted as a paired comparison (mean ± s.d.) between the designated trial types for each mouse (that is, rewarded (left dot in connected pair) vs unrewarded (right dot in connected pair)), with open circles representing a significant difference and black circles representing an insignificant difference as assessed by a two-sided *t*-test (*P* < 0.05). In parentheses next to each mouse identifier is the % classification accuracy of a cross-validated LDA on held-out data. **b**, Schematic of the workflow for statistical analysis of Ach signals. For each mouse, the difference between the maximum and minimum of the *z*-scored Ach signal (blue and orange arrows) across the designated trial type is calculated for the denoted time period. Data are plotted as in **a**. **c**, LDA of photometry signals. The dataset is parsed into a training set and a test set. The axis that maximizes the distance between the means and minimizes the variance of the two training datasets is calculated (LD1 axis), which can classify the data into the indicated trial outcome. To assess LDA performance, the degree to which this classifier accurately classifies the test dataset is determined. **d**, DA and Ach release in ipsilateral and contralateral trials. The averaged *z*-scored sensor signal ± s.e.m. is depicted (DA: *n* = 7 mice; Ach: *n* = 9 mice). **e**, Statistical analysis of the lateralization of DA signals post-side entry. Data are plotted as in **a**. **f**, Statistical analysis of the lateralization of Ach signals post-side entry. Data are plotted as in **b**.

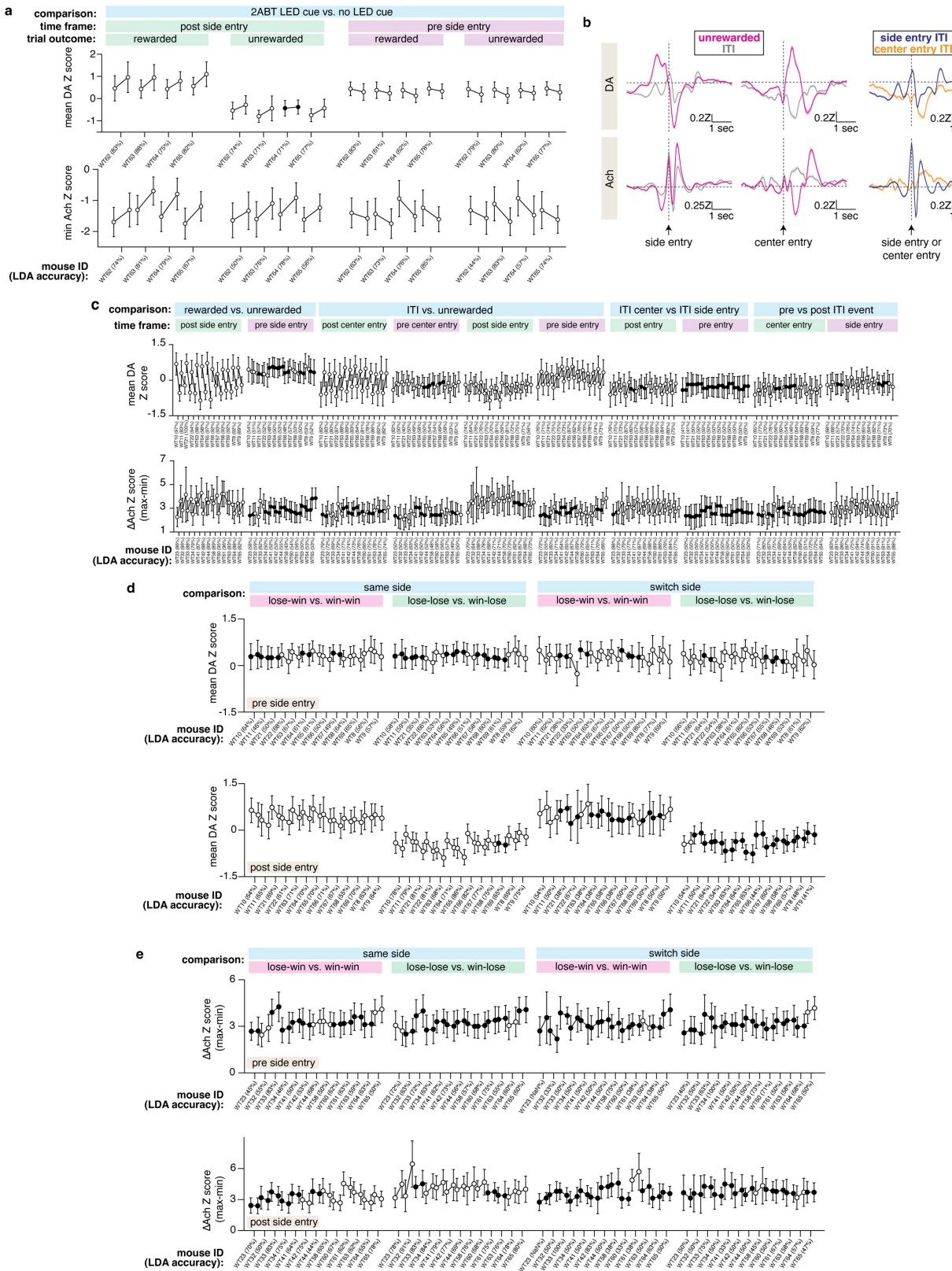

**Extended Data Fig. 4 |** See next page for caption.

**Extended Data Fig. 4 | Analysis of DA and Ach signals from different cue states, from the ITI and from different action-outcome histories.** **a**, Analysis of DA and Ach signals in Fig. 1f. For each mouse, paired comparisons (connected dots) are shown between trials with an LED cue (left dot in connected pair) versus without an LED cue (right dot in connected pair). The mean DA and the mean minimum Ach $z$-scored signal $\pm$ s.d. are shown. Data are depicted as in Extended Data Fig. 3b. Open circles represent a significant difference for each paired comparison (two-sided $t$-test ($P < 0.05$)). **b**, DA and Ach release during unrewarded trials and the ITI aligned to the indicated events.

The averaged $z$-scored sensor signal $\pm$ s.e.m. is depicted (DA: $n = 13$ mice; Ach: $n = 14$ mice). **c**, Analysis of photometry signals in **b**. Data are depicted as in **a**. **d**, Analysis of DA signals in Fig. 1g,h. The left dot and right dot in the connected pair are the first and second condition listed respectively (that is, for the comparison listed as 'lose-win vs. win-win', the left dot in the connected pair represents the lose-win condition and the right dot in the connected pair represents the win-win condition). Data are depicted as in **a**. **e**, Analysis of Ach signals in Fig. 1g,h. Data are plotted as in **a**.

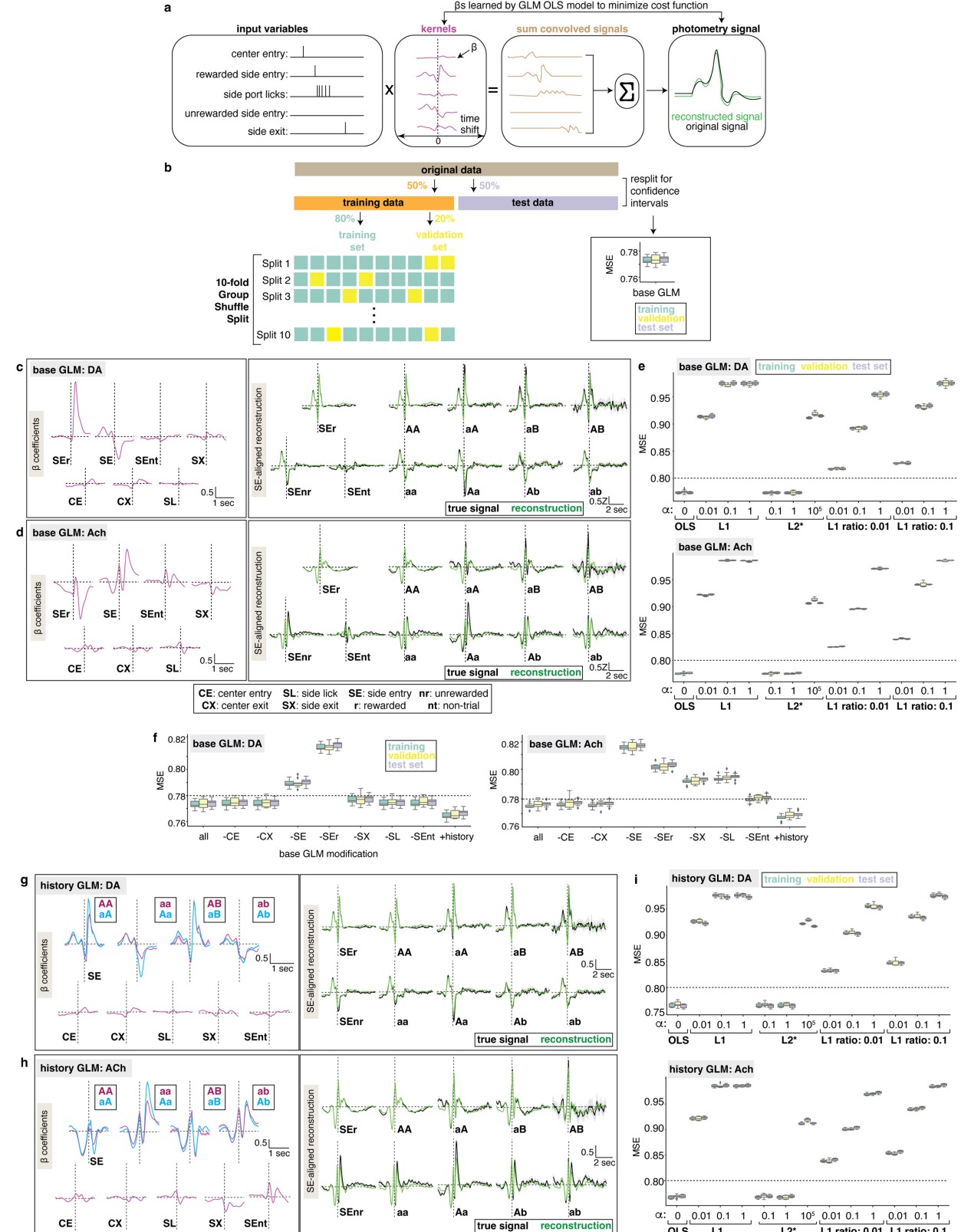

**Extended Data Fig. 5** | See next page for caption.

**Extended Data Fig. 5 | Generation and analysis of GLMs for DA and Ach signals. a**, GLM workflow. Input variables are convolved with their kernels, with each time step consisting of a separate $\beta$ coefficient fit by minimizing a cost function. The convolved signals are summed to generate the reconstructed signal. **b**, Evaluation of GLM performance. The original dataset is parsed into training and test sets. The GLM model is generated from the training set, and its performance is evaluated with MSE and $R^2$. To generate confidence intervals for the MSEs (mock plot shown), the data are resplit ten times for **f** and three times for **e** and **i**. **c**, Kernels and reconstructed DA signals for the base GLM. The average photometry signals with bootstrapped 95% C.I. and the average kernels ± s.d. are depicted ($n = 8$ mice). **d**, Kernels and reconstructed Ach signals for the base GLM. Data are depicted as in **c** ($n = 9$ mice). **e**, Different hyperparameter sweeps over regression models – OLS, lasso regression (L1), ridge regression (L2) and elastic net (L1 + L2), and effect on indicated MSEs of the DA and Ach base GLMs (DA: $n = 8$ mice; Ach: $n = 9$ mice). Box plots are displayed as quartiles (25%, 50% and 75% percentiles) with 1.5 × interquartile range for whiskers and outliers marked as points outside this range. **f**, The effect of omission (−) or inclusion (+) of the indicated input variables on GLM performance, as measured by the effect on indicated MSEs (DA: $n = 8$ mice; Ach: $n = 9$ mice). The box plots are displayed as in **e**. **g**, Kernels and reconstructed DA signals for the history GLM. Data are depicted as in **c** ($n = 8$ mice). **h**, Kernels and reconstructed Ach signals for the history GLM. Data are depicted as in **c** ($n = 9$ mice). **i**, The effect of different hyperparameter sweeps for history GLMs. Data are depicted as in **e** (DA: $n = 8$ mice; Ach: $n = 9$ mice).

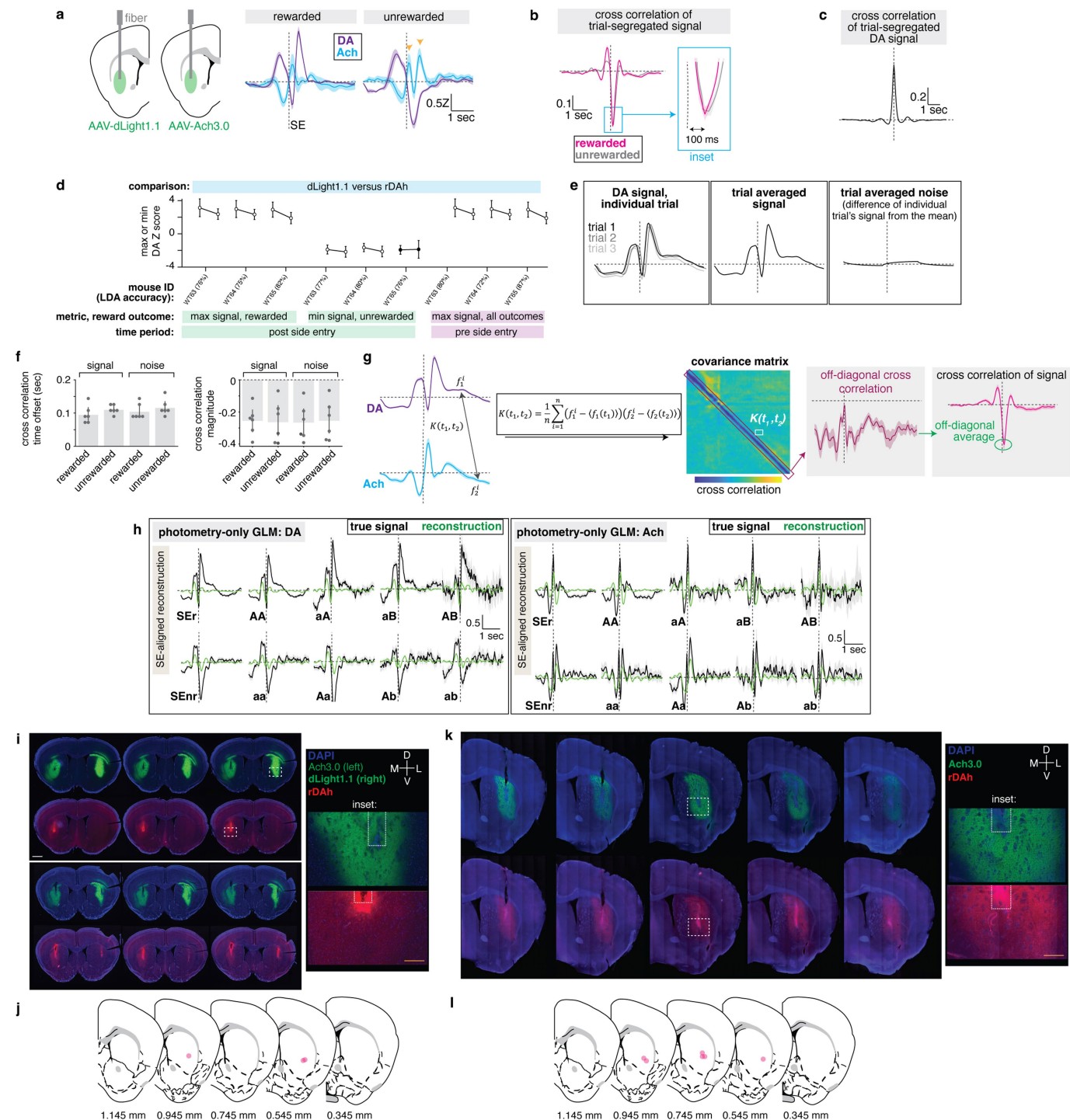

**Extended Data Fig. 6** | See next page for caption.

**Extended Data Fig. 6 | Covariance and GLM analyses of DA and Ach signals, and histology for simultaneous DA and Ach recordings. a**, Injection and implantation for recordings of DA and Ach release from separate mice. The average $z$-score of the sensor signal ± s.e.m. is shown (DA: $n = 7$; Ach: $n = 9$ mice). Orange arrows denote the double rise of Ach referenced in the main text. **b**, Cross-correlation of trial-segregated signals recorded from mice in **a**, in which DA lags Ach. The inset highlights the time offset of the minimum covariance signal (mean ± s.e.m.; DA: $n = 3$; Ach: $n = 3$ mice). **c**, Cross-correlation of trial-segregated DA signals recorded with dLight1.1 from opposite hemispheres of the same brain (mean ± s.e.m.; $n = 4$ mice). **d**, Analysis of dLight1.1 and rDAh signals in Fig. 2a. Paired comparisons are calculated from maximum or minimum signals during the indicated time periods. Data are depicted as in Extended Data Fig. 3a. Open circles represent a significant difference for each paired comparison (mean ± s.d.; two-sided $t$-test ($\underline{P} < 0.05$)). **e**, Schematic of the covariance analyses in Fig. 2e. Cross-correlation of the trial-averaged signal is calculated. The trial-averaged noise is the difference of each trial's signal from the overall mean. **f**, Quantification of the time shift and amplitude of the minimum covariances in Fig. 2e. **g**, Summary of the correlation analyses in Fig. 2f. A covariance matrix is built by calculating the cross-correlation ($K(t_1, t_2)$)

of the DA signal at one time point ($f_1^i$) to the Ach signal at all other time points and vice versa, using the indicated equation where $\langle f_1(t_1) \rangle$ is the mean across all trials for a certain time point, $t_1$. In this covariance matrix, the off-diagonal signal shows a striking negative cross-correlation (purple rectangle). The signals along this off-diagonal are plotted, and the average of this off-diagonal signal is equivalent to the minimum covariance value calculated from trial-averaged signals (green circle). **h**, Reconstructed and true photometry signals from GLMs that only incorporate a photometry variable. The average signals with bootstrapped 95% C.I. are shown. **i**, Images of dLight1.1 and rDAh expression for a representative mouse recorded for Fig. 2a. Coronal sections show the spread of expression across striatum. A higher-resolution image of the recording site (dashed white box) is shown with the fibre tract denoted (dashed white line). Scale bars (white): 1 mm; (orange): 0.2 mm. Although Ach3.0 was expressed in these mice, it was not recorded for Fig. 2a. **j**, Location of the optical fibre tip (pink dot) for mice recorded in Fig. 2a. **k**, Images of Ach3.0 and rDAh expression in the VLS for a representative mouse recorded in Fig. 2b. Images are depicted as in **i**. **l**, Location of the optical fibre tip (pink dot) for mice recorded in Fig. 2b.

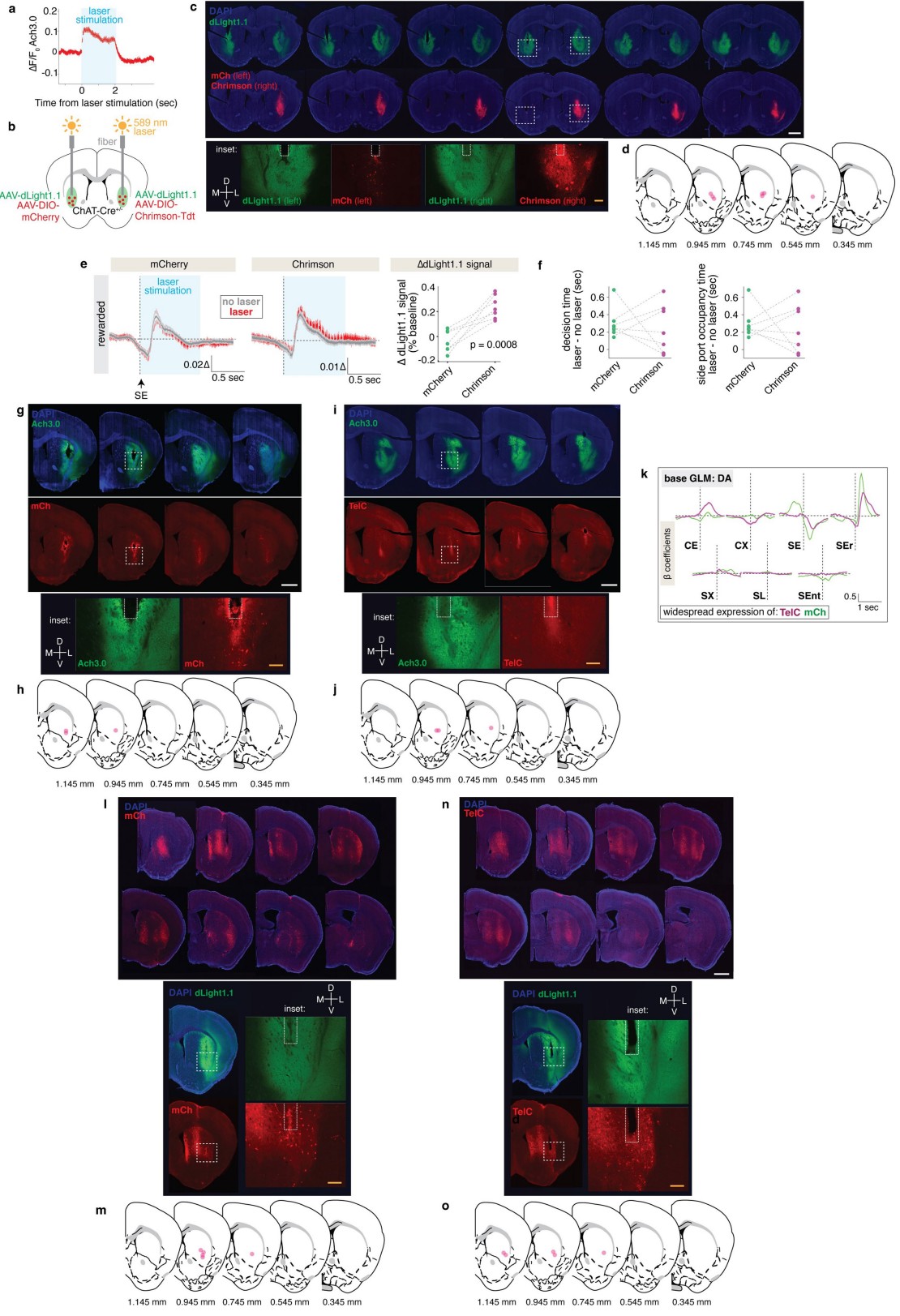

**Extended Data Fig. 7** | See next page for caption.

**Extended Data Fig. 7 | Optogenetic activation of VLS CINs, and histology for TelC expression in VLS CINs. a**, Ach release during optogenetic activation of Chrimson-expressing CINs. Average $\Delta F/F_0 \pm$ s.e.m. of Ach3.0 is depicted (laser stimulation artefacts omitted). **b**, Injection and optical fibre implantation for mice recorded in **e**. **c**, Images of dLight1.1, mCh, and Chrimson expression in the VLS for a representative mouse recorded in **e**. Coronal sections show spread of expression. A higher-resolution image of the recording site (dashed box) is shown with the fibre tract denoted (inset). Scale bars (white): 1 mm; (orange): 0.2 mm. **d**, Location of the optical fibre tip (pink dot) for mice recorded in **c**. **e**, DA signals during optogenetic activation of CINs. Average dLight1.1 $\Delta F/F_0 \pm$ s.e.m. is depicted ($n = 7$ mice). Laser stimulation artefacts are omitted. Quantification of the change in peak dLight1.1 signal from baseline is shown, with each connected pair of dots representing a unique mouse. Significance is calculated from a two-sided $t$-test ($P = 0.0008$). **f**, The decision time and side-port occupancy duration of mice with or without stimulation of the indicated hemispheres ($n = 7$ mice). Data are shown as in **e**. None of the comparisons are significant (two-sided $t$-test; $P > 0.05$). **g**, Images of Ach3.0 and mCh expression for a representative mouse recorded for Fig. 3d. Images are depicted as in **c**. **h**, Location of the optical fibre tip (pink dot) for mice recorded in Fig. 3d. **i**, Images of Ach3.0 and TelC expression for a representative mouse recorded for Fig. 3d. Histology is depicted as in **c**. **j**, Location of the optical fibre tip (pink dot) for mice recorded in Fig. 3d. **k**, DA kernels for the base GLM derived from Fig. 3e. The average kernel $\pm$ s.d. is shown. **l**, Images of dLight1.1 and striatum-wide mCh expression for a representative mouse recorded for Fig. 3e. Histology is depicted as in **c**. **m**, Location of the optical fibre tip (pink dot) for mice recorded in Fig. 3e. **n**, Images of dLight1.1 and striatum-wide TelC expression in the VLS for a representative mouse recorded for Fig. 3e. Histology is shown as in **c**. **o**, Location of the optical fibre tip (pink dot) for mice recorded in Fig. 3e.

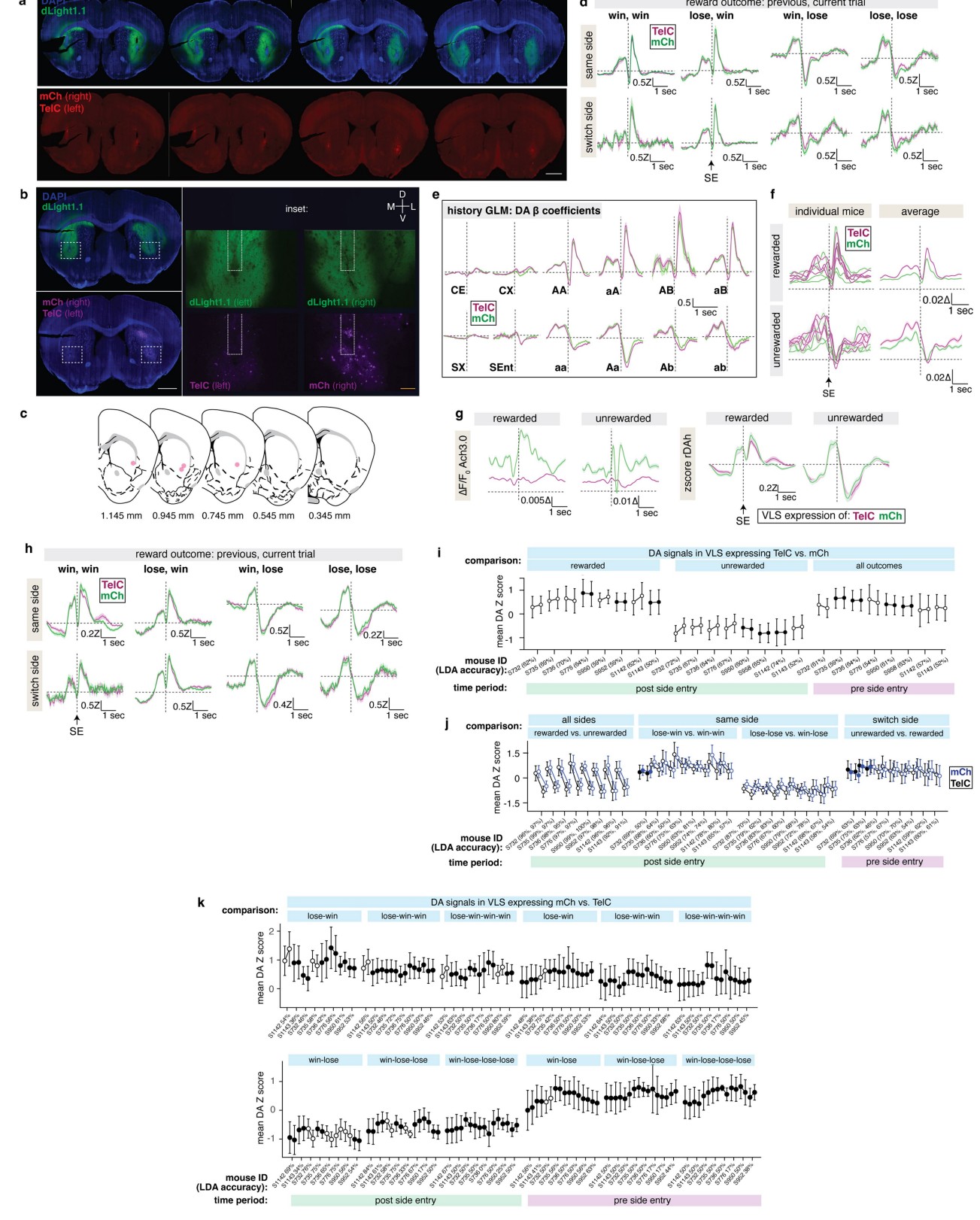

**Extended Data Fig. 8 |** See next page for caption.

**Extended Data Fig. 8 | Histology, photometry recordings and analysis of the effects of CIN-specific TelC expression in the VLS on DA signals. a**, Images of dLight1.1, mCh, and TelC expression in the VLS for a representative mouse recorded for Fig. 3f. Scale bar (white): 1 mm. **b**, Images of dLight1.1, mCh, and TelC expression for a representative mouse recorded for Fig. 3f in which mCh and TelC expression is amplified through antibody staining. Coronal sections show spread of expression. A higher-resolution image of the recording site (dashed box) is shown with the fibre tract denoted (inset). Scale bars (white): 1 mm; (orange): 0.2 mm. **c**, Location of the optical fibre tip (pink dot) for mice recorded in Fig. 3f. **d**, Effect of TelC or mCh expression in CINs on DA release parsed by the action-outcome history of the previous trial (mean $z$-score ± s.e.m.; $n$ = 4 mice). **e**, DA kernels for the history GLM derived from recordings in Fig. 3f with VLS expression of the indicated proteins in CINs. The average kernel ± s.d. are shown. **f**, Effect of TelC or mCh expression in CINs on DA release.

$\Delta F/F_0$ signals for individual mice and the average $\Delta F/F_0$ ± s.e.m. are shown ($n$ = 4 mice). **g**, Simultaneous recordings of Ach and DA from mice with VLS CINs expressing TelC or mCh (mean $z$-score ± s.e.m.; $n$ = 4 mice). **h**, Effect of TelC expression in CINs on DA transients from **g** parsed by action-outcome history of one trial back. Data are depicted as in **d**. **i**, Analysis of DA signals from mice in **g** and Fig. 3f. Paired comparisons are shown between DA signals recorded from hemispheres with CINs expressing TelC (left dot in connected pair) or mCh (right dot in connected pair). The mean signal ± s.d. are shown. Data are depicted as in Extended Data Fig. 3a. Open circles represent a significant difference for each comparison (two-sided $t$-test ($P$ < 0.05)). **j**, Analysis of DA signals from mice recorded in **g** and Fig. 3f. Data are depicted as in **i**. **k**, Analysis of DA signals from mice recorded in **g** and Fig. 3f parsed by the indicated action-outcome one, two and three trials back. Data are shown as in **i**.

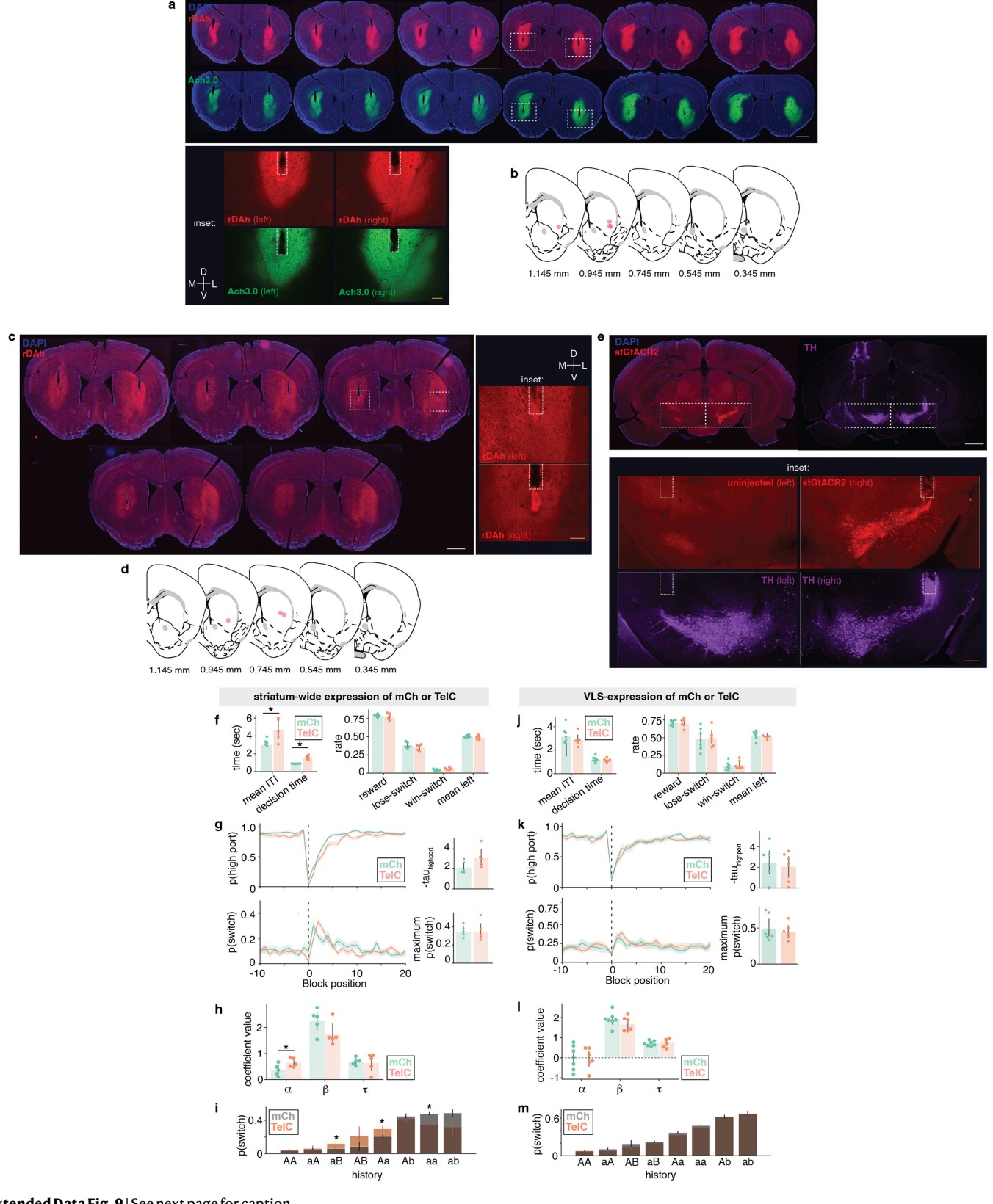

**Extended Data Fig. 9** | See next page for caption.

**Extended Data Fig. 9 | Histology and behaviour of CIN-specific TelC expression in the VLS and optogenetic inhibition of DANs. a**, Images of Ach3.0 and rDAh expression in the VLS for a representative mouse recorded in Extended Data Fig. 8g. Coronal sections show spread of expression. A higher-resolution image of the recording site (dashed box) is shown with the fibre tract denoted (inset). Scale bars (white): 1 mm; (orange): 0.2 mm. Note that TelC and mCh expression is in the same channel as rDAh and is therefore not visible. **b**, Location of the optical fibre tip (pink dot) for mice recorded in Extended Data Fig. 8g. **c**, Images of rDAh in the VLS for a representative mouse recorded for Fig. 3g. Images are depicted as in **a**. **d**, Location of the optical fibre tip (pink dot) for mice recorded in Fig. 3g. **e**, Images of stGtACR2 and tyrosine hydroxylase (TH) expression, a DAN marker, for a representative mouse recorded for Fig. 3g. Histology is depicted as in **a**. **f**, Performance metrics for mice in Fig. 3e with striatum-wide expression of TelC or mCh in CINs. Each dot represents a unique mouse. Bars denote mean with 95% C.I. Significance is determined with a two-sided $t$-test (*$P$: mean ITI = 0.023; decision time = 0.0001). **g**, Probability of occupancy at highly rewarded port or switching (mean ± s.e.m.) for mice in Fig. 3e. Calculated taus and maximum $P$(switch) rates are shown (mean with 95% C.I.) with each dot representing a unique mouse. **h**, RFLR coefficients of mice in Fig. 3e. Bars denote mean with 95% C.I. Significance is determined using a two-sided Wilcoxon rank-sum test (*$P$: $\alpha$ = 0.047). **i**, Conditional switch probabilities for the indicated action-outcome sequence for mice in Fig. 3e. Bars show the mean switch probability with the binomial standard error for the mouse test data. Significance (asterisk) is denoted as >95% C.I. from bootstrapped samples (one-sided, no adjustments). **j**, Performance metrics for mice from in Fig. 3f with VLS-selective CIN expression of TelC or mCh. Data are depicted as in **f**. **k**, Probability of occupancy at highly rewarded port or switching as a function of block position for mice in Fig. 3f. Data are depicted as in **g**. **l**, RFLR coefficients for mice in Fig. 3f. Data are depicted as in **h**. **m**, Conditional switch probabilities for mice in Fig. 3f. Data are depicted as in **i**.

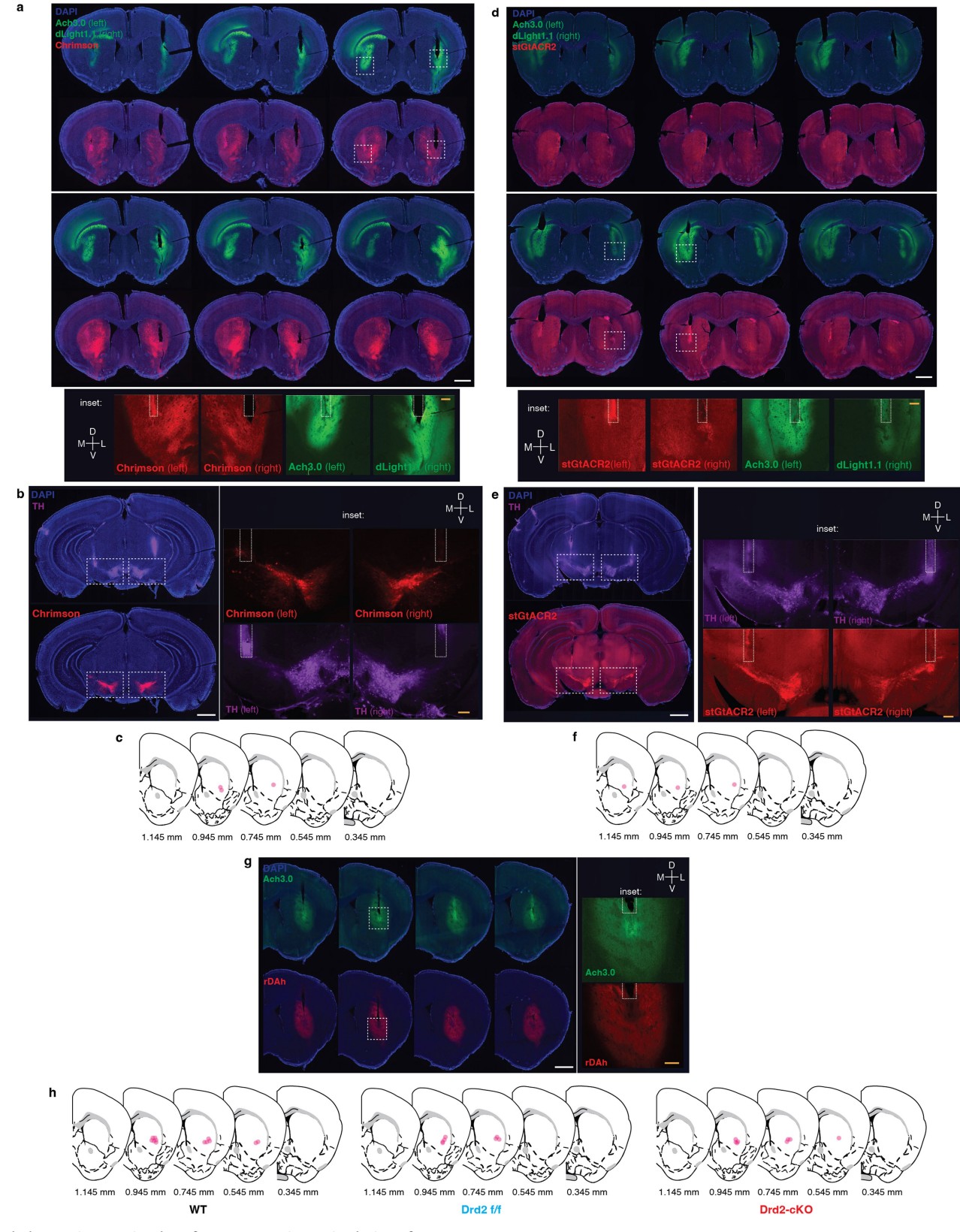

**Extended Data Fig. 10 | Histology for optogenetic manipulation of DANs and *Drd2* loss in CINs. a**, Images of dLight1.1, Ach3.0 and Chrimson expression for a representative mouse recorded for Fig. 4b. Coronal sections show spread of expression. A higher-resolution image of the recording site (dashed box) is shown with the fibre tract denoted (inset). Scale bars (white): 1 mm; (orange): 0.2 mm. **b**, Images of Chrimson and TH expression in DANs for a representative mouse recorded for Fig. 4b. Histology is depicted as in **a. c**, Location of the optical fibre tip (pink dot) for mice recorded in Fig. 4b. **d**, Images of dLight1.1,

Ach3.0 and stGtACR2 expression for a representative mouse recorded for Fig. 4c. Histology is shown as in **a. e**, Images of stGtACR2 and TH expression in midbrain DANs for a representative mouse recorded for Fig. 4c. Histology is depicted as in **a. f**, Location of the optical fibre tip (pink dot) for mice recorded in Fig. 4c. **g**, Images of Ach3.0 and rDAh expression for a representative mouse recorded for Fig. 4g. Histology is depicted as in **a. h**, Location of the optical fibre tip (pink dot) for mice recorded in Fig. 4g.

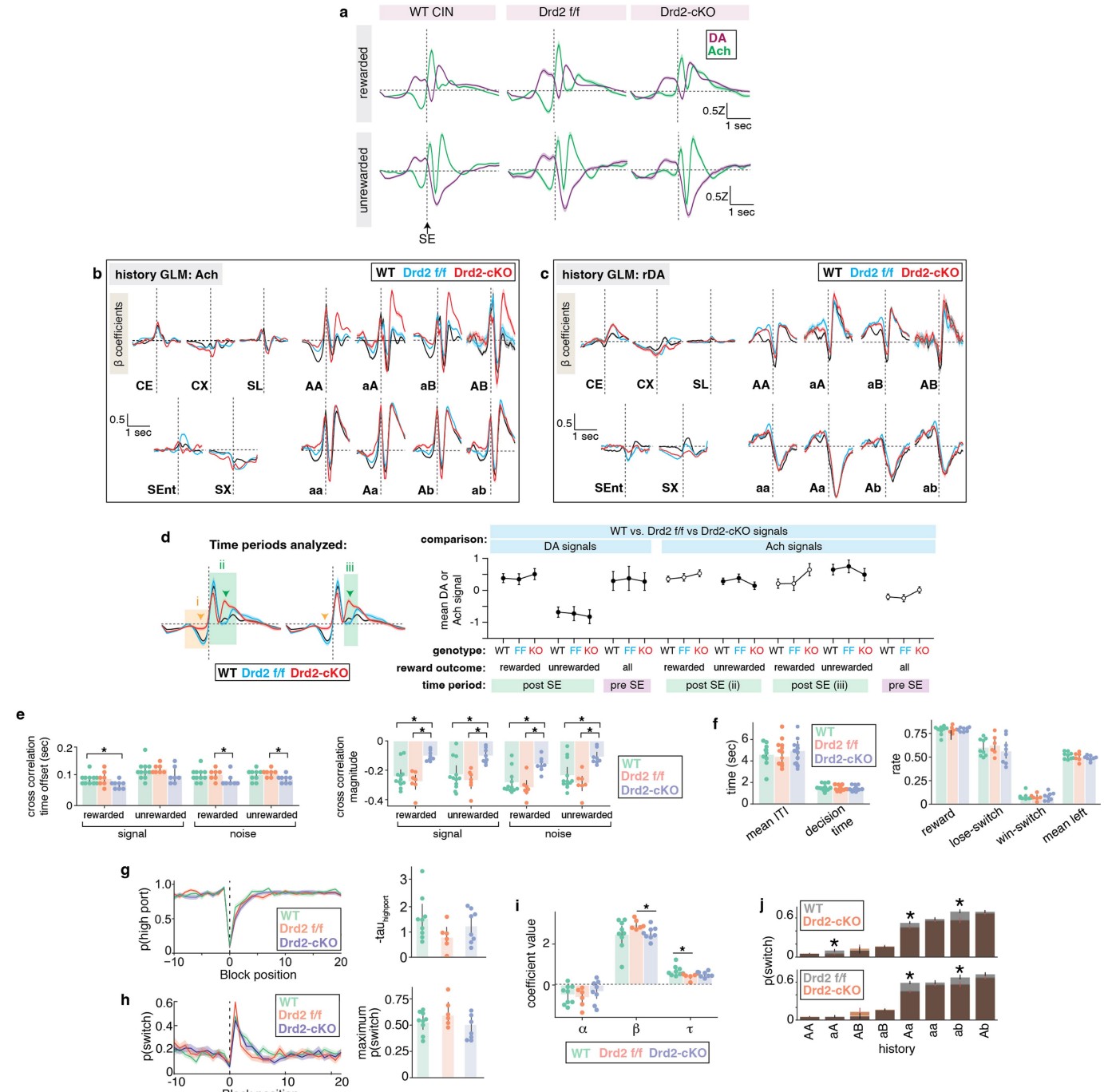

**Extended Data Fig. 11 | Photometry, GLM and behavioural analyses of DA and Ach signals from mice lacking D2R expression in CINs. a**, Ach and DA release from mice in Fig. 4g (mean ± s.e.m.; WT: $n$ = 12 mice; Drd2 f/f: $n$ = 7 mice; Drd2-cKO: $n$ = 8 mice). **b**, Kernels for the history GLM of Ach signals from mice in Fig. 4g. Average kernels ± s.d. are depicted. **c**, Kernels for the history GLM of rDAh signals from mice in Fig. 4g. Data are displayed as in **b**. **d**, Statistical analysis of DA and Ach signals in Fig. 4g. Schematic of the time periods for analysis (left). Comparisons are shown as mean ± s.d. between the indicated genotypes (WT: ChAT-Cre; FF: Drd2 f/f; KO: Drd2-cKO). Open circles denote a significant difference across the three genotypes for a particular condition (two-sided ANOVA, open circle, $P$ > 0.05). A two-sided $t$-test ($P$ < 0.05) confirmed that KO signals were significantly different from FF and WT. **e**, Time shifts and amplitudes of the minimum covariance in Fig. 4h. Bars denote mean with 95% C.I. Significance is calculated using a two-sided $t$-test ($P$ < 0.05,

asterisk; values reported in Supplementary Data 1). **f**, Performance metrics for mice in Fig. 4g. Each dot represents a unique mouse. Bars denote mean with 95% C.I. No significant differences were observed (two-sided $t$-test, $P$ > 0.05). **g**, Probability of occupancy at the highly rewarded port (mean ± s.e.m.) for mice in Fig. 4g. Calculated taus are shown (mean with 95% C.I.), with each dot representing a unique mouse. **h**, Probability of switching (mean ± s.e.m.) for mice in Fig. 4g. Average maximum $P$(switch) rates are shown with 95% C.I., with each dot representing a unique mouse. **i**, RFLR coefficients for mice in Fig. 4g. Bars denote mean with 95% C.I. Significance is determined using a two-sided Wilcoxon rank-sum test (*$P$: $\beta$ = 0.014; $\tau$ = 0.018). **j**, Conditional switch probabilities for the indicated action-outcome sequence for mice in Fig. 4g. Bars show the mean switch probability with the binomial standard error for the mouse test data. Significance (asterisk) is denoted as >95% C.I. from bootstrapped samples (one-sided, no adjustments).

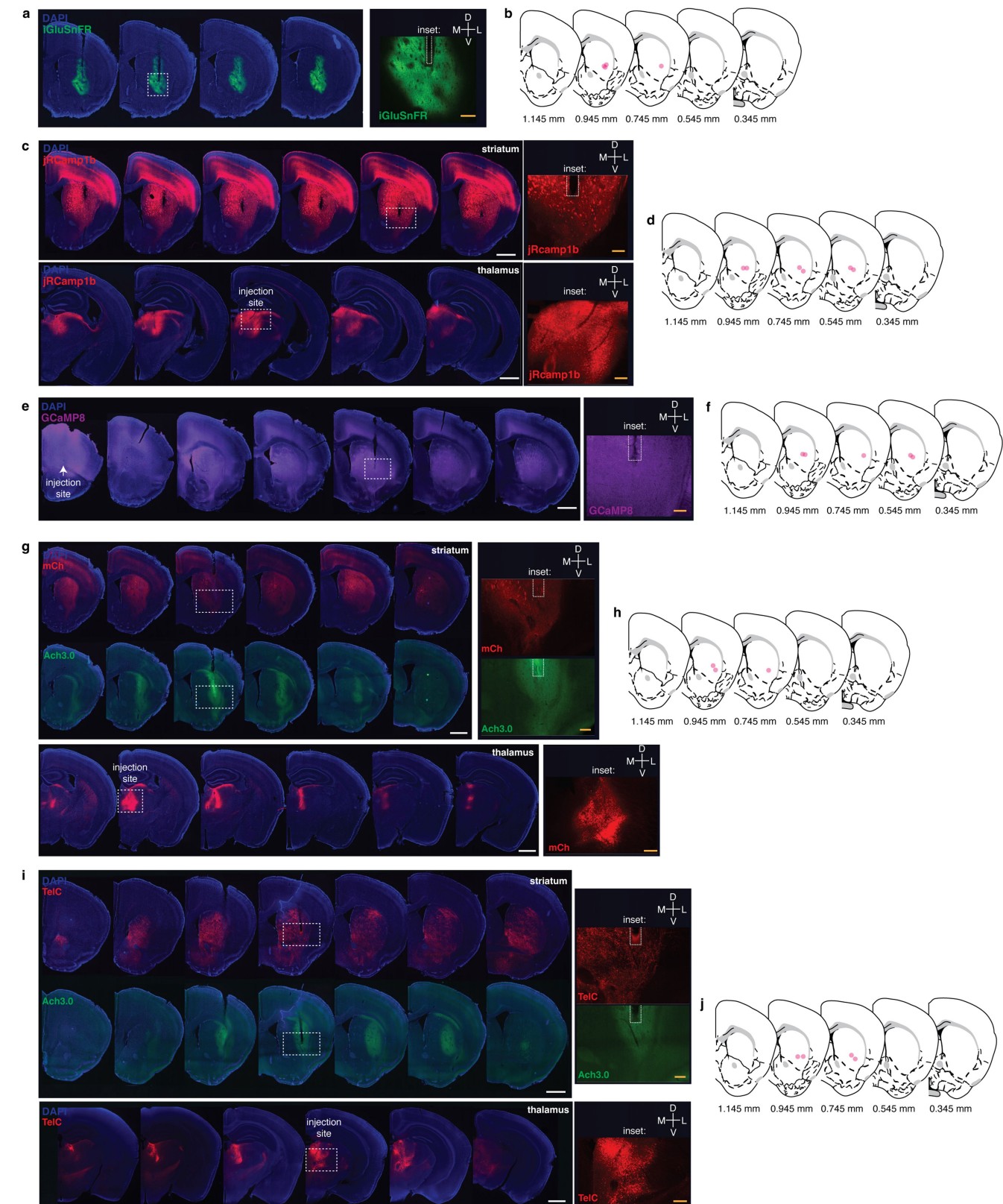

**Extended Data Fig. 12** | See next page for caption.

**Extended Data Fig. 12 | Histology for glutamate release, cortical and thalamic calcium signals and thalamic inhibition. a**, Images of iGluSnFR expression for a representative mouse recorded for Fig. 5b–d. Coronal sections show spread of expression. A higher-resolution image of the recording site (dashed box) is shown with the fibre tract denoted (inset). Scale bars (white): 1 mm; (orange): 0.2 mm. **b**, Location of the optical fibre tip (pink dot) for recorded in Fig. 5b–d. **c**, Images of jRCamp1b expression in thalamus for a representative mouse recorded for Fig. 5d. Histology is shown as in **a**. **d**, Location of the optical fibre tip (pink dot) for mice recorded in Fig. 5d. **e**, Images of cortical GCaMP8 expression for a representative mouse recorded for Fig. 5d. Histology is shown as in **a**. **f**, Location of the optical fibre tip (pink dot) for mice recorded in Fig. 5d. **g**, Images of Ach3.0 and mCh expression for a representative mouse recorded for Fig. 5e. Histology is shown as in **a**. **h**, Location of the optical fibre tip (pink dot) for mice recorded in Fig. 5e. **i**, Images of Ach3.0 and TelC expression for a representative mouse recorded for Fig. 5e. Histology is depicted as in **a**. **j**, Location of the optical fibre tip (pink dot) for mice recorded in Fig. 5e.

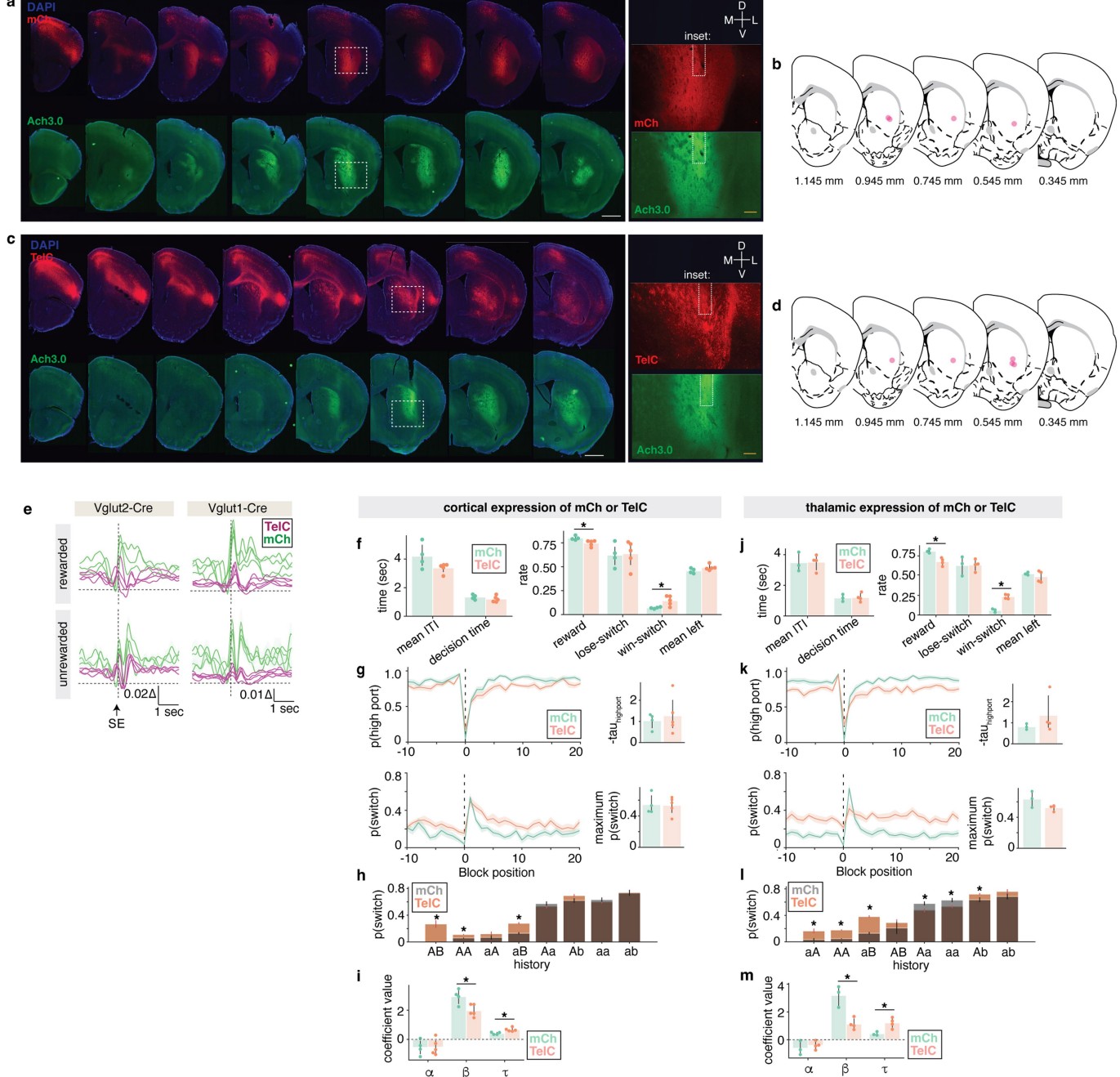

**Extended Data Fig. 13 | Histology for cortical inhibition, and behavioural effects of cortical and thalamic inhibition. a**, Images of Ach3.0 and mCh expression for a representative mouse recorded for Fig. 5f. Coronal sections show the spread of expression. A higher-resolution image of the recording site (dashed white box) is shown with the fibre tract denoted. Scale bars (white): 1 mm; (orange): 0.2 mm. **b**, Location of the optical fibre tip (pink dot) for mice recorded in Fig. 5f. **c**, Images of Ach3.0 and TelC expression for a representative mouse recorded for Fig. 5f. Histology is depicted as in **a**. **d**, Location of the optical fibre tip (pink dot) for mice recorded in Fig. 5f. **e**, Effect of TelC or mCh expression in Vglut2-Cre or Vglut1-Cre mice on Ach release (mean $\Delta F/F_0 \pm$ s.e.m.). **f**, Performance metrics for mice in Fig. 5e. Each dot represents a unique mouse. Bars denote mean with 95% C.I. Significance is determined by a two-sided *t*-test (*$P$: reward rate = 0.022; win-switch = 0.025). **g**, Probability of occupancy at highly rewarded port or switching (mean ± s.e.m.) for mice in

Fig. 5e. Calculated taus and maximum $P$(switch) rates are shown (mean with 95% C.I.) with each dot representing a unique mouse. **h**, Conditional switch probabilities for the indicated action-outcome sequence for mice in Fig. 5e. Bars show the mean switch probability with the binomial standard error for the mouse test data. Significance (asterisk) is denoted as >95% C.I. from bootstrapped samples (one-sided, no adjustments). **i**, RFLR coefficients for mice in Fig. 5e. Bars denote mean with 95% C.I. Significance is determined using a two-sided Wilcoxon rank-sum test (*$P$: $\beta$ = 0.05; $\tau$ = 0.014). **j**, Performance metrics for mice in Fig. 5f. Data are depicted as in **f** (*$P$: reward rate = 0.008; win-switch = 0.0003). **k**, Probability of occupancy at highly rewarded port or switching for mice in Fig. 5f. Data are depicted as in **g**. **l**, Conditional switch probabilities of the indicated treatment groups for mice in Fig. 5f. Data are depicted as in **h**. **m**, RFLR coefficients for mice in Fig. 5f. Data are depicted as in **i** (*$P$: $\beta$ = 0.034; $\tau$ = 0.034).

# Reporting Summary

## Statistics

For all statistical analyses, confirm that the following items are present in the figure legend, table legend, main text, or Methods section.

| n/a | Confirmed | |
|---|---|---|
| ☐ | ☒ | The exact sample size (*n*) for each experimental group/condition, given as a discrete number and unit of measurement |
| ☐ | ☒ | A statement on whether measurements were taken from distinct samples or whether the same sample was measured repeatedly |
| ☐ | ☒ | The statistical test(s) used AND whether they are one- or two-sided *Only common tests should be described solely by name; describe more complex techniques in the Methods section.* |
| ☐ | ☒ | A description of all covariates tested |
| ☐ | ☒ | A description of any assumptions or corrections, such as tests of normality and adjustment for multiple comparisons |
| ☐ | ☒ | A full description of the statistical parameters including central tendency (e.g. means) or other basic estimates (e.g. regression coefficient) AND variation (e.g. standard deviation) or associated estimates of uncertainty (e.g. confidence intervals) |
| ☐ | ☒ | For null hypothesis testing, the test statistic (e.g. *F*, *t*, *r*) with confidence intervals, effect sizes, degrees of freedom and *P* value noted *Give P values as exact values whenever suitable.* |
| ☐ | ☒ | For Bayesian analysis, information on the choice of priors and Markov chain Monte Carlo settings |
| ☐ | ☒ | For hierarchical and complex designs, identification of the appropriate level for tests and full reporting of outcomes |
| ☐ | ☒ | Estimates of effect sizes (e.g. Cohen's *d*, Pearson's *r*), indicating how they were calculated |

*Our web collection on statistics for biologists contains articles on many of the points above.*

## Software and code

Policy information about availability of computer code

| Data collection | OlyVIA software was used to acquire histology images. MATLAB_R2017a, LabJack, and Arduino software were used to acquire photometry and behavioral data |
|---|---|
| Data analysis | MATLAB_R2017b, MATLAB_R2021b, and Python3.0 were used to analyze all the data presented. |

For manuscripts utilizing custom algorithms or software that are central to the research but not yet described in published literature, software must be made available to editors and reviewers. We strongly encourage code deposition in a community repository (e.g. GitHub). See the Nature Portfolio guidelines for submitting code & software for further information.

## Data

Policy information about availability of data

All manuscripts must include a data availability statement. This statement should provide the following information, where applicable:
- Accession codes, unique identifiers, or web links for publicly available datasets
- A description of any restrictions on data availability
- For clinical datasets or third party data, please ensure that the statement adheres to our policy

The data and code that supports the findings of this study are available upon request from the corresponding author.

# Research involving human participants, their data, or biological material

Policy information about studies with [human participants or human data](). See also policy information about [sex, gender (identity/presentation), and sexual orientation]() and [race, ethnicity and racism]().

| | |
|---|---|
| Reporting on sex and gender | No human subjects were used. |
| Reporting on race, ethnicity, or other socially relevant groupings | n/a |
| Population characteristics | n/a |
| Recruitment | n/a |
| Ethics oversight | n/a |

Note that full information on the approval of the study protocol must also be provided in the manuscript.

# Field-specific reporting

Please select the one below that is the best fit for your research. If you are not sure, read the appropriate sections before making your selection.

☒ Life sciences ☐ Behavioural & social sciences ☐ Ecological, evolutionary & environmental sciences

For a reference copy of the document with all sections, see [nature.com/documents/nr-reporting-summary-flat.pdf]()

# Life sciences study design

All studies must disclose on these points even when the disclosure is negative.

| | |
|---|---|
| Sample size | For all experiments, a sample size of at least three was chosen. We determined this to be sufficient based on low variability observed between the photometry signals and behavioral performance we recorded. |
| Data exclusions | Data was not excluded from the analysis. |
| Replication | For all experiments, a replica number of at least three was chosen. We determined this to be sufficient based on low variability observed between the photometry signals and behavioral performance we recorded. |
| Randomization | There was no requirement for randomization. |
| Blinding | In comparisons between different treatment groups, the animals were each assigned a unique identification number which did not reveal their treatment. |

# Reporting for specific materials, systems and methods

We require information from authors about some types of materials, experimental systems and methods used in many studies. Here, indicate whether each material, system or method listed is relevant to your study. If you are not sure if a list item applies to your research, read the appropriate section before selecting a response.

## Materials & experimental systems

| n/a | Involved in the study |
|---|---|
| ☐ ☒ | Antibodies |
| ☒ ☐ | Eukaryotic cell lines |
| ☒ ☐ | Palaeontology and archaeology |
| ☐ ☒ | Animals and other organisms |
| ☒ ☐ | Clinical data |
| ☒ ☐ | Dual use research of concern |
| ☒ ☐ | Plants |

## Methods

| n/a | Involved in the study |
|---|---|
| ☒ ☐ | ChIP-seq |
| ☒ ☐ | Flow cytometry |
| ☒ ☐ | MRI-based neuroimaging |

# Antibodies

| | |
|---|---|
| Antibodies used | goat anti-Choline acetyltransferase (Millipore Sigma #AB144P; 1:200) <br> mouse anti-tyrosine hydroxylase (Immunostar #22941; 1:1000) |

chicken anti-GFP (Abcam ab13970; 1:1500)
rabbit anti-GFP (Novus Biologicals #NB600-308; 1:1000)
rabbit anti-mCherry (Takara Bio #632496; 1:1000)
rabbit anti-GFAP (Abcam ab7260; 1:1500)
Multiple lots of each antibody were used for this manuscript, all of which had equivalent performance.

Validation | All antibodies were validated by the respective manufacturers to work for immunohistochemistry, as stated on their online product pages. The rabbit anti-mCherry was only stated as applicable for use in western blots, but we performed our own validation to confirm that it specifically detects mCherry expressed in mouse brains (i.e. no signal was detected in the absence of fluorophore expression or in the presence of non-mCherry fluorophores, notably GFP).

# Animals and other research organisms

Policy information about studies involving animals; ARRIVE guidelines recommended for reporting animal research, and Sex and Gender in Research

Laboratory animals | The following mice lines were used: C57BL6/J (Jackson labs #000664); ChAT-IRES-Cre (Jackson labs #006410); DAT-IRES-Cre (Jackson labs #006660), Drd2loxP (Jackson labs #020631); Vglut2-IRES-Cre (Jackson labs #028863); Vglut1-IRES-Cre (Jackson labs #023527). All mice were bred on a C57BL/6J genetic background and heterozygotes were used unless noted. For behavior experiments, males at 6-8 weeks of age were used.

Wild animals | No wild animals were used.

Reporting on sex | For behavior experiments, males at 6-8 weeks of age were used. Only males were used to avoid any behavioral variation due to the estrous cycle in female mice and because of recent findings that only male behavior is affected by loss of muscarinic Ach receptors (reference: Razidlo, J. A. et al. Chronic loss of muscarinic M5 receptor function manifests disparate impairments in exploratory behavior in male and female mice despite common dopamine regulation. J. Neurosci. JN-RM-1424-21 (2022). doi:10.1523/JNEUROSCI.1424-21.2022)

Field-collected samples | n/a

Ethics oversight | All animal care and experimental manipulations were performed in accordance with protocols approved by the Harvard Standing Committee on Animal Care, following guidelines described in the US NIH Guide for the Care and Use of Laboratory Animals.

Note that full information on the approval of the study protocol must also be provided in the manuscript.

