## [Peer Review File · Nature]

Manuscript Title: Dopamine and glutamate regulate striatal acetylcholine in decision-making

Reviewer Comments & Author Rebuttals

Reviewer Reports on the Initial Version:

Referees' comments:

Referee #1 (Remarks to the Author):

The authors in this paper interrogate the contribution of CINs to striatal function. Despite the detailed understanding of the mechanisms of interactions between DA and CIN in vitro, how and when this crosstalk happens in vivo remains largely unknown. To understand the factors that shape Ach release during reward-based decision-making and how Ach release contributes to DA release, the authors examined striatal Ach and DA levels during a complex decision-making task in mice. The authors demonstrated that DA dynamics and RPE are independent of Ach release, but DA inhibits Ach levels in a D2-dependent manner. The authors then characterized the inputs for the cortex and thalamus and found that Glu released by both inputs is required for Ach release.

As a whole, I thought this was a fascinating manuscript that attempted to address a very ambitious question. The authors provided data that indicate that Ach release does not modulate the DA release dynamic in the striatum, while DA influences Ach signals via D2Rs. While some of the data are convincing, part of the analysis is confusing and challenging to comprehend. In particular, the behavioral analysis and the DA and Ach GLMs analysis are difficult to interpret, and the conclusions regarding the role of DA and Ach in action and reward history are not convincing. Several debates have been experimentally addressed in this manuscript, but the conclusions are often more confusing than the debates themselves. The contribution of thalamic and cortical inputs is, for example, puzzling, and the conclusions are not entirely clear.

Suggestions:

The authors used a two-armed bandit task and measured DA and Ach signals with fiber photometry in the ventrolateral portion of the dorsal striatum. This figure (Extended figure 1) contains much interesting information about mouse performance and behavioral modeling, but it is hard to read and comprehend.

In Figure 1, quantitative comparisons between DA and Ach signals would be appropriate and clarify the message. Furthermore, the choice and reward outcome history in figure 1, panels f and d, is quite confusing.

To assess the contribution of each behavioral variable to GLM performance, the authors performed a "leave-out analysis" of individual features and analyzed the correlation of Ach and DA signals during reward-based decision-making. The figures (extended figures 3, 4 and 5 as well as Figure 2) are confusing and do not exhaustively convey the conclusions.

In Figure 3, the authors assessed if Ach release is sufficient to modulate DA levels in vivo. The experiment presented in extended figure 6 in the head-fixed mouse is unclear. Furthermore, in this supplementary figure, there is an increase in signal in both mCherry and Chrimson in laser and no laser conditions. The text that corresponds to figure 3d is confusing. Are the authors indeed measuring Ach or DA release with carbon fibers?

The difference in decision time between TeIC and mCH control in extended figure 8 a is not convincing and difficult to interpret. It is difficult to understand the rationale of panel 3g and to comprehend the contrasting results between striatum versus VLS selective loss of Ach release.

Moreover, at the end of the paragraph on page 10, the authors write: “

...modulation of CIN activity is sufficient to drive DA release in vivo”. The experiments that support this conclusion are not clear to me. The experiments showed in figure 3 indeed, indicate an absence of Ach regulation of DA dynamics.

In figure 4, the authors assessed whether optogenetic manipulation of DA neurons in vivo affects Ach levels and whether this effect depends upon D2Rs. Very cleverly, the authors have knock-out the D2s, specifically in CINs. Quantitative analysis is missing and the behavioral differences between groups in extended figure 10 are unclear and do not support the firm conclusion that D2Rs are required for a specific moment within a trial and the switch across multiple histories.

In figure 5, the authors examine the additional inputs required to stimulate Ach release. According to figure 5c, not only do the glu signals increase during consumption, but it is also the case during unrewarded trials that these signals are reduced. Is there a possible explanation for this?

The behavioral effects perturbing the cortical and thalamic inputs into the VLS are puzzling, and based on the presented data, it is challenging to conclude regarding possible differences. Strong differences in the unrewarded trials can be observed in figure 5. How do the authors explain this finding? The results do not convincingly demonstrate the role of cortical and thalamic inputs in the interaction between Ach and DA.

Referee #2 (Remarks to the Author):

In their study, Chantranupong et al. characterized the release of dopamine (DA) and acetylcholine (Ach) in the ventrolateral striatum (VLS) during the decision phase of an operant behavior retrieval test. While it contains interesting observations, some of the conclusions are not directly supported by statistical analysis, and the manuscript is written in a very technical way, making it difficult to follow for a broad readership.

Specific Comments _____

First, the role of DA control on cholinergic interneurons (CIN) through CIN-specific depletion of the DA receptor 2 during a reward-based task had been shown previously, decreasing the novelty of the observations made in this study (see Martyniuk, K. M. et al. Dopamine D2Rs Coordinate Cue-Evoked Changes in Striatal Acetylcholine Levels. *Elife* 2022, cited in the manuscript). Thus, the rationale of this study as stated by the authors, is to provide a “more complete understanding” of DA and Ach

release during reward decision making, which suggest their study belongs to a more specialized journal.

Regarding the statistical analysis, although the authors used a sophisticated approach with a general linear model (GLM) to predict neuromodulator levels from behaviors, overall the study lacks statistical analysis to compare features of DA and Ach signals and their correlations in different conditions. For example, the authors claim that “reward outcome clearly modulates Ach transients”. However, there is no statistical comparisons of signals parameters between rewarded and unrewarded trials (amplitude, time to peak, duration, AUC...). Similarly, Page 5 line 125 the authors claim the “magnitude of the Ach transient is increased during center-to-side transition and decreased during reward acquisition”. The authors should provide a within animal statistical comparison of the amplitudes during the regular and omission trials to support this claim (for example as bar graph beside the trace). This comment applies to the entire manuscript, including to the lateralization analysis (ipsi/contra) which is critical to the entire manuscript.

In the abstract and manuscript the authors state that “DA modulates Ach levels only at specific times”. To confirm this claim, the authors performed a correlation between DA and Ach signals around the side port entry (SE) that they compare to correlation of shuffled data. However, a more rigorous approach would be to select random time points outside of the trials (as many time points as the number of trials over the entire session) to compare with the correlations around the side port entry (Figure 2c). The explanation of the shuffling method is unclear (Page 21 Line 705) and does not inform about the correlation out of the SE trials.

From my understanding the ‘noise analysis’ might correspond to the needed analysis (Figure 2d). The authors analyzed DA and Ach signals co-variation which revealed a sequence of positive/negative/positive co-variation. However, this co-variation appears to be similar in the ‘noise’ and the SE trials. Here as well, statistical comparisons of the co-variations for noise and SE trials should be performed. Finally, as the “noise” is recorded signals and not artefacts ‘noise’ should be renamed “ITI signal” (inter-trial interval signal).

Again, regarding quantification of the results, Page 7 line 205 the author claim “DA lags Ach reveals an anti-correlation that occurs with a positive time lag (Extended Data Fig. 5b)”. While potentially interesting the authors should provide quantification of this lag and statistical comparisons of it during trials and ITI. Similarly on Page 8 Line 230, the authors mention “100 ms lag”, however, it is unclear from where this value has been extracted (Figure 2f is cited).

Finally, the study should include at least one representative high resolution confocal image of neurons (or terminals) located under the optic fiber and expressing each of the tools used in the study. Similarly, at least one representative image of a brain section showing the spread of viral expression should be included for each tool. The location of optic fiber implants of each animal should also be represented on an atlas drawing for each experimental group to evaluate the potential variability of the recording locations.

Minor Comments _____

As there is no difference in recordings of DA and Ach signals from the ipsi- and controlateral side of the port entry, the authors could move this part of the analysis in the supplementary Figure, and

bring in some of the behavioral analysis of Extended Figure 1 in the main Figure 1.

The diagram in Figure 1a is confusing, as it suggests that nAChR reached by the CIN Ach are located in the VTA – DAN should be preplace with DAN-terminals, and VL striatum should be written on top of the diagram.

Figure 1e: please replace +LE and – LED with cue(+) and cue(omitted)

Page 5 line 123: replace ‘more unexpected’ with ‘less expected’

The representation of the AA, aA, Aa and aa cases is confusing (Figure 1f-g). First, each case should have a different color (gradients of the same color for each neurotransmitter?) and be represented in 4 different graphs on the top of each other. Indicating the full legend on the Figure would also be helpful: AA = 2 rewarded trials on the same side, aA: unrewarded trial followed by a rewarded one on the same side...

Figure 2a could be moved to the extended data, and needs statistical comparison to validate the absence of differences between dLight1.1 and rDAh signals.

Figure 5g: the authors locked the signals to the ‘decision’ of the animal. However, it is not possible to know at a millisecond timescale when the decision occurs. The decision might be taken anytime between trial onset and the movement onset toward the delivery port. As stated in the legend, the line represent trial onset and should be labeled as such.

Regarding mice, as they were water restricted to 1 mL per day, this suggest animas were single housed. This should be specified. The fact that this study is limited to male mice is reducing the impact of the findings and should be discussed. Moreover, the arguments provided by the authors in the methods suggesting potentially more variability in females due to the estrous cycle is not always valid as testosterone levels and social rank in males have also been show to heavily influence mouse behavior and increase inter-individual variability.

Page 13 Line 427: DA and Ach are critical neurotransmitters that directly modulate each other’s release in vitro *in the striatum*.

Page 15 Line 483: it is unclear what the authors mean by cholinergic Ach

Referee #3 (Remarks to the Author):

A large body of mostly ex vivo work has led to the hypothesis that striatal cholinergic interneurons (CINs) can facilitate local dopamine release by activating nicotinic receptors on dopamine neuron (DAN) axons, and that local dopamine release can negatively regulate CIN activity (and thereby acetylcholine (ACh) release) by activating D2 DA receptors on CINs. The current study by Chantranupong et al. examines whether these putative interactions occur in vivo within the ventrolateral striatum (VLS) of mice performing a reward motivated decision making task (two-

armed bandit task; 2ABT). Fiber photometry was used to record VLS DA and ACh signals (via genetically-encoded DA and ACh sensors) during task performance, which revealed distinct multiphasic DA and ACh responses. Computational modeling revealed that momentary changes in DA and ACh release depended on important task events (response initiation and reward receipt) and recent training history. Consistent with previous research, surges in DA occurred during response (side port entry) initiation and reward retrieval, which were temporally associated with dips in ACh. Bidirectional optogenetic manipulations of VLS DA terminals confirmed that there is substantial D2 receptor-mediated negative regulatory influence over local ACh release in vivo. Moreover, genetic deletion of D2 receptors in CINs abolished DA-induced inhibition of CIN activity and disrupted the apparent inhibitory influence of DA on task-related ACh release. Two other components of task-related ACh release (surges in ACh during response execution and reward retrieval/consumption) were shown to be driven by glutamatergic inputs from cortex and thalamus, addressing an important gap in knowledge. A separate series of experiments examined the role CINs in regulating DA release. These results were somewhat less compelling, but were nevertheless taken as evidence that in vivo DA release dynamics do not require ACh release by CINs.

This study provides important new information about how these two important neuromodulatory systems interact in the VLS during reward-related decision making. The methods are generally rigorous and well-controlled, and take advantage of cutting-edge and sophisticated techniques and analyses. The results are likely to be of interest to a broad audience primarily within the neuroscience community. The authors do a good job of discussing some of the limitation of their findings and identifying areas for future research. I have identified a few other limitations of the current study below.

- 1) Statistical support is not reported for many of the main conclusions about photometric data, despite claims about significance. This is likely a trivial matter in the majority of cases, but the lack of a consistent and clear approach for discussing findings may obscure some effects.
- 2) In particular, the authors conclude that there ACh release by CINs exerts little if any influence over DA release, or task-related DA dynamics. There is some recognition in the text that optogenetic CIN stimulation augmented DA release during reward delivery (Figure ED 6C), but this result is not indicated in the figure.
- 3) Likewise, the effects of TelC expression in CINs on task-related DA release (Fig 3e and ED 7b,c) are discussed in broad terms without statistical support. The authors acknowledge that there TelC expression was associated with larger amplitude DA signals, but then dismissed this as potentially reflecting variation in autofluorescence or sensor expression. This leaves the question somewhat open. While the z-score data suggest that the basic features of task-related DA dynamics were preserved in this condition, this does not necessarily address whether CINs regulate the absolute magnitude of task-related DA release. Given that the authors are attempting to conclude that there is little ACh modulation of DA, a more in depth analysis of the data by training history may be warranted.
- 4) Moreover, there is clearer evidence that striatum-wide CIN TelC expression dysregulated DA dynamics (Fig 3f, ED 7e), though again these effects are not explicitly indicated or highlighted. The authors conclude that because this broader manipulation of CIN function led to behavioral alterations, it is difficult to interpret changes DA signals. I agree that this does pose some challenges, but the photometry data are separated into epochs based on specific behavioral events, which helps

control for behavioral variation. The authors also note that the reward-prediction error (RPE) like DA signals (increase during reward and dip following absence of reward) was “sustained” following system-wide ACh depletion, but this statement deserves some scrutiny. Was there a quantitative change in the RPE-like signal, as appears to be the case? The data do suggest that RPE computations do not depend on local ACh activity, but this does not mean that ACh plays not role in modulating the amplitude RPE-like DA signals.

5) Specific methods and timelines for individual experiments are not described in sufficient detail. For instance, the optogenetic behavioral studies appeared to focus exclusively on testing the effect of side-port entry contingent light stimulation. There appears to be some variation in the duration of light delivery (0.5 – 2.5sec). Were there any systematic tests of light delivery at other time points, for example contingent on center port entry (during decision period)?

6) There is a heavy focus on the pauses in ACh that are anticorrelated with DA signals, and for good reason given the long-standing coverage of this topic and ongoing interest in the field. However, the glutamate-driven phasic ACh elevations during side-port entry and reward consumption reported here are also notable and somewhat more novel. They are not discussed in “Cholinergic contributions to behavior” section of the Discussion. The behavioral effects of the chronic perturbations of CINs or their glutamatergic inputs may shed some light of this, though a time-locked bidirectional optogenetic manipulation of CINs during the decision making window would help resolve the function of this important neuromodulatory system.

7) It may be useful to add clarification to the descriptions of training history (sequence) analyses. In particular, the photometry data for sequences appeared to focus on the second of the two trials (B for AB sequences), which was clear from the text (p. 5). However the conditional switch probabilities (e.g., Fig. ED 1G) appear to refer to behavior following each two-choice sequence history, but the wording was somewhat ambiguous.

8) In the methods “Behavior apparatus, training, and task” section, it ends with a paragraph explaining how after fiber implants mice were habituated to head-fixation procedures on a wheel. My understanding is that this was done in a couple of non-2ABT studies. Please clarify.

9) The exclusive focus on VLS is a limitation of the current study given that DA dynamics vary considerably across striatal subregions. A rationale for this focus is provided but does not address the issue of generalization. Likewise, a rationale for exclusive use of females is provided but still represents a limitation of the study.

Response to referees' comments with the authors' replies in blue

General response to the referees:

We thank the referees for their critiques and helpful suggestions. In this general response section, we address two concerns that were raised by all three referees. We provide responses to each referee's specific comments in the next section. In addition to the changes described here, we also added new datasets to the study and worked hard to improve the clarity of the presentation. We hope the referees and the editor now find the study ready for publication.

(i) Quantification and statistical analyses of the photometric recordings

In the original submission, in each figure panel, we presented the time course of average fluorescence transients across mice with the standard error of the mean at each time point. This allowed direct visual comparison of the separation of the means relative to the standard error of each signal. We did this because the transients, especially those reflecting changes in Ach, have complex and multi-peaked time courses such that we felt it was better to show the whole waveform as opposed to reducing the description of each transient to a single number (such as its peak or mean). In the extended data figures for the resubmission, we now include quantitative and statistical analyses for all photometry presented in the manuscript. We have updated the results and methods sections to explain these analyses. We present statistical analyses comparing conditions within each mouse as well as, when relevant, across genotypes. We performed two kinds of analyses:

First, we identified a single metric that best captures changes in DA and Ach dynamics across different trial types and reward outcomes. We performed analyses in the following periods: (i) prior to ("pre") side entry, corresponding to the time of transition from the center to side port when the animal's decision is made but before the reward delivery; (ii) immediately after ("post") side port entry, corresponding to the time of potential reward delivery and consumption. Each period was 0.918 seconds long, encompassing 17 data points. For DA signals, which typically had a single peak in each period, we calculated the mean of the Z-scored signal in each period (**Extended Data Fig. 4a**). To quantify Ach signals, which have more complex waveforms, we measured the difference between the maximum and minimum signal in each time window, which we call Δ Ach (**Extended Data Fig. 4b**). Due to the multiphasic dynamics of Ach transients, this metric better captures the differences in signals across conditions compared to the mean. We measure each metric in pairs of conditions for each mouse, which we represent as a connected pair of dots in each figure (**Extended Data Fig. 4a and 4b**). Open circles indicate that the differences between measurements is statistically significant ($p < 0.05$, two-sided t-test). Standard deviations are indicated by the error bars.

From this analysis, we capture clear quantitative differences across many condition pairs for both DA and Ach transients. In each mouse, reward outcome greatly alters DA and Ach transients, with unrewarded trials having consistently and significantly suppressed mean DA from baseline (**Extended Data Fig. 5a, top**) and significantly larger Δ Ach (**Extended Data Figure 5a, bottom**) relative to rewarded trials. Importantly, the changes are significant in post, but not pre, side entry period, consistent with the reward outcome only becoming known after the side port entry. Omission of the 2ABT LED cue significantly alters neurotransmitter dynamics across the entire duration of both rewarded and unrewarded trials (**Extended Data Fig. 5b**). Finally, the history of choice and reward outcome also modulates DA and Ach signals (**Extended Data Fig. 6**): selection of the same side port as in the previous trial modulates DA and Ach signals *post* side entry whereas selection of the opposite side as the previous trial impacts only DA signals *pre* side entry. This influence of history on Ach and DA responses is consistent with reward prediction error arising from the recent pattern of actions and rewards.

Second, given the dynamic and complex nature of our transients, we complemented this analysis with a method to evaluate the separability of signals across conditions using the full waveform in each time

window. We did this for two reasons. First, this allows us to use all points in the time series in the analysis window instead of a single summary metric like mean DA or Δ Ach. Second, because the animals perform many trials, sometimes a small difference in the mean of two signals can be highly significantly different but, because of large trial-to-trial variances about the mean, the differences cannot be used to identify the condition in which the transient arose (i.e., a statistical difference in means across conditions does not necessarily indicate that the conditions are reliably discriminable). Conversely, different waveforms may be differently shaped but have a similar summary statistic. Therefore, we performed linear discriminant analysis (LDA) using all 17 data points in each time window to measure the degree to which, trial-by-trial, the photometry signals could be discriminated across conditions. LDA is a supervised dimensionality reduction technique which projects two classes of data onto a new axis that minimizes the variance of each and maximizes the distance between their means (**Extended Data Fig. 4c**). We split the data into training (70% of trials) and test ('holdout', 30% of trials) sets. We generate an LDA classifier from the training set, which we then use to classify the trials in the test set. If there are robust differences between two groups, the LDA classifier will categorize each trial in the test set with high accuracy whereas if there are no differences it will perform poorly (~50%). In the figures, we denoted the % classification accuracy in parentheses next to each mouse identifier. For instance, reward outcome greatly alters DA and Ach transients, but only during the post side entry period (**Fig. 1d**). Consistent with these robust effects, the LDA classification accuracy for DA and Ach signals is very high when trained on rewarded versus unrewarded trials post (>80%) but not pre (~50-60%) side entry (**Extended Data Fig. 5a**).

(ii) The purpose, design, and interpretation of the experiment testing if optogenetic stimulation of Ach release impacts DA dynamics were confusing.

The purpose of these experiments was to determine whether exogenous activation of CINs is sufficient to evoke DA release during decision making. First, we validated that we could optogenetically evoke Ach release from Chrimson-expressing CINs in the VLS *in vivo* (**Extended Data Fig. 12a**, previously Extended Fig. 6). Next, we determined if optogenetic activation of CINs in mice performing the 2ABT increases DA levels. We coexpressed Chrimson in CINs and dLight1.1 in the VLS to simultaneously activate CINs and detect DA (**Extended Data Fig. 12b**, right hemisphere). To control for any off-target effects of laser stimulation (i.e., heat, light), we coexpressed mCherry in CINs and dLight1.1 in the other hemisphere of the same mouse (**Extended Data Fig. 12b**, left hemisphere). As a mouse performs the 2ABT, we record DA release and stimulate CINs in 25% of the trials interleaved throughout its session. This enables comparison of DA release during normal Ach release (no laser trials, gray) with DA release during elevated Ach release (laser trials, red) (**Extended Data Fig. 12e**). DA signals are aligned to side port entry and the laser stimulation period is denoted in the blue shaded region.

To make the data and their interpretation clearer, we have made the following changes to **Extended Data Fig. 12e**: (a) we plot a shorter and more relevant timeframe of the DA recordings (**left and middle panels**); (b) we quantify the effects of laser stimulation for each mouse by calculating the change in the maximum DA signal (for rewarded trials) or minimum DA signal (for unrewarded trials) during laser versus no laser trials (**right panel**).

In trials in which mCherry-expressing CINs are exposed to the Chrimson-activation light, DA release is comparable to when these CINs are not exposed to light (red and gray curves overlap during the laser stimulation period in blue, **left panel**). Thus, laser illumination of the VLS does not alter DA release. In contrast, activation of Chrimson-expressing CINs evoked a greater amount of DA release during the laser stimulation period (the red curve is above the gray curve during laser stimulation period, **middle panel**). We quantified the change in maximum DA signals during Chrimson stimulation and find that illuminating Chrimson-expressing CINs consistently and significantly increased DA signals compared to illuminating control CINs (**right panel**). Therefore, activation of CINs is sufficient to drive DA release in the VLS. However, this effect is small compared to what is observed *in vitro*. We now incorporate this in

the results section (lines 305-315). In the discussion, we speculate that this may be due to technical limitations of the number of CINs we can recruit with our optogenetic method.

Specific responses to each referee

Referee #1

The authors in this paper interrogate the contribution of CINs to striatal function. Despite the detailed understanding of the mechanisms of interactions between DA and CIN in vitro, how and when this crosstalk happens in vivo remains largely unknown. To understand the factors that shape Ach release during reward-based decision-making and how Ach release contributes to DA release, the authors examined striatal Ach and DA levels during a complex decision-making task in mice. The authors demonstrated that DA dynamics and RPE are independent of Ach release, but DA inhibits Ach levels in a D2-dependent manner. The authors then characterized the inputs for the cortex and thalamus and found that Glu released by both inputs is required for Ach release.

As a whole, I thought this was a fascinating manuscript that attempted to address a very ambitious question. The authors provided data that indicate that Ach release does not modulate the DA release dynamic in the striatum, while DA influences Ach signals via D2Rs. While some of the data are convincing, part of the analysis is confusing and challenging to comprehend. In particular, the behavioral analysis and the DA and Ach GLMs analysis are difficult to interpret, and the conclusions regarding the role of DA and Ach in action and reward history are not convincing. Several debates have been experimentally addressed in this manuscript, but the conclusions are often more confusing than the debates themselves. The contribution of thalamic and cortical inputs is, for example, puzzling, and the conclusions are not entirely clear.

We thank the referee for referring to the work as “fascinating” and for recognizing its “ambitious” nature. Thank you as well for the insightful input, which motivated many edits and rewrites. We hope that we have revised our manuscript in a way that addresses the concerns about data analysis, clarity, and interpretation. Below are our responses to the specific comments:

Suggestions:

The authors used a two-armed bandit task and measured DA and Ach signals with fiber photometry in the ventrolateral portion of the dorsal striatum. This figure (Extended figure 1) contains much interesting information about mouse performance and behavioral modeling, but it is hard to read and comprehend.

We appreciate the referee for pointing this out, and we have made the following changes to clarify the behavioral analyses presented in **Extended Figure 1**:

- We edited the main text to specify what the probability of switching ports ($p(\text{switch})$) and probability of choosing the high reward probability port ($p(\text{high port})$) mean (lines 82-88). We also changed the order of panels **b**, **c**, and **d** to match the order in which they are referenced in the text.
- The goal of panels **b** and **c** is to capture the temporal dynamics of behavioral changes triggered by the change in reward probability that occurs at the block transitions. To quantify differences across mice in $p(\text{switch})$ and the time course of $p(\text{high port})$, we used single value performance metrics of maximum $p(\text{switch})$ and the time constant of $p(\text{high port})$, τ_{highport} , respectively. A lower τ_{highport} means that the mouse more rapidly identifies the new highly rewarded port, and a higher maximum $p(\text{switch})$ indicates greater flexibility of behavior induced by the environmental

change. To aid in interpretation of these metrics, we indicate the maximum $p(\text{switch})$ with a red asterisk in panel **b** and we denote with a red dashed line in panel **c** the exponential fit of $p(\text{high port})$ from which τ is calculated. We explain the significance of these two metrics in the text (lines 82-88) and methods (lines 771-776).

- The goal of panel **e** is to illustrate the timing and sequence of behavioral events of mice performing the 2ABT. To make this more interpretable, we (i) replaced the abbreviations of the behavior events with the full terms, and (ii) included a timeline of the key behavioral events.
- We use a recursively-formulated logistic regression (RFLR) to model the observed switching behavior. The RFLR incorporates evidence about past choices and reward outcomes to accurately predict the mouse's behavior in the 2ABT, as evidenced by a strong overlap in the predicted and true dynamics of $p(\text{high port})$ and $p(\text{switch})$ in panel **h**. To clarify the use of the RFLR and the interpretation of the coefficients in panel **g**, we include an annotated equation and visual diagram of the RFLR in panel **f**. In addition, we edited our description and interpretation of the RFLR model in the main text (lines 96-107). Finally, we improved our description of the RFLR model in the methods section 'Behavior performance analysis' (3rd paragraph).
- To aid in interpretation of how choice and reward history influence the probability of switching on the current trial (panel **j**), we include a new table in panel **i** that summarizes what each history term used in panel **j** represents with regards to port choice and reward outcome.
- Finally, we improve the presentation of the RFLR by omitting panels that were not referenced until the Methods section and added to the confusion.

In Figure 1, quantitative comparisons between DA and Ach signals would be appropriate and clarify the message.

We now include quantitative analysis for all photometry in our manuscript. Please see "General Response to the Referees" above where we address this.

Furthermore, the choice and reward outcome history in figure 1, panels **f** and **d**, is quite confusing.

Thank you for pointing this out. We made the following changes:

- We moved the ipsilateral and contralateral analysis to **Extended Data Fig. 4d**. This allows the reader to focus only on the effects of reward outcome on DA and Ach signals. We examine the potential effects of lateralization within this Extended Figure.
- For panel **f**, we added a label to denote whether the port choices of the previous and current trials are the same (**Fig. 1f**) or different (**Fig. 1g**) and if the reward outcomes for the previous and current trials were "win" or "lose".

To assess the contribution of each behavioral variable to GLM performance, the authors performed a "leave-out analysis" of individual features and analyzed the correlation of Ach and DA signals during reward-based decision-making. The figures (extended figures 3, 4 and 5 as well as Figure 2) are confusing and do not exhaustively convey the conclusions.

To motivate and clarify the significance and use of the GLM for our study, we expanded this section. We also added a panel to illustrate how we assess GLM performance (**Extended Data Fig. 7b**). Please note that the previous Extended Figures 3-5 referenced by the referee are now Extended Figures 7-9.

We state:

"To evaluate the contribution of each behavioral event to DA and Ach dynamics, we developed a general linear model (GLM) to predict neuromodulator signals from behavior. The 2ABT is a complex behavior with multiple events that occur in quick succession and a GLM formally and quantitatively

determines what behavior components can explain the observed signal. We define a set of behavior variables (such as timing of licks, port entries, and rewards) that the GLM can use to explain the neural data (**Extended Data Fig. 7a**). For each variable, the model assigns a kernel, comprised of a time series of β coefficients, that represent the time-dependent contribution of each behavior variable to the photometry signal. Because of the small number of free parameters relative to the number of data points, the GLM can use an ordinary-least squares (OLS) cost function without regularization penalty terms (**Extended Data Fig. 8c**). To reconstruct the fluorescence transients, the kernels are convolved with the appropriate behavior events and summed. If the GLM works well, the reconstructed signal mimics the measured signal well, a property which we can measure as a mean squared error (MSE, which is the cost function minimized by the model) between the two signals and the R^2 value of the fit (**Extended Data Fig. 7b**). A higher MSE and a lower R^2 , judged on test data not used for model training, indicates a greater difference between the real and predicted signal and thus poorer GLM performance. Note that the GLM can fit the trial-averaged signal perfectly without fully explaining trial-to-trial variability across this mean.”

We test three different GLMs to find which one best reconstructs the observed DA and Ach signals. Importantly, because of the small (compared to the number of data points) set of predictor variables used, the models do not overfit the data (**Extended Data Fig. 8c**), allowing us forgo regularization. The simple ‘**base GLM**’ shown in **Extended Data Fig. 8a** and **8b** uses the behavior events shown in **Extended Data Fig. 1e** plus the reward outcome status to predict DA and Ach signals relatively well. However, the base GLM fails to capture the trial history dependence of the transients, which is not surprising as it was not given this information.

To improve the base GLM, we incorporated choice and reward history, as these significantly impact DA and Ach signals (**Fig. 1f** and **1g**). The ‘**history GLM**’ has increased accuracy, as measured by smaller MSE (**Extended Data Fig. 8d**, “+history”) and as seen in improved reconstructions of DA and Ach signals parsed by history (**Extended Fig. 9a** and **9b**).

In the final iteration of the GLM, we incorporated a photometry variable to examine if the DA signal helps predict the Ach signal and vice versa. Each photometry signal was insufficient to predict the other (**Extended Data Fig. 10h**), but when added to the history GLM, enhances its performance. We call this GLM version the ‘**history and photometry GLM**’.

With regards to the “leave-out analysis of individual features”, multiple behavioral events shape DA and Ach signals, and one of the challenges with modeling this relationship is accurately and specifically representing the appropriate set of behavioral features available to the model. Some behavior events may provide redundant information while others may be fully unnecessary to reconstruct the observed neural signals. To assess how each event contributes to DA and Ach signals, we performed a “leave out analysis” in which we omit one behavior variable at a time from the GLM and determine the degradation in GLM performance. The greater the unique contribution of a variable to a signal, the poorer the GLM will perform when the variable is omitted. We find that each behavior feature has distinct effects on the performance of the GLM for DA versus Ach. For instance, dropping information about the timing of the side port entry term (‘-SE’) impacts the Ach and DA GLMs, as seen by an increased MSE (**Extended Data Fig. 8d**); therefore, this behavior event influences both signals. Exclusion of side port exit (‘-SX’) and lick variables (‘-SL’) affect the GLM performance for Ach, but not DA. Altogether, this analysis reveals that DA and Ach transients in the VLS are shaped by different behavioral events.

In Figure 3, the authors assessed if Ach release is sufficient to modulate DA levels in vivo. The experiment presented in extended figure 6 in the head-fixed mouse is unclear. Furthermore, in this supplementary figure, there is an increase in signal in both mCherry and Chrimson in laser and no laser conditions.

The purpose of these experiments was to determine whether optogenetic activation of CINs is sufficient to evoke DA release during decision making. We improved the presentation of the experimental details, the results, and their interpretation (see “General response to the referees”, part (ii)).

The text that corresponds to figure 3d is confusing. Are the authors indeed measuring Ach or DA release with carbon fibers?

We apologize for the confusion. In **Fig. 3d**, we measured Ach release with fiber photometry *in vivo* in the context of the 2ABT behavior task. In contrast, in **Fig. 3c** we recorded DA release with carbon fibers in an *ex vivo* brain slice. To clarify this, we made the following changes (underlined) to specify these different experimental setups.

- In the main text (lines 319-323): Optogenetic activation of CINs in control *ex vivo* striatal slices evokes DA release as measured by carbon fiber amperometry; however, this is abolished if CINs express TelC (**Fig. 3c**). Furthermore, Ach transients recorded *in vivo* with fiber photometry during rewarded trials of the 2ABT are greatly suppressed by TelC expression in CINs compared to expression of a control protein, mCherry (**Fig. 3d; Extended Data Fig. 13a-d**).
- In the legend for **Fig. 3c**: “DA release as measured by carbon fiber amperometry in an acute striatal slice containing CINs coexpressing Chrimson with either a control protein mCherry (mCh) or TelC. Schematic of the experimental setup (top) and amperometry recordings aligned to laser stimulation (bottom).”

The difference in decision time between TelC and mCH control in extended figure 8 a is not convincing and difficult to interpret.

We had used bootstrapping to assess the significance of difference in parameters across conditions, which avoided making assumptions about the distribution from which the data are drawn. This was confusing and unnecessary as we were generally comparing means and standard errors. Therefore, we changed our statistical methods to more accurately assess the data analyzed, and we have updated the manuscript accordingly.

We now use a two-sided t-test to denote the significance of mean ITI, decision time, lose-switch rate, win-switch rate, and mean left rate. These metrics are means drawn from a distribution that approximates normal according to the central limit theorem. Indeed, as the reviewer suspected, the difference in decision time in Extended Figure 8 (now **Extended Data Fig. 18**) is not significant with this test.

We apply the Wilcoxon Rank Sum test to assess significances in τ , maximum p(switch) and RFLR coefficient values across conditions. These metrics arise from exponential curve fits (in the case of $\tau_{highport}$), single values measured across a time course (maximum p(switch)), and the fit of a logistic regression model (RFLR coefficients) such that using a t-test is not appropriate. Instead, we use the non-parametric Rank Sum test to quantify how likely these data are to be derived from the same population.

Finally, to evaluate the history-dependent conditional switch probabilities, we cannot make any assumptions about the underlying distribution of this data. Therefore, we use our previous analysis in which we generated bootstrapped distributions for the parameter in each experimental group, compare the difference in the means of these distributions, and determine whether the 95% confidence bounds of this difference overlaps with the null hypothesis that the means are the same.

It is difficult to understand the rationale of panel 3g and to comprehend the contrasting results between striatum versus VLS selective loss of Ach release.

The referee is referring to panels **3e** and **3f** for “striatum-wide versus VLS-selective loss of Ach release” and panel **3g**, in which we optogenetically inhibit DANs. We describe our rationale below, which we

incorporate into the results section entitled “Local Ach release from CINs is not necessary for DA dynamics during decision-making” (lines 304-356):

The DA signals we observe during the 2ABT could be driven by Ach release from CINs, DAN activity, or a combination of both. To determine if Ach release from CINs contributes to DA release, we blocked Ach release using CIN-specific tetanus toxin expression. Due to the large extent of DA axons, we reasoned that synchronized CIN activity across the striatum may be sufficient to drive DA release within the VLS where we record. Therefore, to test if Ach release is required for DA release, we first perturbed CINs in a striatum-wide manner by expressing TelC in CINs throughout the striatum using a multi-site injection approach (**Fig. 3e**). DA retained its capacity to encode reward outcomes, but the severe behavioral deficits that accompany this perturbation in Ach make it difficult to draw further conclusions.

Given that the proposed mechanism is local - CIN activity triggers DA release within a local DAN axon field - we tested whether DA release is impacted by VLS-selective inhibition of Ach release. In mice performing the 2ABT, we inhibited Ach release in the VLS of one hemisphere using TelC and compared DA release to that of the other hemisphere in which VLS CINs express a control protein (**Fig. 3f**). Surprisingly, this local manipulation did not impact DA dynamics.

Because local Ach release does not impact DA release, DAN activity is likely the major driver of DA dynamics, not CINs. To establish this, we inhibited DANs optogenetically in a separate cohort of mice performing the 2ABT and measured DA release (**Fig. 3g**). Indeed, inhibition of DAN activity robustly reduces DA signals in the VLS. Altogether, these results show that the DA release we observe in the VLS is predominantly due to the activity of DANs, not CINs.

Moreover, at the end of the paragraph on page 10, the authors write: “...modulation of CIN activity is sufficient to drive DA release *in vivo*”. The experiments that support this conclusion are not clear to me. The experiments showed in figure 3 indeed, indicate an absence of Ach regulation of DA dynamics.

The conclusion that optogenetic activation of “CIN activity is sufficient to drive DA release *in vivo*” was drawn from Extended Figure 6 (now **Extended Data Fig. 12**) in which we optogenetically activated CINs during the 2ABT and observed a subtle but significant increase in DA signals during rewarded trials (**Extended Data Fig. 12e**, middle panel). This gain of function experiment supports our statement that Ach release is *sufficient* to drive DA release *in vivo* (i.e., it can drive release). However, this does not mean that Ach release is *necessary* for DA release. The experiments in Figure 3 discussed above address this possibility by blocking endogenous Ach release. As the referee points out, these experiments indeed indicate an absence of Ach regulation of DA dynamics. Thus, while it is possible to elicit DA release by activating CINs, the physiological context in which this occurs remains to be determined.

In figure 4, the authors assessed whether optogenetic manipulation of DA neurons *in vivo* affects Ach levels and whether this effect depends upon D2Rs. Very cleverly, the authors have knock-out the D2s, specifically in CINs. Quantitative analysis is missing and the behavioral differences between groups in extended figure 10 are unclear and do not support the firm conclusion that D2Rs are required for a specific moment within a trial and the switch across multiple histories.

We have now included a quantitative analysis of the differences in DA and Ach release upon loss of D2Rs in **Extended Data Fig. 23a** (previously Extended Figure 10). We compare the mean DA signal during the pre and post side entry periods and do not find any significant differences across the three genotypes. In contrast, Ach signals are significantly increased in *Drd2*-cKO mice compared to the two control groups during both the pre and post side entry periods. For post side entry, we analyzed two time periods. First, we quantified the Δ Ach signal during a 0.918 second window (denoted as ii). There is a significant but small increase in the magnitude of change of the Ach signals in the *Drd2*-cKO mice. Given that the Ach signal in the *Drd2*-cKO mice only differs in the second, but not first peak, post side entry, we reanalyzed the signals during a more relevant time window (denoted as iii). Indeed, the

increase in the Ach signals of the *Drd2*-cKO mice post side entry are largely driven by the change in the second peak.

The referee references Extended Figure 10 (now **Extended Data Fig. 23**) to support the conclusion that “D2Rs are required for a specific moment within a trial”. However, we draw this conclusion based on the photometry data shown in **Fig. 4g**, which demonstrates that loss of D2Rs in CINs impairs Ach release during select moments in rewarded and unrewarded trials, specifically whenever DA signals increase. Thus, DA-mediated inhibition of Ach release is dynamic and contrasts with what is denoted in the circuit diagram (**Fig. 4a**), in which DA constitutively inhibits Ach release.

With regards to the conclusion that “D2Rs are required for...the switch across multiple histories”, this is supported by the behavior analysis in **Extended Data Fig. 23g**. Following certain choice and reward outcome histories, switching rates are reduced in D2R-KO CIN mice, with significant and consistent differences observed for ‘Aa’ and ‘ab’ sequences compared to both control groups. From this, we conclude that loss of D2Rs impairs normal switching behavior. How the specific disruption of Ach release observed during the photometry translates into this behavior effect will be of future interest to deduce.

We have rephrased our conclusions to better align them with the data (lines 405-408): “*Drd2*-cKO mice are impaired in their ability to switch selection ports across some histories when compared to both *Drd2*-ff and WT cohorts, with significantly reduced switching rates in ab and Aa sequences (**Extended Data Fig. 23g**). This supports a role for D2R-dependent reductions in Ach release in promote complex changes in behavior.”

In addition, we reframe our concluding statement in the text for clarity: “*In conclusion, we find that D2Rs are required for DA to repress Ach signals during precise moments within a trial, and loss of this regulation impairs the normal switching behavior of mice.*”

In figure 5, the authors examine the additional inputs required to stimulate Ach release. According to figure 5c, not only do the glu signals increase during consumption, but it is also the case during unrewarded trials that these signals are reduced. Is there a possible explanation for this?

Indeed, this is an interesting observation for which we do not have an explanation. However, glutamatergic inputs likely convey complex signals to striatum, and it appears that projections from cortex and thalamus may differentially contribute side entry and reward evoked CIN activation. Although beyond the scope of this study, it will be important to identify these inputs with retrograde labelling coupled with loss of function to assess their impact on glutamate release in the VLS. We include this in the discussion (lines 579-582):

“Identification of the upstream input(s) that alter the activities of cortex and thalamus to drive this differential release in glutamate will be key to understand what underlies important components of the decision-making process, such as action evaluation, learning, evidence accumulation, and behavioral policies.”

The behavioral effects perturbing the cortical and thalamic inputs into the VLS are puzzling, and based on the presented data, it is challenging to conclude regarding possible differences. Strong differences in the unrewarded trials can be observed in figure 5. How do the authors explain this finding? The results do not convincingly demonstrate the role of cortical and thalamic inputs in the interaction between Ach and DA.

In **Fig. 5**, we only interrogate contributions of cortical and thalamic inputs to Ach signals, *not to the interaction between Ach and DA*. This work was motivated by the finding that perturbations of DA signaling only affects some portions of Ach signals (see **Fig. 4g**), leaving open the question of which other inputs might contribute. Our experiments test the effect of cortical and thalamic neurotransmission on Ach release in the VLS, as both inputs project heavily into the VLS (**Fig. 5a**). We find that loss of

either input alone severely reduces the amplitude of Ach release across rewarded and unrewarded trials (**Fig. 5e and 5f**). This demonstrates that both the cortex and thalamus are required to drive Ach release.

When the reviewer states “the strong differences in the unrewarded trials can be observed in Figure 5”, we think they are referring to the fact that loss of the thalamic input has different effects on Ach dynamics (**Fig. 5f**) versus loss of the cortical input (**Fig. 5e**). Impairing the thalamostriatal input causes Ach signals to be temporally offset from control, whereas loss of the corticostriatal input reduces the second peak of Ach release. We speculate that this may arise for two reasons, which we include in the discussion. First, cortex and thalamus could drive Ach release in different ways. Future studies to interrogate glutamate release at cortical-CIN synapses versus thalamo-CIN synapses using a split glutamate sensor will be informative (such as that developed by the Oka lab: <https://doi.org/10.1101/2022.06.28.497868>). Second, we impaired the thalamic input more severely than the cortical input, which may contribute to the observed differences in Ach release. We only inhibited cortical cells that projected into the VLS as we used a retrograde AAV approach in a Vglut1-Cre line. In contrast, we inhibited a broad region of thalamus rather than only the region that projects to VLS. A retrograde AAV approach was not possible because the Vglut2-Cre line that we used to label thalamic cells also labels some cortical cells. Future studies in which we employ a more specific approach to perturb the glutamatergic inputs onto the CINs in VLS will clarify this situation.

The referee states that “the behavioral effects perturbing the cortical and thalamic inputs into the VLS are puzzling.” Tetanus expression in cortex or thalamus results in severely reduced glutamate release into the VLS, which in turn leads to severely reduced Ach release. The reduction in both Ach and glutamate release into the striatum results in pronounced behavioral defects in locomotion, reward rate, RFLR coefficients, etc. that are observed (**Extended Data Fig. 27**). A more precise loss of function approach that targets CIN-thalamic and CIN-cortex inputs may mitigate these widespread behavioral defects.

Referee #2

In their study, Chantranupong et al. characterized the release of dopamine (DA) and acetylcholine (Ach) in the ventrolateral striatum (VLS) during the decision phase of an operant behavior retrieval test. While it contains interesting observations, some of the conclusions are not directly supported by statistical analysis, and the manuscript is written in a very technical way, making it difficult to follow for a broad readership.

We thank the referee for their thoughtful input on our work. To address the technical aspects of the writing, we have edited the manuscript using clearer language. Moreover, we have added statistical analyses to support our conclusions. We hope that we have revised our manuscript in a way that addresses these concerns.

Specific Comments

First, the role of DA control on cholinergic interneurons (CIN) through CIN-specific depletion of the DA receptor 2 during a reward-based task had been shown previously, decreasing the novelty of the observations made in this study (see Martyniuk, K. M. et al. Dopamine D2Rs Coordinate Cue-Evoked Changes in Striatal Acetylcholine Levels. *Elife* 2022, cited in the manuscript). Thus, the rationale of this study as stated by the authors, is to provide a “more complete understanding” of DA and Ach release during reward decision making, which suggests their study belongs to a more specialized journal.

The referee notes that the study by Martyniuk et al. used the same strategy as we did to generate CIN specific deletion of D2Rs in a reward-based task. However, this does not impact the novelty of our observations for several reasons. First, the behavioral context is fundamentally different in our study and Martyniuk et al. Although both behavioral assays rely on reward-based decision making, our mice must do so in a probabilistic context with constant switches of reward probability. This enables us to

reveal effects of choice and reward history on DA and Ach signals that the task used by Martynuik et al. cannot examine.

Second, our findings differ in a crucial way from those obtained by Martynuik et al. We find that Ach pauses are abolished whenever they coincide with DA increases, whereas Martynuik et al. found that these pauses were reduced, not gone. Our findings are consistent with the proposed *in vitro* circuit in the field and are important to share with the neuroscience community, especially given the long-standing debate about the mechanisms that underlie Ach pauses *in vivo*.

Third, we observe different phases of Ach modulation in our study, notably those that are DA dependent and independent. Our findings reveal that Ach pauses during behavior are diverse, and they may have important contextual or functional dependencies. Our work reveals an important point that different mechanisms underlie what the field has collectively referred to as an Ach 'pause'.

Overall, our work is of interest to the broader neuroscience community as Ach pauses have long been observed but their origins remain unclear. Moreover, our work provides insights beyond just the impact of D2Rs on Ach release but addresses fundamental questions in the field as to what is encoded in Ach transients, whether Ach release influences DA release during behavior, and how cortical and thalamic inputs into striatum shape Ach dynamics.

Regarding the statistical analysis, although the authors used a sophisticated approach with a general linear model (GLM) to predict neuromodulator levels from behaviors, overall the study lacks statistical analysis to compare features of DA and Ach signals and their correlations in different conditions. For example, the authors claim that "reward outcome clearly modulates Ach transients". However, there is no statistical comparisons of signals parameters between rewarded and unrewarded trials (amplitude, time to peak, duration, AUC...).

We agree with the referee about the importance of statistical analysis to compare DA and Ach signals, and we have now included quantitative analysis for all photometry in our manuscript in the extended data figures. Please see "General Response to the Referees" above where we address this in part (i).

Similarly, Page 5 line 125 the authors claim the "magnitude of the Ach transient is increased during center-to-side transition and decreased during reward acquisition". The authors should provide a within animal statistical comparison of the amplitudes during the regular and omission trials to support this claim (for example as bar graph beside the trace). This comment applies to the entire manuscript, including to the lateralization analysis (ipsi/contra) which is critical to the entire manuscript.

We agree with the referee that a within animal comparison is crucial to demonstrate the robustness of our observations, and we have included this in the panels listed below. We perform statistical analyses as explained above in "General Response to the Referees". This analysis was not performed for the other panels as the recordings were done from separate mice and so these within animal comparisons are not valid.

- Lateralization of DA and Ach signals (**Extended Data Fig. 4e and 4f**)
- Reward outcome on DA and Ach signals (**Extended Data Fig. 5a**)
- Presence or absence of 2ABT cue (**Extended Data Fig. 5b**)
- Choice and reward history (**Extended Data Fig. 6a and 6b**)
- Magnitude of rDAh versus dLight1.1 signals (**Extended Data Fig. 10d**)
- Optogenetic activation of CINs and effect on DA signals (**Extended Data Fig. 12e**)
- Effect of tetanus expression in CINs on DA release (**Extended Data Fig. 16c and 16d**)

With regards to the lateralization of the signals, we recorded from mice that express DA and Ach in both hemispheres as a robust way to address this (**Extended Data Fig. 2e-h**). We compared the mean DA and Δ Ach after side port entry. For DA signals, there are no consistent differences across ipsi or

contra rewarded or unrewarded trials, and the LDA based classification is poor (**Extended Data Fig. 4e**). For Ach signals, some mice show statistically different transients for left- and right choice trials, but these are not systematically larger or smaller for ipsiversive versus contraversive movements (**Extended Data Fig. 4f**). Importantly, these differences, when present, are small and do not support high LDA classification accuracy of left versus right choice trials. This contrasts with the robust, consistent, and significant differences that are observed for rewarded versus unrewarded comparisons (**Extended Data Fig. 4e and 4f**, 'rewarded vs unrewarded').

In the abstract and manuscript the authors state that "DA modulates Ach levels only at specific times". To confirm this claim, the authors performed a correlation between DA and Ach signals around the side port entry (SE) that they compare to correlation of shuffled data. However, a more rigorous approach would be to select random time points outside of the trials (as many time points as the number of trials over the entire session) to compare with the correlations around the side port entry (Figure 2c). The explanation of the shuffling method is unclear (Page 21 Line 705) and does not inform about the correlation out of the SE trials.

Our statement that "DA modulates Ach levels only at specific times" is not based on the correlation between DA and Ach signals around the side port entry time (**Fig. 2c**) as the referee notes. Instead, it is based on the following data. We incorporate these comments into the discussion (lines 531-545) for clarity.

First, DA inhibits Ach signals in a D2R-dependent manner *only at specific time points during a trial* - when DA is transiently elevated (**Fig. 4g**). This inhibition is not constitutive as implied in the circuit diagram. Without this D2R-dependent regulation, the anticorrelation that exists between DA and Ach signals is significantly reduced (**Fig. 4h**).

Second, the anticorrelation between DA and Ach signals can be *disrupted at specific times* by mechanisms that remain to be determined. We performed a covariance analysis in which we calculate how the variance about the mean of DA at one time point influences the variance about the mean of the Ach signal at another time point (**Extended Data Fig. 10g**). This analysis reveals trial-by-trial variance in which DA and Ach signals are dominated by strong negative correlation (**Fig. 2e**, see blue diagonal line). However, at a specific moment immediately after side port entry, this is transiently disrupted (**Fig. 2e**, see inset). We speculate that additional inputs triggered by side port entry may override this underlying anticorrelation between DA and Ach. Taken together, these results demonstrate that transient DA release at specific moments within a trial inhibits Ach, which contributes to an anticorrelated relationship between DA and Ach. This anticorrelation can in turn be momentarily disrupted via unknown mechanisms.

The referee suggested that we select random time points outside the trial to compare the correlations around the side port entry. We think that there is a misunderstanding due to our incomplete labeling of **Fig. 2c**, and we apologize for this confusion. The shuffling in **Fig. 2c** (green trace) is not done point by point (i.e., randomizing the order of the time points), but rather by calculating the covariance of the Ach signal in one session with the DA signal from a different session. This approach preserves the statistics and autocorrelation structure of each signal. These signals are not segregated by trial identity or structure; thus, the correlations are not aligned to side port entry as the referee states, but rather a time shift of zero, which we now denote.

As an alternative method to test for spurious correlations, we introduce a new shuffling method in which we offset the DA fluorescence transient by a random number of time steps (between 1 and 10000) and calculate the cross-correlation between the original Ach transient and the shifted DA transient. As is clear (**Fig. 2c**, orange trace), misaligning the trials in this way destroys the correlation between the signals. We have added a descriptor for the axes of **Fig. 2c** and included a clearer explanation for how we performed this shuffling in the methods section (lines 813-826).

From my understanding the ‘noise analysis’ might correspond to the needed analysis (Figure 2d). The authors analyzed DA and Ach signals co-variation which revealed a sequence of positive/negative/positive co-variation. However, this co-variation appears to be similar in the ‘noise’ and the SE trials. Here as well, statistical comparisons of the co-variations for noise and SE trials should be performed. Finally, as the “noise” is recorded signals and not artefacts ‘noise’ should be renamed “ITI signal” (inter-trial interval signal).

We apologize for the lack of clarity concerning the ‘noise analysis’ in **Fig. 2d** – it is not derived from the inter-trial interval signal. We have made changes in the figures and text to clarify this (lines 266-276 (results); lines 827-833 (methods)), and we include a description below.

A response can be separated into two components: signal (average response to each stimulus) and noise (deviation of each response from the average, i.e., a measure of trial-to-trial variability), which we now illustrate for clarity in **Extended Data Fig. 10e**. Cross correlation allows one to quantify how similar are the dynamics of two components. Determining the relationship of the noise of two signals can reveal what drives their dynamics. If the noise of two responses is correlated, it suggests that either there is a common input - direct or indirect - that drives these signals, or that one signal regulates the other. However, if the noise lacks correlation, these responses may be shaped by intrinsic neuron properties, segregated inputs, or different temporal filtering.

Based on these principles, in the co-variance analysis of **Fig. 2d**, we first take the mean of all trial-segregated signals (center port entry until side port exit). We perform cross correlation on the means by fixing one signal, shifting the other signal, and calculating their covariance. Next, we perform the same analysis on the noise. We find strong anticorrelation in both the signal and noise components (**Fig. 2d**). This suggests that DA and Ach signals may interact directly or indirectly.

As the referee suggested, we have done statistical comparisons of cross correlations and quantified two key features of the minimum cross correlation: its magnitude and its time shift from side port entry (**Extended Data Fig. 10f**). We find that there is a consistent ~100 msec positive time shift which is not statistically significantly different across reward outcome or across signal and noise components.

Again, regarding quantification of the results, Page 7 line 205 the author claim “DA lags Ach reveals an anti-correlation that occurs with a positive time lag (Extended Data Fig. 5b)”. While potentially interesting the authors should provide quantification of this lag and statistical comparisons of it during trials and ITI. Similarly on Page 8 Line 230, the authors mention “100 ms lag”, however, it is unclear from where this value has been extracted (Figure 2f is cited).

We agree with the referee that the “100 ms lag” we mention is difficult to visualize. For Extended Data Fig. 5b, which is now **Extended Data Fig. 10b**, we now highlight this 100 ms lag with an enlarged inset that makes this feature clearer. We also do the same for the correlation analysis in **Fig. 2c** and **2d**. We now quantify the time offset and magnitude of this minimum cross correlation across rewarded and unrewarded trial outcomes for both signal and noise components of the response (**Extended Data Fig. 10f**).

When we cite a 100 ms time lag in the covariance analysis of **Fig. 2f**, we refer to the negative correlation (strong blue signal) that is shifted from the diagonal and delayed in the positive direction. We have overlaid a dashed line in **Fig. 2f** that more clearly indicates the diagonal and makes this offset clearer. Mathematically, the average of this off-diagonal signal yields the minimum anticorrelation signal in **Fig. 2d**, which we previously quantified has a ~100 ms time lag (**Extended Data Fig. 10f**, left plot). We illustrate this concept in the schematic of **Extended Data Fig. 10g**. For clarity, we modify the text (lines 281-285) and reference **Extended Data Fig. 10g** to aid in this interpretation:

*“This results in a 2-dimensional function – $K(t_1, t_2)$ – that describes the relationships between fluctuations in DA and Ach at specific times, such as entry into the side port. This revealed a strong time-lagged negative covariance (**Fig. 2e**, strong blue signal), which we call the off-diagonal (**Fig. 2f**)*

showing that, at most time points, changes in DA precede changes in Ach by approximately 100 ms (Extended Data Fig. 10g), which is consistent with prior analysis (Extended Data Fig. 10f)."

Finally, the study should include at least one representative high-resolution confocal image of neurons (or terminals) located under the optic fiber and expressing each of the tools used in the study. Similarly, at least one representative image of a brain section showing the spread of viral expression should be included for each tool. The location of optic fiber implants of each animal should also be represented on an atlas drawing for each experimental group to evaluate the potential variability of the recording locations.

We agree with the referee that histology and fiber implant locations are crucial for interpretation of the results. For all photometry recordings in our study, we have included a panel in the Extended Data Figures (i.e., **Extended Data Fig. 2**) that show the following: (a) multiple slices spanning the anterior to posterior axis that show the spread of the viral expression from a representative animal, (b) an inset with a high resolution image of the neurons underneath the optic fiber, with the fiber tract outlined, and (c) a schematic marking the fiber placement for each animal within an experimental group, denoted with a pink dot.

Minor Comments

As there is no difference in recordings of DA and Ach signals from the ipsi- and contralateral side of the port entry, the authors could move this part of the analysis in the supplementary Figure, and bring in some of the behavioral analysis of Extended Figure 1 in the main Figure 1.

We moved the ipsi- and contralateral analysis to **Extended Data Fig. 4e** and **4f**. We thank the referee for their suggestion to split the behavior analysis, but we decided to keep this in **Extended Data Fig. 1** as it is easier to interpret the behavior analysis in one consolidated figure instead of divided amongst two figures.

The diagram in Figure 1a is confusing, as it suggests that nAChR reached by the CIN Ach are located in the VTA – DAN should be replaced with DAN-terminals, and VL striatum should be written on top of the diagram.

We have made these changes in **Fig. 1a**.

Figure 1e: please replace +LE and – LED with cue(+) and cue(omitted)

We have made these changes in **Fig. 1e**.

Page 5 line 123: replace 'more unexpected' with 'less expected'

We have made the change.

The representation of the AA, aA, Aa and aa cases is confusing (Figure 1f-g). First, each case should have a different color (gradients of the same color for each neurotransmitter?) and be represented in 4 different graphs on the top of each other. Indicating the full legend on the Figure would also be helpful: AA = 2 rewarded trials on the same side, aA: unrewarded trial followed by a rewarded one on the same side...

We modified **Fig. 1f** and **1g** to improve its clarity. Instead of representing the trial outcome and reward histories using a double letter code, we replaced this information with text to make it more straightforward to interpret. First, we added a label that denotes whether the port choices of the previous and current trial are the same (**Fig. 1f**) or switched (**Fig. 1g**). Second, we denote the reward outcomes of the previous and current trial using "win" and "lose" terms, with different colors to highlight the outcome of the previous trial.

We appreciate the referee's suggestions to alter our plots. We assigned gradients of the same color to each neurotransmitter, but we found that the plots are clearer when we use different and high contrast colors that represent the previous trial's reward outcome. We also overlaid four different plots on a single graph as the referee suggested, but we found that this the plot was too cluttered and obscured the differences we wanted to highlight; therefore, for clarity we overlay two signals per graph.

Figure 2a could be moved to the extended data, and needs statistical comparison to validate the absence of differences between dLight1.1 and rDAh signals.

We do not claim that there is an absence of differences between dLight1.1 and rDAh signals. While the timing of the peaks and dips of the two sensor signals are comparable, there are key differences in their amplitudes (**Fig. 2a**). We performed a statistical analysis in which we compare the maximum or minimum DA signal for rewarded or unrewarded trials, respectively (**Extended Data Fig. 10d**). Across all mice, the magnitude of the rDAh response is lower than dLight1.1. This likely reflects the higher affinity of rDAh for DA, which leads to quicker saturation or slower kinetics of the signal. We edited the main text (lines 262-265) to make these points clear:

*“Both sensors yield comparable signals, but with consistently reduced amplitudes for rDAh (**Fig. 2a**), likely reflecting its slower kinetics and higher affinity for DA compared to dLight1.1.”*

Figure 5g: the authors locked the signals to the 'decision' of the animal. However, it is not possible to know at a millisecond timescale when the decision occurs. The decision might be taken anytime between trial onset and the movement onset toward the delivery port. As stated in the legend, the line represent trial onset and should be labeled as such.

We agree with the referee, and we have changed the label to “side port entry”.

Regarding mice, as they were water restricted to 1 mL per day, this suggest animas were single housed. This should be specified. The fact that this study is limited to male mice is reducing the impact of the findings and should be discussed. Moreover, the arguments provided by the authors in the methods suggesting potentially more variability in females due to the estrous cycle is not always valid as testosterone levels and social rank in males have also been show to heavily influence mouse behavior and increase inter-individual variability.

We include the detail that our behavior mice are singly housed in the methods (line 694). We agree with the referee about the potential influence of testosterone and social rank on male mouse behavior and have omitted this comment in the manuscript. Our motivation to use males is based on two lines of reasoning. First, estrogen is known to modulate cholinergic signaling in the rodent brain. Levels of choline acetyltransferase mRNA and protein as well as choline uptake is altered by estrogen. Second, male, not female, behavior is impacted by perturbation of muscarinic Ach receptors on DANs in the ventral tegmental area, suggesting gender-dependent differences in Ach signaling within the striatum. As a result, we decided to perform our studies exclusively in males, but we note in the discussion that that future studies in female mice will be informative (lines 561-566):

“Finally, in our study we exclusively used males due to recent findings that male, but not female, exploratory behavior is affected by loss of muscarinic Ach receptors on DANs in the ventral tegmental area⁷⁰. In addition, estrogen regulates cholinergic signaling in rodents, through effects on choline acetyltransferase expression, Ach release, and choline transport⁷¹⁻⁷⁶. Therefore, future studies to determine if our results are generalizable to female mice will be informative.”

Page 13 Line 427: DA and Ach are critical neurotransmitters that directly modulate each other's release in vitro *in the striatum*.

We have made the change.

Page 15 Line 483: it is unclear what the authors mean by cholinergic Ach

Thank you. This was a typo. We have changed “cholinergic Ach” to “Ach”.

Referee #3

A large body of mostly *ex vivo* work has led to the hypothesis that striatal cholinergic interneurons (CINs) can facilitate local dopamine release by activating nicotinic receptors on dopamine neuron (DAN) axons, and that local dopamine release can negatively regulate CIN activity (and thereby acetylcholine (ACh) release) by activating D2 DA receptors on CINs. The current study by Chantranupong et al. examines whether these putative interactions occur *in vivo* within the ventrolateral striatum (VLS) of mice performing a reward motivated decision making task (two-armed bandit task; 2ABT). Fiber photometry was used to record VLS DA and ACh signals (via genetically-encoded DA and ACh sensors) during task performance, which revealed distinct multiphasic DA and ACh responses. Computational modeling revealed that momentary changes in DA and ACh release depended on important task events (response initiation and reward receipt) and recent training history. Consistent with previous research, surges in DA occurred during response (side port entry) initiation and reward retrieval, which were temporally associated with dips in ACh. Bidirectional optogenetic manipulations of VLS DA terminals confirmed that there is substantial D2 receptor-mediated negative regulatory influence over local ACh release *in vivo*. Moreover, genetic deletion of D2 receptors in CINs abolished DA-induced inhibition of CIN activity and disrupted the apparent inhibitory influence of DA on task-related ACh release. Two other components of task-related ACh release (surges in ACh during response execution and reward retrieval/consumption) were shown to be driven by glutamatergic inputs from cortex and thalamus, addressing an important gap in knowledge. A separate series of experiments examined the role CINs in regulating DA release. These results were somewhat less compelling, but were nevertheless taken as evidence that *in vivo* DA release dynamics do not require ACh release by CINs.

This study provides important new information about how these two important neuromodulatory systems interact in the VLS during reward-related decision making. The methods are generally rigorous and well-controlled, and take advantage of cutting-edge and sophisticated techniques and analyses. The results are likely to be of interest to a broad audience primarily within the neuroscience community. The authors do a good job of discussing some of the limitations of their findings and identifying areas for future research. I have identified a few other limitations of the current study below

Thank you for the kind words, for the accurate summary, and for referring to our work as “well-controlled”, “cutting-edge” and “important”. We thank the referee for their insights into our work and for suggesting that it will be of broad interest to the neuroscience community. We hope that we have addressed the limitations of our study that the referee has pointed out.

1) Statistical support is not reported for many of the main conclusions about photometric data, despite claims about significance. This is likely a trivial matter in the majority of cases, but the lack of a consistent and clear approach for discussing findings may obscure some effects.

We agree with the referee about the importance of statistical analysis to compare DA and ACh signals, and we have now included quantitative analysis for all photometry in our manuscript in the extended data figures. Please see “General Response to the Referees” above where we address this in part (i).

2) In particular, the authors conclude that there ACh release by CINs exerts little if any influence over DA release, or task-related DA dynamics. There is some recognition in the text that optogenetic CIN stimulation augmented DA release during reward delivery (Figure ED 6C), but this result is not indicated in the figure.

The other referees also had this concern. We clarify this experiment, the results, and their interpretation above in the “General response to the referees”, part (ii).

3) Likewise, the effects of TeIC expression in CINs on task-related DA release (Fig 3e and ED 7b,c) are discussed in broad terms without statistical support. The authors acknowledge that their TeIC expression was associated with larger amplitude DA signals, but then dismissed this as potentially reflecting variation in autofluorescence or sensor expression. This leaves the question somewhat open. While the z-score data suggest that the basic features of task-related DA dynamics were preserved in this condition, this does not necessarily address whether CINs regulate the absolute magnitude of task-related DA release. Given that the authors are attempting to conclude that there is little ACh modulation of DA, a more in depth analysis of the data by training history may be warranted.

To provide support for our statement that autofluorescence and sensor expression levels contribute to the variability in absolute dLight1.1 signals, we recorded from three mice which express dLight1.1 at the same titer in both hemispheres of the VLS. As is clear below, there is large variability in the amplitudes of the $\Delta F/F_0$ DA signal. Given this underlying variability, absolute values of DA release are difficult to interpret unless the effects are very robust as in the cortical and thalamic inhibition experiments of Fig. 5. Please note that for these reasons we do not use $\Delta F/F_0$ to quantify changes in fluorescence in the text but instead rely on Z-scored transients.

As requested by the referee, we performed a more in-depth analysis of the data based on training history. We expressed dLight1.1 in hemispheres in which CINs express tetanus toxin or a control protein (**Extended Data Fig. 15a-c, 16e-f**). We parsed the DA signals by choice and reward outcome histories and find that the mean DA signals from the control and tetanus hemispheres are similar across multiple histories of all mice (**Extended Data Fig. 15d, 16b**), which we quantified (**Extended Data Fig. 16d**). To complement this statistical analysis, we fit signals from the tetanus and control hemispheres with a history GLM and found that the shape of the kernels for DA signals from both conditions are similar (**Extended Data Fig. 15e**). Altogether, these results suggest that loss of ACh release in the VLS does not impair the ability of DA signals to be modulated by reward outcome and choice history. We have updated the manuscript with these findings.

However, when the referee requested “a more in depth analysis of the data by training history”, they may have referred to training history with respect to behavior. In the 2ABT task, mice learn at variable rates (two to four weeks to become proficient). Therefore, it is not an ideal behavior task to assess training history or learning rates. A different task with a less bespoke training procedure will be necessary for such analysis in the future.

4) Moreover, there is clearer evidence that striatum-wide CIN TeIC expression dysregulated DA dynamics (Fig 3f, ED 7e), though again these effects are not explicitly indicated or highlighted. The authors conclude that because this broader manipulation of CIN function led to behavioral alterations, it

is difficult to interpret changes DA signals. I agree that this does pose some challenges, but the photometry data are separated into epochs based on specific behavioral events, which helps control for behavioral variation.

We agree with the referee that it would be informative to separate the data into epochs based on specific behavioral events to control for behavioral variation. To address this point, we performed GLM analysis on our data in **Fig. 3e** (formerly Fig. 3f). A GLM allows one to precisely quantify the contribution of each behavioral event to the photometry signal and to dissociate the effects of changes in the timing of behavioral events due to tetanus toxin expression. There are two outcomes: first, the timing of behavioral events is altered by tetanus expression, but the underlying DA signal associated with each behavioral event is unchanged. In this case, the DA kernels from the tetanus condition will be similar to the control. Alternately, both the timing of events and the relationships between events and release of DA change. In this case, the shape of the DA kernels from the tetanus condition will be different than those of the control. We find that the latter is true (**Extended Data Fig. 14e**), and that there are widespread effects on both the timing and release of DA signals associated with all underlying behavioral events. Nevertheless, given the severe changes in the patterns and timing of task-related behaviors and the complex dependence of DA signaling on task history, it is difficult to conclude whether there is a quantitative change in the RPE encoding of DA signals. We note this in the results section (lines 330-335), copied below:

*“The severe behavioral changes underscore the importance of CINs in regulating striatal function; however, they obscure interpretation of the effects of Ach loss on the reward-encoding properties of DA. Nevertheless, DA retained its capacity to encode for reward, such that DA signals (**Fig. 3e**) and their associated GLM kernels (**Extended Data Fig. 14e**) maintain opposing polarity with reward outcome. Thus, reward encoding features of DA can persist in the absence of Ach.”*

The authors also note that the reward-prediction error (RPE) like DA signals (increase during reward and dip following absence of reward) was “sustained” following system-wide ACh depletion, but this statement deserves some scrutiny. Was there a quantitative change in the RPE-like signal, as appears to be the case? The data do suggest that RPE computations do not depend on local ACh activity, but this does not mean that ACh plays no role in modulating the amplitude RPE-like DA signals.

We agree with the reviewer that our statement on the changes in RPE signals should be modified. We have changed our wording of RPE being “sustained” to DA “retained the capacity to encode for reward” (lines 332-333) as this is more accurate with our data. We also note in our discussion, copied below, that we cannot conclude that ACh does not modulate the RPE signal encoded by DA. We suggest key experiments in the future to help discern this, specifically via deletion of $\beta 2$ receptors in DANs.

“However, given the severe perturbations on behavior, Ach could still play a role in modulating RPE encoding of DA. Nevertheless, specific deletion of muscarinic and nicotinic receptors in DA neurons are necessary to reveal if Ach can regulate other aspects of DA release.”

5) Specific methods and timelines for individual experiments are not described in sufficient detail. For instance, the optogenetic behavioral studies appeared to focus exclusively on testing the effect of side-port entry contingent light stimulation. There appears to be some variation in the duration of light delivery (0.5 – 2.5sec). Were there any systematic tests of light delivery at other time points, for example contingent on center port entry (during decision period)?

The rationale for the optogenetic stimulation time is described below. We have included these details in the methods to provide clarity.

For the optogenetic activation of CINs (**Extended Data Fig. 12e**), a duration of 1.5 seconds was used. In these experiments, laser stimulation was triggered by side port entry for a set duration. If the laser stimulation time is set for too long, it will extend beyond the end of the trial and lead to ectopic effects in the next trial. Conversely, if the laser stimulation time is too short, it may be insufficient to perturb DA

signals. We chose a stimulation time of 1.5 seconds because it was shorter than the average side port occupancy time of the mice in this experimental cohort. We state this in the methods (lines 739-741): *“We used a stimulation duration that would not persist past the side port entry and introduce ectopic effects on the next trial.”*

For the optogenetic inhibition of DANs in **Fig. 3g**, we used a laser onset time of 5 seconds. In this cohort, the DAT-Cre mice had very long consumption times. This may be a result of the genotype as we did not observe such long consumption times for ChAT-Cre and wild type mice. To account for this, we extended the stimulation period. For these experiments, we performed systematic tests of light delivery upon center port entry. However, this led to a drastic decline in performance such that we were unable to collect sufficient trials to record DA release. Thus, we only include side port inhibition in **Fig. 3g** for which the impact on behavioral performance was less severe.

6) There is a heavy focus on the pauses in ACh that are anticorrelated with DA signals, and for good reason given the long-standing coverage of this topic and ongoing interest in the field. However, the glutamate-driven phasic ACh elevations during side-port entry and reward consumption reported here are also notable and somewhat more novel. They are not discussed in “Cholinergic contributions to behavior” section of the Discussion.

We expanded upon the discussion of glutamate-driven phasic Ach release in the section, now titled “Contributions of Ach release to behavior”. Transient Ach elevations during side port entry and reward consumption suggest a role for this neurotransmitter in regulating these key behavior events. Ach release has long been known to be associated with movement, and Ach release in the 2ABT may directly drive key movements of mice into the side port and their sustained licking bouts during reward consumption. Alternately, Ach release may reflect changes in the salience associated with these behavior events. Future experiments to precisely manipulate different periods of Ach release coupled with high-resolution behavior tracking will reveal how Ach modulates these behaviors.

We are also interested to uncover the mechanisms that sustain glutamate release from the cortex and thalamus throughout the consumption period of rewarded trials, and what in turn causes glutamate release to drop in unrewarded trials. Identification of the upstream input(s) that alter the activities of cortex and thalamus to drive this differential release in glutamate will be key to understand what underlies important components of the decision-making process, including action evaluation, learning, evidence accumulation, and behavioral policies.

Finally, defining if and how the varying dynamics of DA, Ach, and glutamate during a trial modulates striatal plasticity will be of great interest. *In vivo* work demonstrated that the coincidence of a DA peak and an Ach dip is necessary to induce long term potentiation in corticostriatal synapses, as glutamate release that occurs during this period will depolarize SPNs. However, during the 2ABT, we do not observe temporal patterns of neurotransmitter release that would be permissive for SPN plasticity. When Ach dips in a D2R-dependent manner, glutamate levels are concurrently reduced by unknown mechanisms. This suggests that SPN plasticity may not be modulated once an animal becomes proficient in the task, and instead, the main role of the striatum may be to execute the actions in a manner modelled by the RFLR or an alternative algorithm. It will be crucial to determine the dynamics of glutamate, DA, and Ach release across learning, during which striatal plasticity is likely to be key. Altogether, these studies will lead to a greater understanding of how neurotransmitter release is precisely coordinated in the striatum to regulate behavior.

The behavioral effects of the chronic perturbations of CINs or their glutamatergic inputs may shed some light of this, though a time-locked bidirectional optogenetic manipulation of CINs during the decision making window would help resolve the function of this important neuromodulatory system.

We agree that bidirectional optogenetic manipulation would help clarify the impact of this neuromodulatory system on behavior. We first tested the effects of optogenetic activation of CINs, in

which we observed a subtle but consistent increase in DA during unrewarded trials (**Extended Data Fig. 12e**). To determine if this DA increase led to behavior effects, we analyzed the effects on side port occupancy time and decision time on trials with and without CIN activation (**Extended Data Fig. 12f**). We did not observe any consistent changes in behavior, which is not surprising given the subtle increase in DA release.

Conversely, we performed optogenetic inhibition of CINs using the opsin Jaws. Below we show the Ach signals recorded when Jaws-expressing CINs are activated by laser stimulation. Due to the technical limitations of our optogenetic set-up, the laser stimulation artifacts obscure the dLight1.1 signal recorded in the green channel. Therefore, we inhibited CINs in a pulse-like manner to visualize dLight1.1 signal during the off periods of the laser. However, we struggled with strong rebound excitation in the CINs that occurs once the inhibition period is over (see below, yellow panel, left) which has been previously reported (Zucca et. al; PMID: 29578407). Attempts to ramp down the laser stimulation power or to use minimal pulse patterns did not mitigate this. This rebound excitation makes it difficult to interpret any effects of CIN inhibition on DA signals and behavior. We did not pursue these experiments further as tetanus toxin is a much cleaner loss of function tool.

7) It may be useful to add clarification to the descriptions of training history (sequence) analyses. In particular, the photometry data for sequences appeared to focus on the second of the two trials (B for AB sequences), which was clear from the text (p. 5). However the conditional switch probabilities (e.g., Fig. ED 1G) appear to refer to behavior following each two-choice sequence history, but the wording was somewhat ambiguous.

We thank the referee for pointing this out. We amended the text and the axes labels (**Extended Data Fig. 1j**) to make it clear that the conditional switch probability $p(\text{switch})$ refers to the current choice, and this is impacted by the history, which refers to the choice and reward outcomes of the two previous trials, which we now illustrate for clarity (**Extended Data Fig. 1i**). In addition, we omit the use of the two-letter sequence for the photometry data in **Fig. 1f** and **1g** and replace them with clearer text descriptions.

8) In the methods “Behavior apparatus, training, and task” section, it ends with a paragraph explaining how after fiber implants mice were habituated to head-fixation procedures on a wheel. My understanding is that this was done in a couple of non-2ABT studies. Please clarify.

Habituation to head-fixation on a wheel was done for all recorded mice that performed the 2ABT. The purpose is to restrain the mouse’s head to enable secure and consistent attachment of the fiber optic patchcord to the optical fiber implant in the brain. Habituation is necessary to reduce the stress of the

mice to this head-fixation step and to ensure proficient performance in the 2ABT during photometric recordings. We added an explanation in the methods to clarify this (lines 715-718).

9) The exclusive focus on VLS is a limitation of the current study given that DA dynamics vary considerably across striatal subregions. A rationale for this focus is provided but does not address the issue of generalization. Likewise, a rationale for exclusive use of females is provided but still represents a limitation of the study.

We agree with the reviewer that it remains to be determined whether our results are generalizable, given the heterogeneity of the striatum. This will be best addressed by repeating these experiments in other striatal regions. We have addressed this issue of generalization in the discussion (lines 559-561):

“While our study focused on the VLS, given the spatial and functional heterogeneity of striatum, a comprehensive survey of these inputs across different striatal regions is necessary to resolve different conclusions across past studies.”

We also agree about exclusive use of males. We think the reviewer meant males and not females in their comment, and we have expanded upon this in the discussion. Our motivation to use males in this study is based on two lines of reasoning. First, estrogen is known to modulate cholinergic signaling in the rodent brain. Levels of choline acetyltransferase mRNA and protein as well as choline uptake is altered by estrogen. Second, male, not female, behavior is impacted by perturbation of muscarinic Ach receptors on DANs in the ventral tegmental area, suggesting gender-dependent differences in Ach signaling within the striatum. As a result, we decided to perform our studies exclusively in males, but we note in the discussion that that future studies that compare Ach signaling in female versus male mice will be meaningful (lines 561-566):

“Moreover, in our study we exclusively used males due to recent findings that male, but not female, exploratory behavior is affected by loss of muscarinic Ach receptors on DANs in the ventral tegmental area⁷⁰. In addition, estrogen regulates cholinergic signaling in rodents, through effects on choline acetyltransferase expression, acetylcholine release, and choline transport⁷¹⁻⁷⁶. Therefore, future studies to determine if our results are generalizable to female mice will be informative.”

Reviewer Reports on the First Revision:

Referees' comments:

Referee #1 (Remarks to the Author):

The authors have replayed my questions and the manuscript is now clearer. I congratulate them for this fantastic work

Referee #2 (Remarks to the Author):

The revised manuscript contains important quantifications of the data presented in the study, which significantly strengthen the conclusions of the paper. The authors also made an effort in rewriting the manuscript to make it more accessible. However, some of the points were only partially addressed, and some methodological aspects remain unclear. These need to be tackled before the manuscript can be published.

Specific comments _____

The quantification presented in Extended Figure 4a and b is an important addition, but should be included in the main Figure 1, especially as this Figure includes unused white space. Moreover, it is unclear why the authors did not perform the average or max-min analysis for both DA and Ach. Uniformizing both analyses with min-max would make the results comparable across neurotransmissions, and increase accessibility of the results and interpretations.

Inclusion of the justification why experiments were performed only in males is important and need to be included in the rational of the study (i.e., in the introduction).

The effort of the authors to include multiple images of a representative mouse for each mice in the Extended data Figures is highly valuable, and reflects the quality of the stereotaxic surgeries of the study. However, the insets still do not depict the plasma membrane expression of the sensors. The authors need to provide at least one high magnification (confocal) image of neurons expressing each of the sensors (green and red sensors for DA and ACh), even if the background signal is high (ideally this image should be included in the main Figure). Similarly, in Figure 3b, it is unclear what the arrows are pointing at (not defined in the legend) and epifluorescence images are blurry and do not allow to interpret the image.

As previous studies have addressed the question of DA/ACh interaction in reward-based decision making tasks, and provided important data (Martyniuk et al. 2022), the fifth sentence of the abstract should be amended: Whether and how this circuit contributes to striatal function in vivo remains *largely* unknown. Same comments applies to the introduction (line 33) and discussion (line 475).

From the methods, Figure and results sections, it remains unclear what an omission trial is. Is it a

trial where the center port LED does not signal the initiation of a trial? If this is the case, is that data from events, when mice just poked in the left/right port without trial initiation? Thus, mice should not get the reward in omission trial. However, the neural DA and ACh responses look similar to rewarded trials. This cue omission should be described clearly.

Finally, as suggested, it would be valuable to compare DA and ACh signals at SE between trial-SE and ITI-SE. This would provide more information than comparing rewarded and unrewarded trials, as for both rewarded and unrewarded trials, mice were expecting the reward, and the differential of signal is due to the reward consumption. Computing the differential signal between SE of unrewarded-trials and of ITI would allow to differentiate signals related to the motor action of going to the port, from the signal related to the decision to retrieve a reward after the trial onset, based on the probability of getting the reward.

If animals do not enter the side port during ITI, quantifying the correlation of the DA and ACh signals during the entire ITI would also be interesting, and could address whether these two signals are only correlated during reward consumption/expectation.

Overall, the manuscript contains high quality data interesting findings, but could still be synthesized to increase accessibility for a broad readership.

Minor comments _____

In all histological images, as the optic fibers diameter is 200 μm , the representation of the fiber on the histological images should have the same size as the scale bars (it is currently less than half of them).

It would be interesting to include an average learning curve of all mice over the 4 weeks of training within Extended Data Figure 1.

Line 70: An LED -> A LED (same in legend of Figure 1)

Line 71: That *the* mouse

Line 123: (VLS) (Extended... -> VLS, Extended...

Referee #3 (Remarks to the Author):

In their resubmission, the authors have added important quantitative analyses to support their conclusions. They have also taken important steps to clarify their rationale for certain strategies and their interpretation of findings.

I commend the authors for conducting the new analysis of epoch-specific effects of striatum-wide CIN TeLC expression on phasic DA release dynamics. As they note, broad changes in DA release were detected. From their discussion it is clear they understand the issues involved, but emphasize that the pronounced behavioral effects of this manipulation make data interpretation problematic. However, this discordant finding together with their data showing that in vivo optogenetic stimulation of CINs is sufficient to evoke DA release, begins to weaken the significance of the current study.

Evidence that local VLS CIN-mediated ACh release is not a major determinant of DA dynamics and outcome encoding is notable but I have some remaining questions and concerns.

First, CIN TeLC expression should chronically reduce local ACh tone, which may lead to compensatory changes in the regulation of phasic DA release. A more critical test would be to transiently disrupt CIN activity. The authors present in their rebuttal some evidence of an attempt at such an experiment using the inhibitory opsin Jaws, which posed some challenges. Of course, other approaches (chemogenetics, conventional pharmacology) would be suitable for transiently disrupting ACh transmission. In the absence of such data, it would be good for the authors to discuss the potential limitations of their chronic TeLC approach.

Another crucial consideration is the timing and behavioral significance of potential ACh regulation of DA release. The authors cite Mohebi et al. 2019 in their rationale because this earlier study argued that local neuromodulatory control over terminal DA release may account for divergence between striatal release and DA neuron firing. That study described both similarities and differences, mostly relating to occasions when increased DA release seemed to encode reward prediction or motivation (e.g., during port approach). First, it seems appropriate for the authors to highlight in the current study those features of dopamine release that were previously shown to differ from DA neuron firing patterns, since it is these DA release effects that are suspected to be modulated by ACh. Second, the current paper focuses heavily on phasic release dynamics and does not address the question of whether slower changes in DA release, potentially related to average reward rate, may be influenced by ACh release. Although one might argue that this goes beyond the scope of the current study, a lack of critical interrogation of the putative ACh-mediated regulation of DA release limits the significance and overall novelty of this work, particularly given that the Martyniuk et al. 2022 study already provides some in vivo evidence of another element of the current study, relating to DA's influence over ACh release.

Minor:

For the new paired metrics showing differences in release across hemispheres (Fig 16C), it would be good to clarify what left and right points correspond to. One can infer from other clues but more could be done to help readers.

Response to referees' comments with the authors' replies in blue

Referee #1

The authors have replayed my questions and the manuscript is now clearer. I congratulate them for this fantastic work

We thank the referee for reviewing the manuscript and for their kind feedback.

Referee #2

The revised manuscript contains important quantifications of the data presented in the study, which significantly strengthen the conclusions of the paper. The authors also made an effort in rewriting the manuscript to make it more accessible. However, some of the points were only partially addressed, and some methodological aspects remain unclear. These need to be tackled before the manuscript can be published.

We thank the referee for their thoughtful input on our work and for recognizing the power of the statistical analyses that we added. In this round, we have revised the manuscript to address their concerns about the points that were partially addressed.

Specific comments

The quantification presented in Extended Figure 4a and b is an important addition, but should be included in the main Figure 1, especially as this Figure includes unused white space. Moreover, it is unclear why the authors did not perform the average or max-min analysis for both DA and Ach. Uniformizing both analyses with min-max would make the results comparable across neurotransmissions, and increase accessibility of the results and interpretations.

We thank the referee for their suggestion to include analysis of the photometry signals in Figure 1. For the data in **Fig. 1d**, which compares DA and Ach signals during rewarded and unrewarded trials, we now move the statistical analysis of these signals to the adjacent panel (**Fig. 1e**). This enables side-by-side comparison of the data with the analysis, and it highlights the robust and consistent effects that reward outcome has on DA and Ach signals in the behavior task.

We agree with the referee that a common metric to analyze both DA and Ach signals would enable easier interpretation of the results; however, we found that different metrics are required to accurately represent the dynamics of DA versus Ach. The Δ max-min signal consistently captures the complex changes in Ach signals across rewarded and unrewarded outcomes (**Fig. 1e**) – Ach shows multiple peaks and dips that are task modulated but average out to near zero, rendering the mean uninformative. The same metric does not work for capturing changes DA signals across task conditions (**Rebuttal Fig. 1**, see below). In our study we use two DA sensors – dLight1.1 and rDAh. Both sensors have comparable signals, but there are differences in the amplitude and timing due to the slower kinetics and higher affinity for DA of rDAh compared to dLight1.1 (**Fig. 2a**). Because of these differences in sensor properties, the Δ max-min value for rDAh signals during rewarded versus unrewarded trials are very similar, as seen below for a representative mouse (**Rebuttal Fig. 1a**). Therefore, this metric fails to capture the effects of reward outcome on rDAh signals (**Rebuttal Fig. 1b**, top plot). In contrast, the mean signals for both rDAh and dLight transients does so robustly (**Rebuttal Fig. 1b**, bottom plot). As a result, we use the mean to quantify the task-evoked changes in all DA signals presented in the manuscript. Furthermore, we show the time courses of the mean and standard errors of fluorescence transients for all sensors in the figures, allowing the reader to directly observe and judge the significance of the differences in signals across conditions.

For clarity, we include a description for our use of two different metrics for DA versus Ach signals in the methods section (see underlined text, lines 881-884):

“... Because of the multiphasic dynamics of Ach transients, Δ Ach more accurately captures the differences in signals across conditions compared to the mean. In contrast, the mean is a more robust metric than Δ DA to quantify DA signals independent of which DA sensor was used due to differences in the kinetics of dLight1.1 versus rDAh.”

Rebuttal Fig. 1 | Comparison of metrics for quantitative analysis of DA signals

- a.** DA signals in rewarded and unrewarded trials for a representative mouse expressing dLight1.1 (left) and rDAh (right). The averaged Z scored sensor signal for the indicated trial type (bold line) is depicted with the standard error of the mean (SEM) (shaded region). Dashed lines denote the difference in the maximum and minimum DA signal for rewarded (purple) and unrewarded (orange) outcomes. The calculated Δ maximum-minimum for each signal is indicated.
- b.** Quantification and statistical analysis of DA signals from mice expressing dLight1.1 (green box) and rDAh (red box). For each mouse, paired comparisons (connected dots) are shown between rewarded trials (left dot in connected pair) versus unrewarded trials (right dot in connected pair), calculated from signals taken during the post side entry period. Open circles represent a significant difference for each paired comparison, and black circles represent an insignificant difference (two-sided t-test ($p < 0.05$)). Error bars denote standard deviation. The mean of the post-side entry signal captures the clear differences in rewarded vs. unrewarded transients seen with both sensors, whereas the Δ maximum-minimum does not.

Inclusion of the justification why experiments were performed only in males is important and need to be included in the rational of the study (i.e., in the introduction).

We have integrated the following (see underlined text) into the introduction (lines 22-25, line 51):

“CINs may also differentially regulate behavior across males versus females, as exploratory behavior in males, but not females, is affected by loss of muscarinic Ach receptors on DANs in the ventral tegmental area³⁰. Moreover, estrogen modulates cholinergic signaling through effects on choline acetyltransferase expression, Ach release, and choline transport³⁰⁻³⁶.”

“To provide a more complete understanding of what factors shape Ach release during reward-based decision making and how Ach release contributes to DA release, we examined striatal Ach and DA levels during a complex reward-based decision-making task in male mice.”

We also include detailed motivation of using male mice in the results (see underlined text) as we felt this was better integrated with our description of the behavior task conditions (lines 72-73):

“To examine the local circuit interactions between striatal Ach and DA (Fig. 1a) during, and their contributions to, such behaviors, we monitored neuromodulator levels in mice performing a dynamic and probabilistic two-port choice task modeled after paradigms that engage striatal pathways and require striatal activity for optimal performance^{54–56}. In our study, we exclusively used males because cholinergic signaling and its impact on behavior varies across males and females^{31–36}.”

The effort of the authors to include multiple images of a representative mouse for each mice in the Extended data Figures is highly valuable, and reflects the quality of the stereotaxic surgeries of the study. However, the insets still do not depict the plasma membrane expression of the sensors. The authors need to provide at least one high magnification (confocal) image of neurons expressing each of the sensors (green and red sensors for DA and ACh), even if the background signal is high (ideally this image should be included in the main Figure). Similarly, in Figure 3b, it is unclear what the arrows are pointing at (not defined in the legend) and epifluorescence images are blurry and do not allow to interpret the image.

We now include high magnification confocal images of neurons from a representative mouse with expression of the following sensors: (a) dLight1.1 for DA recordings (**Extended Data Fig. 2i**), (b) Ach3.0 for Ach recordings (**Extended Data Fig. 2j**), and (c) Ach3.0 and rDAh for simultaneous DA and Ach recordings (**Fig. 2c**). These images reveal that plasma membrane expression is present for all three sensors used in this study. Although the referee suggested that we incorporate these images into the main figures, we could not do this for dLight1.1 and Ach3.0 due to the lack of available white space in Fig. 1. Please note that we had already included analysis of sensor point mutants (**Extended Data Fig. 3a-3f**), which confirmed that the fluorescence transients we observe require interactions at the neurotransmitter binding site of these sensors.

We agree with the referee that the images in **Fig. 3b** are blurry, and we now replace them with clearer and higher resolution images of choline acetyltransferase and tetanus coexpression in CINs. We now indicate what the white arrows reference in the figure legend for **Fig. 3b**: *“White arrows denote two representative CINs that coexpress ChAT and TelC.”*

As previous studies have addressed the question of DA/ACh interaction in reward-based decision making tasks, and provided important data (Martyniuk et al. 2022), the fifth sentence of the abstract should be amended: Whether and how this circuit contributes to striatal function in vivo remains **largely** unknown. Same comments applies to the introduction (line 33) and discussion (line 475).

We thank the referee for pointing this out, and we have made the changes for all instances in the text, as underlined below:

“Whether and how this circuit contributes to striatal function in vivo remains largely unknown.” (abstract)

“Despite a detailed mechanistic understanding of the interaction between DA and Ach in vitro, if, when, and how these control DA and Ach levels to regulate striatal function in vivo are largely unknown.” (line 36)

“However, whether these interactions regulate neuromodulator levels in vivo, particularly during decision making, is largely unknown.” (line 498)

From the methods, Figure and results sections, it remains unclear what an omission trial is. Is it a trial where the center port LED does not signals the initiation of a trial? If this is the case, it that data from events, when mice just poked in the left/right port without trial initiation? Thus, mice should not get the reward in omission trial. However, the neural DA and ACh responses look similar to rewarded trials. This cue omission should be described clearly.

The referee is referring to **Fig. 1e**, which is now **Fig. 1f** in this revised version. We refer to these sessions as “cue-omission” sessions. In these we turn off the center and side port LEDs that, indicate, respectively that the mouse can initiate a trial and that the side ports are available for selection (see **Fig. 1b**; yellow circles refer to an LED that is on and gray circles refer to an LED that is off). All other task parameters remain the same, including reward probabilities. Thus, the mice can receive rewards in the absence of LED cues after performing the correct center port entry to side port entry motor sequence. As mice have received extensive training on the task, they can often perform the motor actions in the correct order without the LEDs. In **Fig. 1f**, rewarded trials that the mice perform in the absence of these LED cues are denoted as “2ABT cue: omitted” and rewarded trials in the presence of LED cues are denoted as “2ABT cue: present”. The referee correctly notes

that “the neural DA and Ach responses for the cue omission trials look similar to rewarded trials” because these responses are indeed from rewarded trials. However, although the signals have similar dynamics with and without active LEDs, their amplitudes are different, which is consistent with changes in reward expectation caused by the lack of cues: the magnitude of the signals prior to side port entry are reduced for DA but increased for Ach, while the signals post side port entry are increased for DA and decreased for Ach. These changes are statistically significant (**Extended Data Fig. 5a**). To clarify the details of this experiment, we modified our description of this experiment in the results section (see underlined text below, lines 166-174):

*“Further support for reward-outcome modulation of both DA and Ach is revealed during sessions in which the LED cues that signal activation of the center and side ports are turned off (i.e., “cue-omission” sessions), but all other task conditions (i.e., reward probabilities, task structure) remain the same. In this scenario, the mice can obtain rewards by performing the proper center port entry to side port entry sequence as in a normal trial; however, due to the absence of the LED cues, the rewards are less expected due to the absence of the LED cues that normally signal that the center and side ports are active. Compared to normal rewarded trials with LED cues present, the DA transient in cue-omission rewarded trials is significantly increased after side port entry but decreased before side entry (**Fig. 1f, Extended Data Fig. 5a**). Ach transients are also affected by cue omission, although in the opposite direction to DA transients (**Fig. 1f, Extended Data Fig. 5a**).”*

We include additional clarification in the figure legend for **Fig. 1f**:

“DA and Ach release during rewarded trials in which the LED cues that signal center and side port activation are present or omitted during the 2ABT. Data are depicted as in (d) (n = 4 mice).”

Finally, we include an additional description of this experiment in the methods for clarification (lines 748-749):

“In experiments in which the LED cue is omitted (“cue-omission” trials), we turned off the LEDs located above the center and side ports but left all other task parameters and recording conditions unchanged.”

Finally, as suggested, it would be valuable to compare DA and ACh signals at SE between trial-SE and ITI-SE. This would provide more information than comparing rewarded and unrewarded trials, as for both rewarded and unrewarded trials, mice were expecting the reward, and the differential of signal is due to the reward consumption. Computing the differential signal between SE of unrewarded-trials and of ITI would allow to differentiate signals related to the motor action of going to the port, from the signal related to the decision to retrieve a reward after the trial onset, based on the probability of getting the reward. If animals do not enter the side port during ITI, quantifying the correlation of the DA and ACh signals during the entire ITI would also be interesting, and could address whether these two signal are only correlated during reward consumption/expectation.

We agree with the referee that a comparison of side entry aligned DA and Ach signals during unrewarded trials versus during the ITI is meaningful. During the ITI, reward is not available, but it is difficult to know if it is “expected” by the mouse as side port entry has been associated with reward during training; thus, the signals during the ITI likely reflect a combination of motor and expectation signals.

We perform this new analysis as requested by the referee. We analyzed DA and Ach signals associated with center and side port entries that occur during the ITI and are separated from other behavioral events (all other port entries and exits) by at least 15 time sample entries (0.81 seconds). As the motor action is similar for center and side port entry, comparison of these signals may help identify if there are both motor and expectation components to the signals. For both DA and Ach signals, we perform comparisons (**Extended Data Fig. 5b**) between the following sets of signals: (a) those evoked by normal side port entries in unrewarded trials versus during the ITI; (b) those evoked by center port entry in unrewarded trials versus during the ITI. We include quantitative analysis of these signals in **Extended Data Fig. 5c**.

Upon side port entry, DA and Ach signals change to a greater degree during unrewarded trials than during the ITI. DA signals dip more while Ach signals rise more after side port entry of unrewarded trials, whereas the opposite happens prior to side port entry (**Extended Data Fig. 5b**, left column). Likewise, the dynamics of DA and Ach signals diverge greatly and in complex ways upon center port entry during the ITI versus during an unrewarded trial (**Extended Data Fig. 5b**, middle column). Notably, center port and side port entries during the ITI evoke very different DA and Ach dynamics, suggesting that the motor action of port entry alone is not

sufficient to explain these signals (**Extended Data Fig. 5b**, right column). Altogether, this reveals that while the motor action contributes to some components of DA and Ach signals around center and side port entry, reward expectation shapes these signals in complex ways. We integrate these findings into the text (lines 175-186) and describe our workflow to extract the ITI signals from our recordings in the methods (lines 841-846).

As a complementary method to analyze ITI signals, we also assessed the GLM kernels for side entry events that occur within an unrewarded trial and during the ITI (“SE” and “SEnt”, respectively in **Extended Data Fig. 8a**). For easier comparison of these two events, we repositioned the kernels and reconstructed signals for SE and SEnt so that they are now adjacent, and we include an overlay below for clarity (**Rebuttal Fig. 2**). These kernels represent the time-dependent contribution of behavioral events to DA and Ach signals, and they reveal that reward outcome greatly influences side port entry signals. The contribution of the side entry event to the neuromodulator transient is much smaller during the ITI than during an unrewarded trial. For the DA kernel, only a small dip is observed at positive timeshifts, and the second peak is greatly reduced for the Ach kernel. This corroborates our analysis above and demonstrates that motor action may contribute to the DA and Ach signals, but reward outcome further modulates this signal in significant ways.

Rebuttal Fig. 2 | Comparison of GLM modelling of side entry signals associated with unrewarded trials and the ITI. Kernels (left) and side entry (SE) aligned reconstructed and true signals (right) for the base GLM model. SE refers to side entry and SEnt refers to side entries that occur during the inter-trial-interval. The average signal (bold line) and 95% confidence interval (shaded region) are depicted (n = 8 mice).

Finally, as the referee suggested, it is important to assess the correlation of DA and Ach signals during the ITI. Indeed, this analysis can be found in **Fig. 2f**, which we annotate below for clarity (**Rebuttal Fig. 3**). During unrewarded trials, the mice rapidly exit the side port and the signal approximately one second following this event occurs during the ITI, which we highlight with a white arrow. The ITI signal is also present prior to side port entry, which we denote with an orange arrow. DA and Ach remain highly anticorrelated during this period and to the same degree as they are during reward consumption and expectation. We now highlight this point in the results (see underlined, lines 302 and 307):

*“This revealed a strong time-lagged negative covariance (**Fig. 2e**, strong blue signal), which we call the off-diagonal (**Fig. 2f**) showing that, at most time points within the trial and during the ITI, changes in DA precede changes in Ach by approximately 100 ms (**Extended Data Fig. 10g**), which is consistent with prior analysis (**Extended Data Fig. 10f**). Intriguingly, this time dependent analysis reveals that the negative, off-diagonal covariance nearly disappears when the animal enters the side port in both rewarded and unrewarded trials (**Fig. 2e**, insets). This analysis highlights the dynamic and context dependent relationship between DA and Ach within the trial and during the ITI.”*

Rebuttal Fig. 3 | Comparison of GLM modelling of side entry signals associated with unrewarded trials and the ITI. Full time-dependent covariance analysis of DA and Ach signals in unrewarded trials, taken from Fig. 2f. Arrows denote the ITI period before trial initiation (yellow arrow) and after the end of a trial (white arrow). The average DA and Ach signals (bold lines) with SEM (shaded region) are shown within the top and left subplots, respectively. An enlarged inset of the region outlined in the white box is shown to the right of each matrix, with the time (sec) indicated on the bottom of the inset (n = 6 mice).

Overall, the manuscript contains high quality data interesting findings, but could still be synthesized to increase accessibility for a broad readership.

We thank the referee for this feedback. We made small changes throughout the manuscript that, we hope, continue to increase the accessibility of our study.

Minor comments

In all histological images, as the optic fibers diameter is 200 μm , the representation of the fiber on the histological images should have the same size as the scale bars (it is currently less than half of them).

We thank the referee for this observation, and we have corrected the sizes of all fiber implant representations in the histological images.

It would be interesting to include an average learning curve of all mice over the 4 weeks of training within Extended Data Figure 1.

The goal of our study is to understand how DA and Ach signals contribute to decision making. To do so, we used a dynamic and probabilistic two-port choice task in which mice engage striatal pathways to achieve a stable and comparable performance across multiple expert mice (see performance metrics in **Extended Data Figure 1b, 1c, 1d, and 1g**). This task was not designed to study learning and the training trajectory of each mouse varies due to bespoke training steps. Although all mice take ~ 4 weeks to reach proficiency in the 2ABT, there is very high variability in learning rates across different mice and across the different stages of training. We agree that studying the effects of manipulating striatal DA and Ach on learning is a fascinating topic. However, performing this requires a behavior that is learned through a highly stereotyped process. In future studies, it will be of great interest to assess how DA and Ach contribute to learning in the context of a behavior assay designed to enable comparable learning rates across mice.

Line 70: An LED -> A LED (same in legend of Figure 1)

We have corrected this.

Line 71: That *the* mouse

We have corrected this.

Line 123: (VLS) (Extended... -> VLS, Extended...

We apologize, but we are unsure what the referee requested us to change. If the referee is referring to the lack of clarity for the VLS abbreviation, we made edits to line 123, which is now line 127 (see underlined text), to make this clearer:

“...expressed in separate hemispheres within the ventrolateral portion of the dorsal striatum (abbreviated as VLS) (Extended Data Fig. 2a-d). The VLS is a region associated with controlling the behavior of mice in reward-based decision-making tasks⁶¹”

Referee #3

In their resubmission, the authors have added important quantitative analyses to support their conclusions. They have also taken important steps to clarify their rationale for certain strategies and their interpretation of findings.

We thank the referee for their input.

I commend the authors for conducting the new analysis of epoch-specific effects of striatum-wide CIN TeLC expression on phasic DA release dynamics. As they note, broad changes in DA release were detected. From their discussion it is clear they understand the issues involved, but emphasize that the pronounced behavioral effects of this manipulation make data interpretation problematic. However, this discordant finding together with their data showing that in vivo optogenetic stimulation of CINs is sufficient to evoke DA release, begins to weaken the significance of the current study.

Evidence that local VLS CIN-mediated ACh release is not a major determinant of DA dynamics and outcome encoding is notable but I have some remaining questions and concerns.

We appreciate the referee's praise, appreciation for the new experiments and analyses, and thoughtful input during the review process. Below are our responses to their questions and concerns.

First, CIN TeLC expression should chronically reduce local ACh tone, which may lead to compensatory changes in the regulation of phasic DA release. A more critical test would be to transiently disrupt CIN activity. The authors present in their rebuttal some evidence of an attempt at such an experiment using the inhibitory opsin Jaws, which posed some challenges. Of course, other approaches (chemogenetics, conventional pharmacology) would be suitable for transiently disrupting ACh transmission. In the absence of such data, it would be good for the authors to discuss the potential limitations of their chronic TeLC approach.

The referee highlights an important point that chronic and irreversible disruption of CIN activity via tetanus toxin expression may alter Ach tone in the VLS, a caveat that can be addressed with more transient methods of CIN inhibition. However, it is also worth pointing out that Ach release was inhibited after the mice have learned the task as adults, and, given our estimates that it takes about ~10 days for neurotransmitter release to be inhibited after AAV injection, and this inhibition remains for about 20 more days prior to photometric and behavioral analyses. Thus, the approach is not trial-to-trial, but neither is it a lifelong chronic manipulation. Nevertheless, we agree that a trial-to-trial or session-to-session inhibition strategy would have been powerful, and we pursued a variety of such approaches, which we outline below.

First, we attempted to optogenetically inhibit the CINs, but this led to rebound excitation that makes interpretation of the results difficult. Due to optical cross talk issues and auto-fluorescence generated by the optogenetic laser, it was impossible to use continuous illumination to optogenetically inhibit CINs while performing photometry, which might have prevented rebound firing during pulsed inhibition.

Second, we tried to chemogenetically inhibit the CINs via expression of hm4di DREADD. Despite extensive troubleshooting with regards to hm4di expression levels and hm4di agonist injection parameters (i.e., dose, concentration, timing of recordings post injection), in our hands pharmacological activation of the inhibitory DREADD did not reduce Ach levels in the VLS as measured by photometric recordings. Therefore, we could not pursue this approach further.

Finally, conventional pharmacology is another approach to transiently perturb this circuit, but we did not use this method for several reasons. First, two independent studies already used pharmacology to examine if CINs contribute to DA release in the striatum, but they reached opposing conclusions. Using local infusion of a $\beta 2$ nAChR subunit inhibitor *in vivo*, Krok et al. (*bioRxiv* 2022) found that phasic DA release was not altered whereas Liu et al. (*Science* 2022) observed the opposite (see manuscript references 41 and 69). We believe that these pharmacological infusion experiments may suffer from the difficulty in performing proper positive and negative controls *in vivo*. It is technically challenging to validate that the $\beta 2$ -containing nAChRs on DAN axons are inhibited and that no other such receptors within and beyond the striatum are affected. This is due to the lack of control over the spread of the drug within the brain. This is of particular concern given that the $\beta 2$ subunit is expressed in many cell types in the striatum and across the brain, including the lateral geniculate nucleus and superior colliculus, amongst others. We discuss these limitations in the discussion as the referee suggested (lines 518-523), which we copy below:

“Loss of CIN-mediated Ach release following chronic expression of tetanus toxin may lead to compensatory changes in the regulation of DA release. Although pharmacological inhibition of cholinergic effects on DAN axons using a $\beta 2$ nAChR antagonist has been used to transiently inhibit this circuit, opposite conclusions of the effects on DA signals were observed across studies^{41,69}. It will be informative to resolve these conflicting findings with alternate strategies that rapidly and reversibly inhibit CINs.”

Another crucial consideration is the timing and behavioral significance of potential ACh regulation of DA release. The authors cite Mohebi et al. 2019 in their rationale because this earlier study argued that local neuromodulatory control over terminal DA release may account for divergence between striatal release and DA neuron firing. That study described both similarities and differences, mostly relating to occasions when increased DA release seemed to encode reward prediction or motivation (e.g., during port approach). First, it seems appropriate for the authors to highlight in the current study those features of dopamine release that were previously shown to differ from DA neuron firing patterns, since it is these DA release effects that are suspected to be modulated by ACh. Second, the current paper focuses heavily on phasic release dynamics and does not address the question of whether slower changes in DA release, potentially related to average reward rate, may be influenced by ACh release. Although one might argue that this goes beyond the scope of the current study, a lack of critical interrogation of the putative ACh-mediated regulation of DA release limits the significance and overall novelty of this work, particularly given that the Martyniuk et al. 2022 study already provides some *in vivo* evidence of another element of the current study, relating to DA's influence over ACh release.

We thank the referee for bringing up the important point that not all DA signals are proposed to be modulated by CINs, which we now address in our manuscript. Mohebi et al. revealed that there are mismatches in the firing of DANs and the release of DA in the striatum on multiple timescales, which might be explained by spike-independent modulation of DANs by CINs. These mismatches occur during motivated approach behaviors in a bandit task, specifically at trial initiation and just prior to food port entry. They are also present on a longer timescale when reward history shapes DA release. Specifically, DA levels are higher in more recently rewarded trials, suggesting value encoding by DA, but DAN firing rates remain constant. These findings conflict with recent studies that reveal well-matched axonal and somatic signaling in DANs (see manuscript references 48 and 49).

Unlike our study, the Mohebi study does not monitor ACh signaling and the relationship between ACh and DA on a rapid time scale (they use ~ 1 min sampling), and they perform no perturbations of ACh or DA to examine the effects on the other neuromodulator. In fact, the word ‘acetylcholine’ appears in the manuscript only once. Thus, although the Mohebi et al study is an important one, it is hardly a study of ACh and ACh/DA interactions and, as the reviewer states, only “suggests” periods in which CINs might modulate DA release.

To define the role of CINs in regulating DA release in striatum, we used a task in which mice perform flexible and dynamic decision making and DA dynamics encode evolving reward value and expectation. During moments of motivated approach analogous to those in the Mohebi study in which there is a mismatch between DA release and DAN firing (i.e., center port entry and immediately prior to side port entry), we find that loss of ACh release did not significantly alter DA dynamics (**Fig. 3f**). Therefore, phasic DA release during the 2ABT does not require ACh release.

Next, we analyzed whether CINs may impact DA release on a longer timescale, particularly with respect to the encoding of reward rate. We parsed DA signals by the choice and reward outcome of the previous trial, which

reveals changes in the encoding of reward expectation by DA that are analogous to the Mohebi study (**Fig. 1g and 1h**). However, across all possible choice and reward outcomes, VLS-selective loss of Ach release did not impact DA signals (**Extended Data Fig. 15d, 16b, and 16d**). To address if Ach can modulate DA dynamics on an even longer timescale, we extended our analysis of DA signals parsed by the choice and reward histories of two and three trials back. For instance, “lose-win-win” means we look at the DA signal of a rewarded trial that is preceded by a loss and then a win (**Extended Data Fig. 16e**). Importantly, we analyzed “lose-win-win” rather than “win-win-win” signals because the former ensures that we look at a unique subset of trials (i.e., “win-win-win” trials also include previously analyzed “win-win” trials). We did not perform this analysis for more than three trials back due to insufficient numbers of trials for longer history sequences. Regardless of history length, we found that DA signals are not significantly different when Ach release is ablated in the VLS (**Extended Data Fig. 16e**). Taken together, while CINs are proposed to modulate DA release during multiple contexts and timescales of decision making, we do not find evidence for Ach-dependent regulation of DA release in the VLS.

We incorporate these findings in the revised manuscript as follows (see underlined text):

In the introduction (lines 42-43):

“The ability for CINs to evoke DA release independently of somatic firing has been proposed as a mechanism to explain differences between DAN somatic activity and striatal DA levels during motivated approach behaviors and during reward value encoding on a longer timescale⁴⁷; however, recent studies report robust correlation between somatic and axonal signaling^{48,49}.”

In the results (lines 362-371):

“Surprisingly, VLS-specific loss of Ach release did not impact DA dynamics in rewarded or unrewarded trials recorded with dLight1.1 (Fig. 3f) or with rDAh in recordings in which we simultaneously validated the suppression of Ach release (Extended Data Fig. 16a, 17a, and 17b). Notably, phasic DA release remained the same during motivated approach behaviors (i.e., around center port entry and immediately prior to side port entry), which are proposed to be instances in which DA levels and DAN activity are poorly correlated, suggesting time periods during which CINs may act to drive DA release⁴⁷. Quantitative analysis supports our observations – the changes in mean DA across the two hemispheres are variable and lead to poor LDA classification performance (Extended Data Fig. 16c). To address whether CINs might mediate the discrepancies between DAN firing and DA release across longer timescales⁴⁷, we parsed DA signals by the choice and reward outcome histories of one, two, and three trials back (Extended Data Fig. 15d; Extended Data Fig. 16b and 16d and 16e), but we did not observe any significant differences in DA dynamics or the underlying GLM kernels of DA for each input feature upon loss of Ach (Extended Data Fig. 15e).”

In the discussion (lines 511-515):

“A mismatch in the firing of DANs and the release of DA in the striatum, notably during motivated approach behaviors and during reward value encoding by DA over extended timescales, suggests that there may be spike-independent modulation of DA release. Because synchronous CIN activation robustly triggers DA release in vitro, CINs are a prime candidate to drive DA release in striatum independently of the cell body activity of DANs. However, we did not find evidence that this effect is functionally important in vivo in the behavioral context of a reward-guided behavior: DA reward-encoding dynamics are unaffected by loss of Ach release in the VLS, and present even in the absence of Ach release throughout the entire striatum.”

The referee notes that the Martynuik study limits the significance and overall novelty of our findings. While Martynuik et al. used the same strategy as in our study to generate CIN specific deletion of D2Rs in a reward-based task, their work does not impact the novelty of our observations for several reasons. First, the behavioral context is fundamentally different in the two studies. Although both behavioral assays rely on reward-based decision making, our mice must do so in a probabilistic context with periodic switches of reward probability. This enables us to reveal effects of choice and reward history on DA and Ach signals that Martynuik et al. could not examine. Second, our findings differ in a crucial way from those obtained by Martynuik and colleagues. We find that Ach pauses are abolished whenever they coincide with DA increases, whereas Martynuik et al. found that these pauses were reduced, not gone. Our findings are consistent with the proposed *in vitro* circuit in the field and are important to share with the neuroscience community, especially given the long-standing debate about the mechanisms that underlie Ach pauses *in vivo*. Third, we observe different phases of Ach modulation in our study, notably those that are DA dependent and independent. Our work establishes that Ach pauses during behavior are diverse, and they may have important contextual or functional dependencies. Our work also reveals an important point that different mechanisms underlie what the field has collectively referred to as an Ach ‘pause’. This is of interest to the broader neuroscience community as Ach pauses have long been

observed but their origins *in vivo* remain unclear. Finally, our work provides insights beyond just the impact of D2Rs on Ach release but addresses fundamental questions in the field as to what is encoded in Ach transients, whether Ach release influences DA release during behavior, and how cortical and thalamic inputs into striatum shape Ach dynamics.

Lastly, referencing our experiment in which we perform striatum-wide loss of Ach and assess the effects on DA release (**Fig. 3e**), the referee mentioned that “*broad changes in DA release were detected. From their discussion it is clear they understand the issues involved, but emphasize that the pronounced behavioral effects of this manipulation make data interpretation problematic. However, this discordant finding together with their data showing that in vivo optogenetic stimulation of CINs is sufficient to evoke DA release, begins to weaken the significance of the current study.*” While we can evoke DA release in a CIN-dependent manner, this is a gain-of-function experiment that only allows us to conclude that this circuit is *sufficient* to affect DA release, but not whether it is *necessary and relevant* in a physiological context. Loss-of-function experiments such as those using CIN-selective tetanus toxin expression allow us to make an important conclusion that local VLS-selective Ach release is not required for phasic DA release, which is an important and novel finding.

Minor:

For the new paired metrics showing differences in release across hemispheres (Fig 16C), it would be good to clarify what left and right points correspond to. One can infer from other clues but more could be done to help readers.

For this figure, the left point in the paired metric is DA release when CINs express TelC and the right point is DA release when CINs express mCh (control). We have added the following text to the plot for increased clarity: “*DA signals in VLS expressing TelC vs. mCh*”.

We also clarify this in the figure legend: “*DA signals recorded from hemispheres with CINs expressing TelC (left dot in connected pair) or mCh (right dot in connected pair)*”.

For additional clarity in all quantitative analyses of the photometry signals, we have now edited the figure legends to specify what the left versus right dot in the connected pairs represent.

Reviewer Reports on the Second Revision:

Referees' comments:

Referee #2 (Remarks to the Author):

Overall the authors very carefully and appropriately addressed all the points I raised, and should be commended for their effort to perform experiments at high standards and to convey them in a clear way. I only have minor suggestion to even improve slightly the clarity of their presentation.

Regarding the cue-omission trials, it might be clearer to call them "2ABT w/o cues" and "2ABT with cues".

Figure 1d: I would recommend to move the legend (rewarded / unrewarded to the top of the diagram, and add horizontal bars to highlight the pre-(side entry) side and post-(side entry) between that legend and the average traces. Moreover, it would also be also be very helpful to integrate part of the information on Extended data Figure 4a into this panel.

Figure 1e: Include 'comparison' within the blue box: rewarded vs. unrewarded trial comparison
Put the pre-(side entry) comparisons before the post-(side entry) one
Add the pre-(side entry) and post-(side entry), on the top of panel 1d and 1e, using the same color code

Instead of writing the mouse ID, only write the performance of the mouse (%) and organize the mice depending on their LDA accuracy. Please add an x-axis title: Mouse LDA accuracy (%)

Figure 2d-h: add the unit for all y-axis scale bars (Z) – sec can also simply be abbreviated 's'

Figure 4: stGtACR2 activation wavelength is bleu – It would be helpful to match the opsin sensitive color to the diagram (red for Chrimson and blue for stGtACR2). To ease understanding for the readers, adding VTA excitation and VTA inhibition below Chrimson stim and stGTACR2 stim would be helpful.

Abstract: in vivo role of this circuit \ role of this circuit in vivo

Finally, regarding the minor (VLS) (Extended... -> VLS, Extended... comment, I meant that it is smoother to not close and open parentheses in a row. I apologize this was not clear.

Response to referees' comments with the authors' replies in blue

Referee #2

Overall the authors very carefully and appropriately addressed all the points I raised, and should be commended for their effort to perform experiments at high standards and to convey them in a clear way. I only have minor suggestion to even improve slightly the clarity of their presentation.

We thank the referee for their kind feedback and suggestions. We have addressed their remaining suggestions below.

Regarding the cue-omission trials, it might be clearer to call them “2ABT w/o cues” and “2ABT with cues”.

We have made the change in **Fig. 1f** to the figure labels as suggested.

Figure 1d: I would recommend to move the legend (rewarded / unrewarded) to the top of the diagram, and add horizontal bars to highlight the pre-(side entry) side and post-(side entry) between that legend and the average traces. Moreover, it would also be very helpful to integrate part of the information on Extended data Figure 4a into this panel.

We have moved the legend to the top of the diagram and added bars to indicate the pre and post side entry period in this panel.

We agree that it would be helpful to integrate **Extended Data Figure 4a**, which describes the quantification of DA and Ach signals, into the main figure; however, due to space constraints we will keep this panel in the extended data section to ensure that the central data panels are included in the main figure.

Figure 1e: Include ‘comparison’ within the blue box: rewarded vs. unrewarded trial comparison
Put the pre-(side entry) comparisons before the post-(side entry) one
Add the pre-(side entry) and post-(side entry), on the top of panel 1d and 1e, using the same color code
Instead of writing the mouse ID, only write the performance of the mouse (%) and organize the mice depending on their LDA accuracy. Please add an x-axis title: Mouse LDA accuracy (%)

We have made the changes to the ‘comparison’ label. We moved the pre-side entry analysis to now be before the post side entry analysis, and we have maintained the same color code to annotate pre and post side entry in **Fig. 1d** and **1e**.

We appreciate the referee’s suggestions regarding LDA accuracy. In our opinion, including the mouse ID is valuable information as it not only enables greater transparency but also allows the reader to better interpret the data as they know which signals come from which mice.

Given that the LDA accuracy is relatively similar across the indicated comparisons (i.e., for pre side entry, 12 out of 14 LDA values are within 52-59%), we do not think that sorting by LDA accuracy will be informative. Instead, preserving the order of the mouse identity is more meaningful to allow for ease of comparison between the pre and post side entry analyses.

Figure 2d-h: add the unit for all y-axis scale bars (Z) – sec can also simply be abbreviated ‘s’

Thank you for this suggestion. The y-axis values for **Fig. 2d** and **2e** are not Z-scores but rather correlation between two Z-scored signals, which does not have a unit. For the **Fig. 2f** and **2h**, we would like to keep the 'sec' abbreviation to be consistent with all our other figures.

Figure 4: stGtACR2 activation wavelength is bleu – It would be helpful to match the opsin sensitive color to the diagram (red for Chrimson and blue for stGtACR2). To ease understanding for the readers, adding VTA excitation and VTA inhibition below Chrimson stim and stGTACR2 stim would be helpful.

The stimulation was performed in both the SNc and VTA, not just the VTA. For clarity, we have instead added 'optogenetic stimulation of DANs' in the schematic of **Fig. 4a**. While we agree that matching the opsin colors would be helpful, for this particular figure we use red and blue extensively to denote the 'Drd2 f/f' and 'Drd2-cKO' genotypes. We feel that using a unique color to represent the opsin stimulation is clearer in this context.

Abstract: in vivo role of this circuit \ role of this circuit in vivo

Thank you for this suggestion. We have made the change.

Finally, regarding the minor (VLS) (Extended... -> VLS, Extended... comment, I meant that it is smoother to not close and open parentheses in a row. I apologize this was not clear.

Thank you for the clarification.